# Low-dimensional encoding of decisions in parietal cortex reflects long-term training history

Kenneth W. Latimer [1] ✉ & David J. Freedman [1]

Neurons in parietal cortex exhibit task-related activity during decision-making tasks. However, it remains unclear how long-term training to perform different tasks over months or even years shapes neural computations and representations. We examine lateral intraparietal area (LIP) responses during a visual motion delayed-match-to-category task. We consider two pairs of male macaque monkeys with different training histories: one trained only on the categorization task, and another first trained to perform fine motion-direction discrimination (i.e., pretrained). We introduce a novel analytical approach—generalized multilinear models—to quantify low-dimensional, task-relevant components in population activity. During the categorization task, we found stronger cosine-like motion-direction tuning in the pretrained monkeys than in the category-only monkeys, and that the pretrained monkeys' performance depended more heavily on fine discrimination between sample and test stimuli. These results suggest that sensory representations in LIP depend on the sequence of tasks that the animals have learned, underscoring the importance of considering training history in studies with complex behavioral tasks.

The activity of single neurons in the macaque lateral intraparietal area (LIP) encodes task-relevant information in a variety of decision-making tasks[1]. As a result, LIP has been proposed to support many different neural computations underlying perceptual decision making, including abstract visual categorization[2,3]. Throughout a lifetime, animals learn to make many different kinds of decisions in a variety of tasks and contexts, and different animals collect a unique set of experiences that shape their perceptual and decision-making skills and strategies[4,5]. In contrast, experiments designed to study neural mechanisms of decision making often focus on neurons recorded during a specific task in isolation. However, previously learned neural representations and strategies may impact how a cortical region is recruited when learning a new task. To understand the generality and flexibility of neural representations which support decision making, we aim to compare decision-related LIP activity in animals performing the same tasks, but with different long-term training histories.

In this study, we consider monkeys trained on two tasks using a delayed-match-to-sample paradigm which share the same structure, timings, stimuli, and motor response: a visual motion direction discrimination task and an abstract categorization task. On each trial, the monkey views two stimuli—sample and test—which are separated by a delay period. The monkey must decide if the motion direction of the test stimulus matches the direction of the sample stimulus according to a learned rule. We compare neural responses during categorization in monkeys that are only trained on the categorization task (category-only monkeys) to monkeys that are first trained on the discrimination task before learning the categorization task (pretrained monkeys).

It is plausible that extensive motion discrimination training during the discrimination task could have a lasting impact on the monkeys' visual motion sensitivity, accompanied by changes in neural responses to visual motion. Similarly, different training histories may lead to different behavioral strategies to perform the categorization task, and different strategies may be reflected by different cortical computations[6]. There are several reasons that LIP is a strong candidate region for showing experience-related changes in the representation of visual motion as a result of training on perceptual decision-making

---

[1]Department of Neurobiology, University of Chicago, Chicago, IL, USA. ✉e-mail: latimerk@uchicago.edu

and categorization tasks. First, while works from a number of groups have focused on LIP's role in planning saccadic eye movements and directing spatial attention[7], many studies have also shown that LIP contributes to the analysis of visual stimuli placed in neurons' response fields[8–11]. In particular, LIP neurons robustly respond to visual motion presented in their response fields[12,13], likely because LIP receives direct projections from upstream motion processing centers, such as the middle temporal (MT) and medial superior temporal (MST) areas[14,15]. Second, many previous studies have found category-selective responses in single neurons in LIP during abstract visual categorization tasks[16,17]. This contrasts with earlier sensory areas like MT which do not encode abstract category information even after extensive training[16,18]. However, the causal contribution of MT to motion-direction discrimination[19] and coarse depth discrimination[20] depends on training history. Third, many LIP cells show persistent category-related activity during the delay period which could contribute to working memory as part of a network that includes other regions that support categorization, such as the prefrontal cortex (PFC)[21–24]. Previous studies have found that delay-period activity in LIP depends on training[13], and that stimulus encoding and working-memory dependent sustained-firing activity in PFC during cognitive tasks depends on training[25,26]. Finally, inactivation experiments demonstrate that LIP plays a causal role in sensory evaluation in motion categorization tasks[27]. We therefore hypothesize that differences in training history could result in differences in LIP population activity during the categorization task which reflect behaviorally relevant aspects of the neural computations underlying performance of the categorization task studied here[2,28]. Specifically, we aimed to determine how training regimes may influence the encoding of specific motion-direction information beyond category in LIP, the geometry of mixed selectivity of direction and category tuning in the population, and signatures of working memory-related dynamics during the delay period.

Determining how neural populations implement the computations involved in the categorization task—how sample category is computed and then stored during the delay period, or how the test stimulus is compared to the sample—based solely on single-neuron analyses may be obscured by the mixture of category- and direction-related responses in individual neurons (e.g., by analyzing tuning curves of single neurons). We therefore take a dimensionality reduction approach to compare the low-dimensional geometry of population responses to better illuminate how LIP encodes different task variables[29]. Disentangling population responses that encode motion direction (which may reflect bottom-up, sensory-driven signals) from activity encoding correlated cognitive variables (category or match/non-match computations) could support comparisons of LIP responses across animals better than only characterizing the combined response. However, because category and direction are correlated, separating stimulus-driven responses from the higher-level computations that occur at each stage of the trial requires analyzing LIP responses to both the sample and test stimuli across all trials. Our analysis must also include trials with variable timings and lengths because the monkey may respond (and end the trial) anytime during the test stimulus presentation. In addition, we sought to examine single-trial activity for signatures of working memory-related dynamics, not just trial-averaged responses.

Here, we introduce the generalized multilinear model (GMLM) as a model-based dimensionality reduction method for quantifying population activity during flexible tasks. By applying the GMLM to the LIP populations, we quantified population-level differences in LIP activity between animals and compared those differences with behavioral performance with respect to the animals' training histories. We observed category and direction selectivity in LIP during the categorization task in all subjects. However, we found stronger cosine-like motion-direction tuning in LIP (which is consistent with bottom-up input from motion processing areas like MT) during the categorization

task in monkeys first trained on the discrimination task compared to monkeys trained only on categorization. During the test stimulus presentation when the monkeys had to compare the incoming test stimulus to the remembered sample stimulus, sample category could be more reliably decoded from LIP responses irrespective of the test-stimulus direction in the category-only monkeys than in the pretrained monkeys. Behaviorally, the pretrained monkeys were more likely to make categorization errors when the sample and test stimuli were similar than the category-only monkeys. In addition to the effects observed in the mean population response, we examined the structure of single-trial variability in the population which could reflect signatures of different strategies or neural computations in the task. Specifically, we found a difference in oscillatory, single-trial dynamics during the delay period (when the sample stimulus must be stored in working memory) between the category-only and pretrained monkeys by introducing a novel dynamic spike history component to the GMLM. Together these results suggest that different subjects may recruit distinct behavioral and neuronal strategies for performing the categorization task, and that long-term training history may play a role in shaping these differences. Low-dimensional encoding of the categorization task in LIP may therefore reflect an animal's unique training history or a particular task strategy (or both).

## Results
### Tasks and behavior
We examined LIP recordings in four monkeys performing two related tasks in which they were required to determine if sequentially presented motion directions matched according to a learned rule (Fig. 1A). In both tasks, the monkey viewed two random dot motion stimuli (sample and test) separated by a delay period. To receive a reward, the animal responded by releasing a touch bar if the two stimuli match or by continuing to hold the touch bar on non-match trials. On non-match trials, the first non-matching test stimulus was followed by a brief delay and a second test stimulus which always matched the sample category (requiring the monkey to release the touch bar). The delayed-match-to-sample task (discrimination task) was a memory-based, fine-direction discrimination task in which the sample and test motion stimuli matched only if they are in the exact same direction. By contrast, the delayed-match-to-category task (categorization task) was a rule-based task in which the stimuli matched if the directions belong to the same category (red or blue) according to a learned arbitrary category rule. In the categorization task, two matching stimuli could be nearly 180° apart but belong to matching categories, while neighboring directions on different sides of the category boundary did not match. The tasks therefore used the same structure and stimuli, and many of the sample-test pairs were rewarded for the same responses in both contexts (e.g., sample and test stimuli of the same direction match in both tasks). However, they required performing different perceptual and/or cognitive computations.

In one pair of monkeys B and J[22], the monkeys were trained only to perform the categorization task (i.e., without first training the monkeys on fine discrimination), and LIP recordings were made after training was completed (category-only populations). The second pair of monkeys pretrained monkeys D and H[13], was first trained extensively on the discrimination task, and LIP recordings were obtained after training (discrimination populations). The monkeys were then retrained on the categorization task, and a set of LIP recordings was made during an intermediate stage of training (when the monkeys' performance stabilized; category-early populations). After the category-early recordings, the monkeys received additional training which overemphasized near-category-boundary sample stimuli (the most difficult conditions where the monkeys' performance was lowest) so that the monkeys' performance increased. After the second training stage was complete, a final set of LIP neurons was recorded during the categorization task (category-late populations). In this study, the monkeys did not

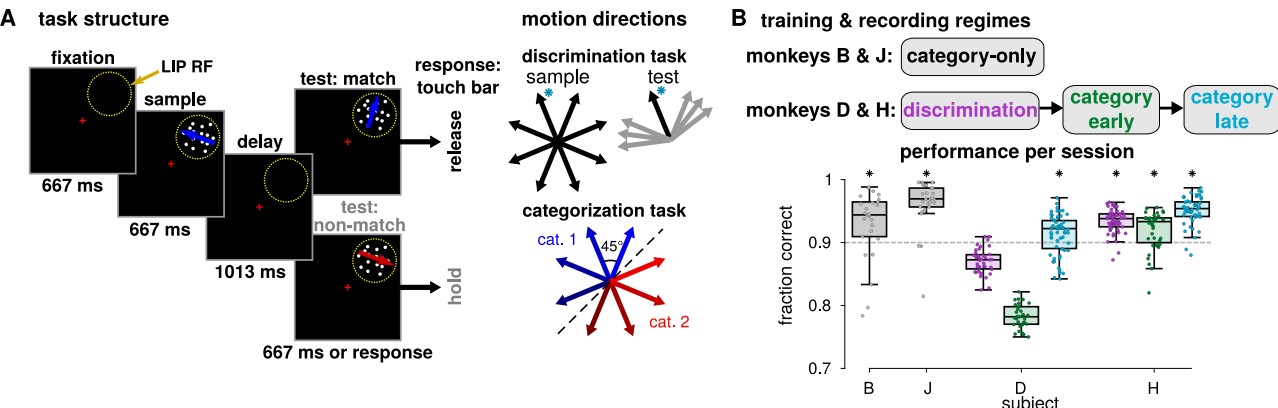

**Fig. 1 | Motion direction discrimination and categorization tasks. A** In both tasks, the animal fixated and viewed a motion direction stimulus (sample). Following a delay period, a second stimulus (test) was presented. The monkey signaled if the sample and test stimuli matched by releasing a touch bar, otherwise the monkey was required to hold the touch bar. In the discrimination task, the sample and test stimuli matched only if the directions were exactly the same. In the categorization task, the stimuli matched if they belonged to the same category: the motion directions were split into two equally sized categories with a 45°–225° boundary (red and blue directions; the boundary was constant for all sessions). The motion stimuli were placed inside the LIP cell's response field (yellow circle) during recording. **B** (Top) Training and recording regimes for the four monkeys. (Bottom)

Performance during each recording session (dots) for each animal is summarized by the box plots. Colors correspond to the task and training period (discrimination, category-early or late, and category-only). All four monkeys learned to perform the categorization task with a median per session performance of at least 90% per session, while chance level performance was 50% (asterisks denote $p < 0.01$, one-sided sign test, Holm–Bonferroni corrected; B $n = 26$, $p = 1.25 \times 10^{-3}$; J $n = 27$, $p = 2.46 \times 10^{-5}$; D-disc. $n = 39$, $p = 1$; D-early $n = 33$, $p = 1$; D-late $n = 59$, $p = 7.74 \times 10^{-4}$; H-disc. $n = 55$, $p = 4.28 \times 10^{-14}$; H-early $n = 40$, $p = 1.11 \times 10^{-3}$; H-late $n = 50$, $p = 1.13 \times 10^{-12}$). Box plots show the median and first and third quartiles over sessions and the whiskers extend to a 1.5 interquartile range from the edges.

perform both tasks during a single session; they were switched exclusively to the categorization task and retrained over the course of months. Both pairs of animals learned to perform the categorization task at high levels after training was completed (Fig. 1B). In total, we analyzed eight LIP populations recorded during sets of sessions at a particular task or training stage from four animals.

## Quantifying direction and category tuning in LIP

Single-neuron responses show a range of direction and category tuning across the individual neurons in the LIP datasets (Fig. 2A). We applied a receiver operating characteristic analysis for category selectivity in individual neurons. We found that the average category-selective tuning in the category-only population is greater than in the pretrained populations during the sample period (category-only vs pretrained category-late, rCTI increase of 0.051, 95% bootstrapped CI [0.044–0.059]; Fig. 2B). These results suggest differences in task representations in LIP may exist across the pairs of monkeys. Category tuning in the category-only monkey J was even stronger during the delay period compared to the sample stimulus period (one-sided bootstrap test; $p < 0.01$). In addition, monkey H showed significant category tuning during the delay period, but not during the sample period ($p < 0.01$). We found similar results using a parametric tuning curve model with cosine-direction tuning (Supplementary Fig. 2).

However, there are several challenges for teasing apart direction and category tuning in single neurons. For example, some neurons appear to exhibit changes in stimulus-direction preference during the course of the trial (e.g., Fig. 2A bottom, middle; the cell's preferred direction shifts from red to blue during the sample stimulus period). Traditional tuning curves using spike counts within a fixed window may therefore not capture dynamic representations in this task. In addition, category and direction are highly correlated: tuning to both variables may not be fully identifiable or separable given the responses of a single neuron in a single time window. Finally, the single-cell analysis does not demonstrate how the whole population supports the combination of category and direction encoding during the task[30]. We therefore next turned to population analysis approaches to better summarize the coding properties of a set of cells.

## Low-dimensional representation of direction and category in LIP

We sought to compare quantitatively the low-dimensional geometry of task variable encoding in LIP spike train responses during both sample and test stimulus presentations across the different populations (Fig. 3A). To accomplish this, we developed the generalized multilinear model (GMLM) as a dimensionality reduction approach that could analyze statistically the LIP population responses to each complete trial during the discrimination and categorization tasks (Fig. 3B). The GMLM is a tensor-regression extension of the generalized linear model (GLM) which describes a single neuron's spiking response to different task events through a set of linear weights or kernels[31–35]. By taking a regression approach, we directly quantify low-dimensional activity as a function of the task variables, thereby performing dimensionality reduction and statistical testing within a common framework[36]. The GMLM fits the data from all cells in a dataset into a compact representation by forming a low-rank tensor of linear kernels that best captures the common motifs in the mean responses to the task events —the sample and test stimuli and the touch-bar release—shared by the population. The neural populations used in this study were not simultaneously recorded and the model does not consider noise correlations between neurons. As the number of components increases, the GMLM will match GLMs fit to the individual cells. However, the GMLM's assumption of common response motifs in the population enables us to explicitly seek out low-rank structure that would not be directly recovered by single-cell GLMs. As with exponential family principal components analysis[37], the GMLM can be applied directly to spike train data rather than requiring smoothed spike rates. The GMLM inherits the GLM's flexibility for modeling trials with variable structure, including behavioral events like the touch-bar release. The statistical definition of the model can help account for uncertainty in the model fits[36]. Stimulus category and direction are low-dimensional variables, and motion direction tuning in sensory regions such as area MT can be well-captured by simple parametric models[38]. Therefore, the GMLM is well-suited for modeling how LIP populations represent combinations of these task variables during the categorization task.

The model's parameters include a set of stimulus components, where each component contains a single temporal kernel (or linear

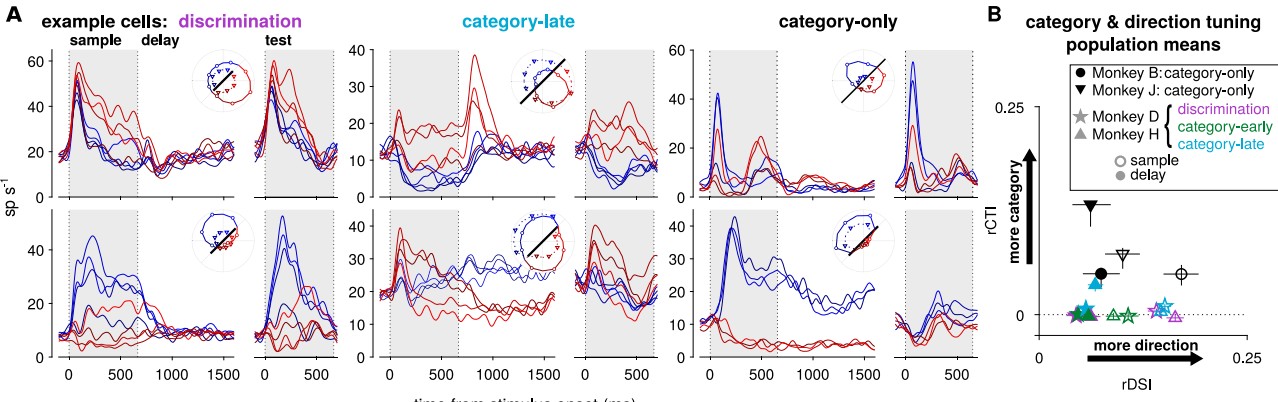

**Fig. 2 | LIP recordings during discrimination and categorization tasks. A** Mean firing rates of six single LIP cells recorded during each task. Colors correspond to the stimulus direction and category. Firing rates aligned to the sample stimulus onset are averaged by sample direction (left), and the test stimulus aligned rates are averaged over test direction (right). Although motion categories were not part of the discrimination task, the directions are labeled blue or red for consistency. (Inset polar plots) The mean firing rate for each sample direction during the sample stimulus presentation (circles and solid lines; 0–650 ms after motion onset) and delay period (triangles and dotted lines; 800–1450 ms after motion onset). The solid black line denotes 20 sp s$^{-1}$. **B** Mean category and direction tuning measured in single cells during the sample (open symbols) and delay periods (filled symbols) in each population (number of cells per population: B $n = 31$, J $n = 29$, D-disc. $n = 81$, D-early $n = 63$, D-late $n = 137$, H-disc. $n = 89$, H-early $n = 106$, H-late $n = 114$). The receiver operating characteristic (ROC) based category tuning index (rCTI[45]) measures category-specific responses in a range of −0.5 to 0.5, where positive values indicate category-selective neurons while negative values indicate more selectivity for within-category directions. Similarly, positive values of the ROC-based direction selectivity index (rDSI[13]) indicate direction selectivity. Error bars denote a 95% interval over bootstraps. Individual cell results are shown in Supplementary Fig. 1.

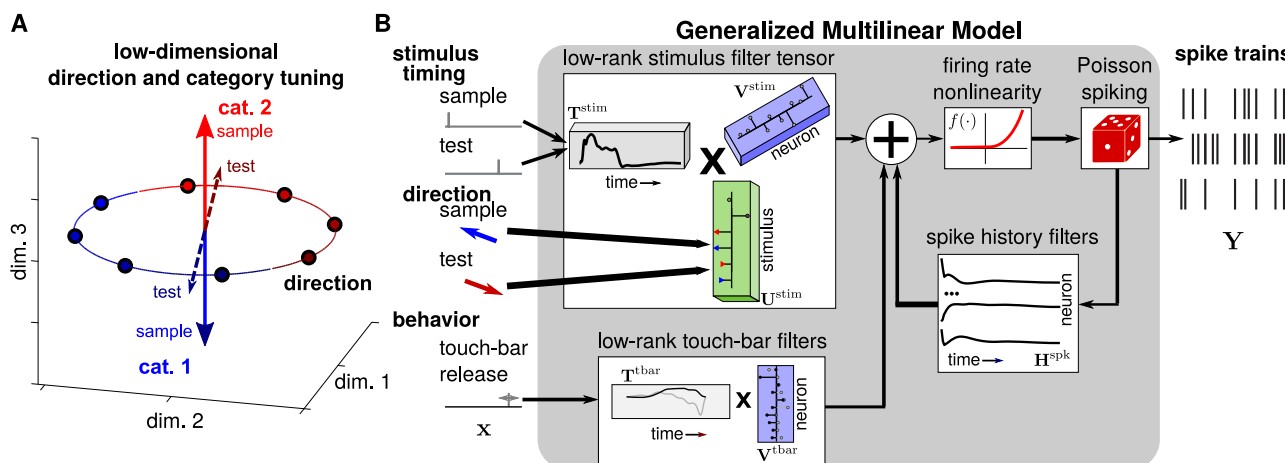

**Fig. 3 | The generalized multilinear model for dimensionality reduction of neural populations. A** We hypothesized that the different stimulus directions and categories—including whether it is the sample or test stimulus—could be modeled as vectors as a function of time in a low-dimensional space where the dimensionality is the number of factors. Here, we constrain the direction tuning, but not category, to be the same over the sample and test stimuli. Thus, dimensionality reduction in our framework can take into account shared temporal dynamics and stimulus tuning information across the two stimulus presentations, possibly reflecting similar bottom-up input. Individual neurons' stimulus tuning is a linear projection of the low-dimensional space. **B** Diagram of the GMLM. Incoming stimuli are factorized into temporal events and stimulus weights that encode direction and category information (left; Supplementary Fig. 3E–I). A set of temporal kernels and stimulus coefficients filter the linearized stimuli (left) into a low-dimensional stimulus response space. The touch-bar release event is similarly filtered using a low-dimensional set of temporal kernels. Each individual neuron's firing rate at each time bin is a nonlinear function (here, $f(\cdot) = \exp(\cdot)$) of the sum of a linear weighting of the low-dimensional stimulus subspace, a linear weighting of the touch-bar subspace, and recent spike history. Spike trains (right) are modeled as a Poisson process given the instantaneous rate. Given the recorded spike trains, stimuli, and behavioral responses, the stimulus filter tensor, touch-bar filters, and spike history filters can be fit to the data.

filter) and a set of weights for the stimulus identity. Each component temporally filters the incoming stimulus onset events, and weights the filtered stimuli linearly by stimulus identity. As a result, each individual component contributes to the population encoding for all stimuli (not just a single motion direction or sample/test presentation). Each individual neuron's tuning to the motion stimuli is a linear combination of the stimulus components. The model also includes a low-dimensional set of components to represent the touch-bar release event: a set of temporal kernels describe the population response to a touch-bar release such that each neuron's touch-bar tuning is a linear

combination of those kernels. Each spike train is then defined as a Poisson process in which the instantaneous firing rate is given by the sum of the filtered stimuli and touch-bar release, plus a linear function of recent spike history. In this formulation, the set of stimulus components is a tensor that represents each neuron's responses over time to each motion stimulus. The factorized representation of the stimulus into temporal and identity weights captures shared temporal dynamics between different directions or between sample and test stimuli. As the number of components (i.e., rank) in the kernel tensor increases, the model approaches a GLM fit to each cell individually (that is, each

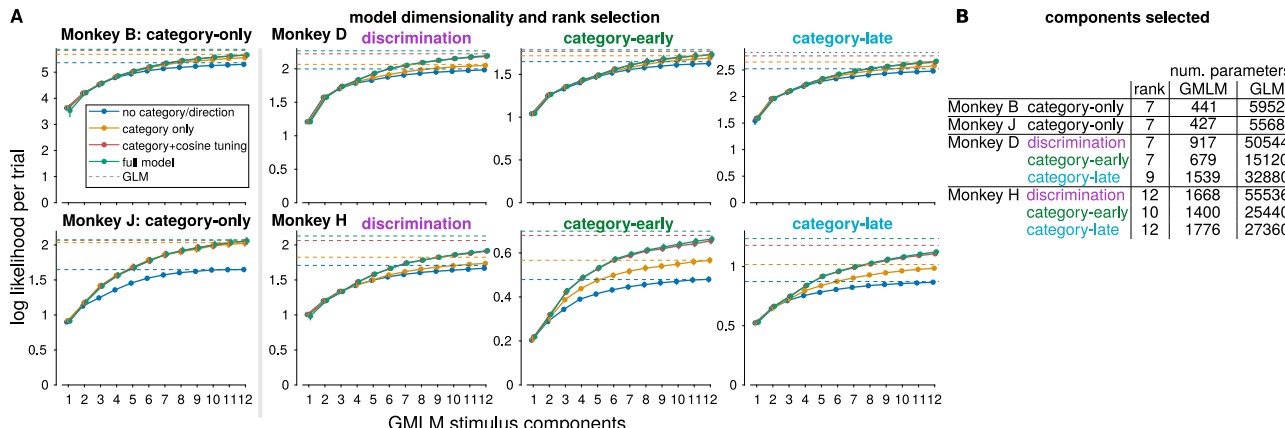

**Fig. 4 | A low-rank model captures task-relevant LIP activity during the DMS and DMC tasks. A** We determined the dimensionality of the population activity by estimating model fitness while increasing the number of components. Each point shows the mean cross-validated log-likelihood per trial of the GMLM relative to a baseline model without any stimulus terms (i.e., the "rank 0" model) for two monkeys during the category task. The log-likelihood shows the average log probability of observing the spike train from a withheld trial from one neuron given the model relative to the log probability of that observation under the null model without stimulus-dependent terms. In a Gaussian model, the log-likelihood is proportional to the squared error and it is related to the variance explained. Here, we instead use the Poisson likelihood which is more appropriate for quantifying spike count observations[37,80]. The traces show the GMLM with different amounts of category or direction information included. The dashed lines show the corresponding GLM fits (the full-rank model). The GMLM accounts for most of the log-likelihood with a small number of components. **B** Number of stimulus components (rank) selected for the GMLM for each LIP population to account for 90% of the log-likelihood. The number of stimulus parameters in the low-rank GMLM is compared to total parameters in the equivalent single-cell GLM fits for each population.

fiber of the tensor along the temporal mode is a GLM filter for one neuron for one stimulus).

We confirmed that the stimulus-related activity in the LIP population responses was low-dimensional. We varied the number of components to include in the GMLM (i.e., the rank of the stimulus kernel tensor). We compared the GMLM to the corresponding single-cell GLM fits, where the GLMs represent the "full-rank" model. We computed the average likelihood per trial averaged over the neurons in each population for the GMLM fit, relative to the GMLM without any stimulus terms (Fig. 4A). We selected the rank by the number of components needed to explain, on average, 90% of the explainable log-likelihood per trial over all the neurons in each population (Supplementary Fig. 4A). The GMLM required 7–12 stimulus components per population, thereby using only a fraction—less than 8%—of the number of parameters compared to GLM fits to individual cells (Fig. 4B). An example of the GMLM with seven stimulus tensor components fit the LIP population from monkey B is shown in Supplementary Fig. 5A, B. Different components can have different temporal response dynamics and different stimulus tuning properties: for example, component 5 shows strong differentiation between the two stimulus categories (red and blue), while component 2 does not. The responses to the stimulus for individual cells as a combination of components are illustrated in Supplementary Fig. 5C, D.

To gain intuition about how LIP dynamics may support the transformation of motion direction input into a representation of category, we visualized the low-dimensional population tuning to the sample stimuli. The top three dimensions of the trajectories show large, stimulus-independent transient responses (Fig. 5, inset). We therefore subtracted the mean response over stimuli and plotted the top three dimensions in the mean-removed responses[39]. The two category-only LIP populations showed primarily two-dimensional responses with strong category separation, with some direction tuning in the smaller third dimension in monkey J (Fig. 5). For the pretrained monkeys, the trajectories during the discrimination task reflected the stimulus geometry: the model shows two-dimensional transient activity organized by stimulus motion direction. The responses show little stimulus-specific persistent activity during the delay period[13]: the trajectories return to the origin after stimulus offset. During the category-early phase, the low-dimensional LIP response

reflects the stimulus geometry, but the top dimension is aligned to the task axis (i.e., blue and red directions are separated along dimension 1). The trajectories are still two-dimensional without clear delay period encoding. After training was completed, monkey D's category-late LIP activity showed strong direction tuning during the stimulus presentation which is elongated along the task axis (that is, the axis most oriented to the category along the 135°–315° stimulus directions). In contrast, LIP in monkey H had a three-dimensional stimulus response in the late period: two dimensions reflecting the circular motion directions during the stimulus presentation and a third orthogonal axis for category that was sustained through the delay period. Similar orthogonal stimulus input and working memory representations have been observed in other decision-making tasks[36,40]. In summary, the low-dimensional stimulus components of the LIP activity differed across animals such that the pretrained monkeys' LIP showed strong, circular representations of motion direction reflecting the physical features of the stimuli, while the category-only monkeys had lower-dimensional responses that more strongly reflected the task-relevant categories.

## Comparison to demixed principal components analysis

We next compared our approach to demixed principal components analysis (dPCA), an extension of principal components analysis (PCA) for recovering task-relevant subspaces in neural population responses[41]. We used dPCA to find components that captured the population encoding of the sample stimulus direction during sample and delay periods of the trial. Stimulus-dependent activity was primarily contained within the first few components, confirming the low dimensionality of stimulus-dependent activity (Fig. 6A). The LIP activity projected into the stimulus-dependent dPCs shows similar patterns to the GMLM subspaces (Fig. 6B). In addition, we found similar subspaces by performing PCA on the GLM filters that were fit to each neuron individually (Supplementary Fig. 10).

There are several limitations common among dimensionality reduction methods, such as PCA, dPCA, or tensor components analysis[42,43], for assessing the specific questions of interest in the current study. For example, bottom-up sensory-driven signals might be shared across the sample and test stimuli during the categorization task, while abstract category computations might differ across

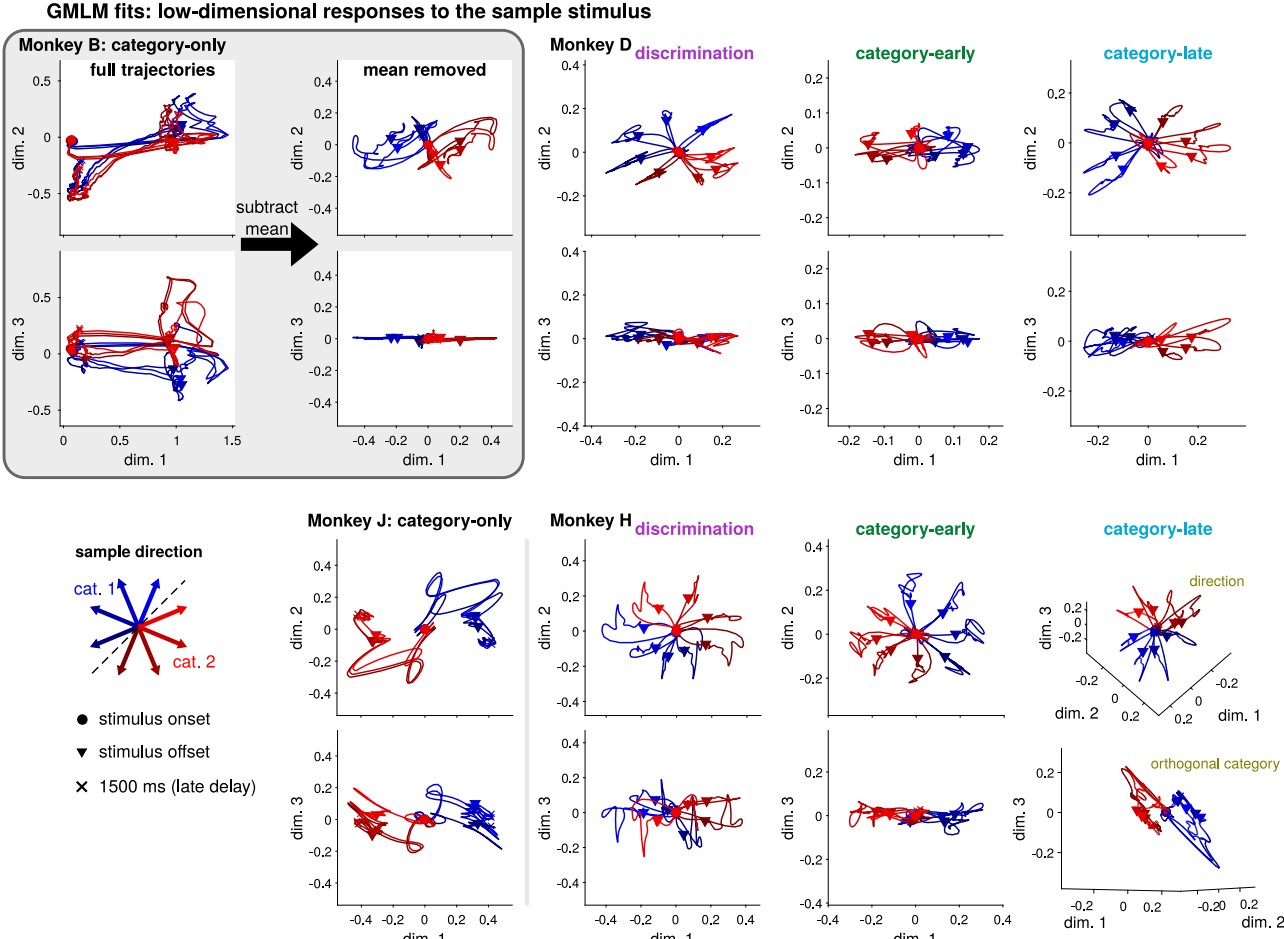

**GMLM fits: low-dimensional responses to the sample stimulus**

**Fig. 5 | Low-dimensional representations of motion direction and category during the sample stimulus presentation and delay period.** The top three dimensions of the GMLM's sample stimulus encoding for each of the eight LIP populations with the mean response over all directions removed through the first 1500 ms of each trial were computed by taking the higher-order singular value decomposition of the stimulus kernel tensor in the GMLM to get the dimensions that captured the most variance in the tensor. The inset for monkey B shows the top three dimensions including the mean (left) and the top three dimensions that remain after removing the mean (right). The two plots for each monkey show the top dimension on the *x*-axis plotted against the second or third dimensions on the *y*-axis (except for monkey H shown in the 3D plots). The red and blue traces show the response to each motion direction from stimulus onset (circles), to stimulus offset (triangles), and into the delay period (xs denote 1500 ms after sample motion onset). Supplementary Fig. 7 shows the full trajectories (i.e., without removing the mean) for all populations. Supplementary Fig. 8 shows the three-dimensional trajectories (of the cosine-tuned model) as a function of time.

presentations. Therefore, comparing data across both stimulus presentations may help to disentangle sensory-driven activity from higher-level responses. However, there are several reasons it is difficult to perform dimensionality reduction to answer these questions with typical methods like dPCA. First, the monkey's behavioral response is variable and determines the end-time of each trial. Dimensionality reduction approaches based on peristimulus time histograms (PSTH) including dPCA require temporally aligned trials of equal length, thereby limiting those approaches' ability to quantify task-related responses through the complete trial[36]. Second, including both sample and test stimuli on each trial also results in many distinct conditions which may be impractical for PSTH-based methods (for eight motion directions in the categorization task, dPCA would require computing the mean firing rate independently over 64 stimulus combinations). Third, methods such as PCA and dPCA do not directly provide quantification about how the stimulus direction and category are represented in the low-dimensional subspace: additional statistical modeling is required after dimensionality reduction, rather than performing the dimensionality reduction with the statistical questions in mind. Finally, structure in the spike trains beyond mean rate would be missed by PCA or dPCA (e.g., oscillatory dynamics related to working memory[44]). Therefore, our method could recover task-dependent, low-

dimensional activity similar to existing approaches, while also having the advantage of directly fitting the subspace to the data within a flexible statistical framework.

## Contribution of direction and category to LIP population responses

We asked how each task variable contributed to the low-dimensional LIP responses. To do so, we quantified model fit as we monotonically increased the complexity within a set of nested models by including more information about the stimulus identity and category (Fig. 4A). We designed a set of four nested models (i.e., different linearizations of the motion stimuli with increasing complexity) in order to assess what stimulus information is encoded by an LIP population (Supplementary Fig. 3E–I). The simplest model was the no category or direction tuning model. In this model, the linear weights for the stimulus identity only defined whether the stimulus was the sample or test. This model can only capture the average trajectories in time during the task over all stimulus directions. The second model, the category-only model, includes stimulus category weights, but does not consider specific motion directions. The category-only model includes stimulus identity information for the category one and category two motion directions for both the sample and test stimuli. This way, the model can capture

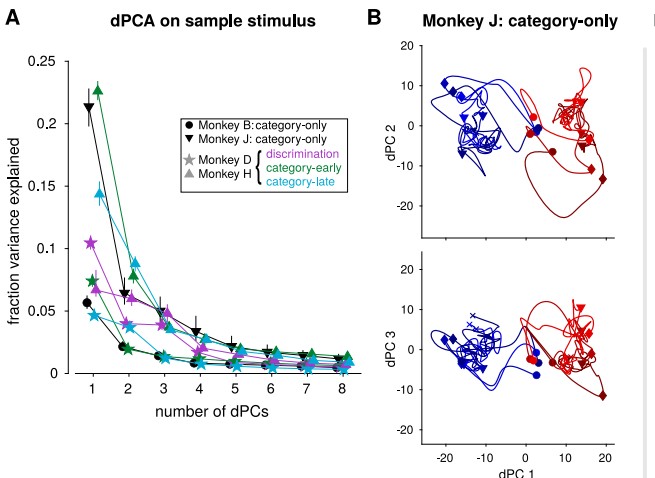

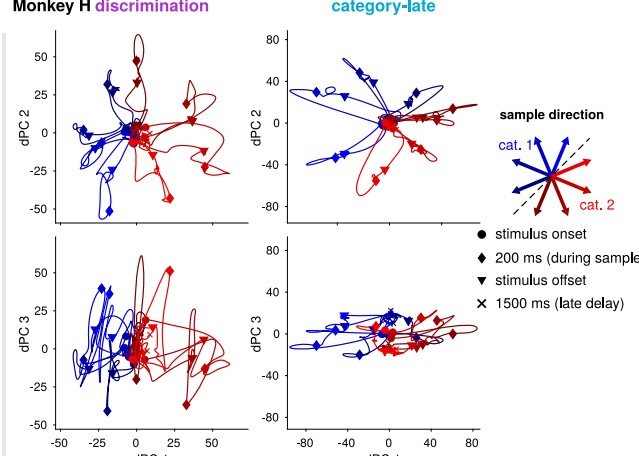

**Fig. 6 | Low-dimensional responses to the sample stimulus recovered by dPCA reflect stronger motion-direction tuning in pretrained animals than in category-only animals. A** The percent of the sample-direction-related activity explained by each stimulus dPC. Most of the variance is explained by the first two or three dPCs for all eight populations. Each trace shows the results for a single LIP population. The points denote the median and error bars show a 95% interval over bootstraps. **B** The low-dimensional activity from two LIP populations during the categorization task projected into the first three dPCs for motion direction. Each trace shows the population response to one sample stimulus direction (denoted by color) from sample onset to 1500 ms after stimulus onset (near the end of the delay

period). The markers indicate stimulus onset and offset. In this case, the motion direction components from dPCA simply correspond to PCA performed on the data with the mean response of each neuron across all motion directions subtracted. Because the category is determined completely by motion direction, dPCA cannot find separate "category" and "direction" components. The left column shows the three dPCs for category-only monkey J. The middle and right columns show the dPCs for pretrained-monkey H during the discrimination task and after training on the categorization task, respectively. The dPCs for all populations are shown in Supplementary Fig. 9.

category tuning, which may be different for the two stimulus presentations. While the discrimination task had no category, we still fit category weights to the discrimination populations as a control (i.e., to ask what the model produces if category was not actually a behavioral factor in the task because category and direction are correlated variables). The third model included cosine direction tuning and category. In addition to the category weights from the previous model, this GMLM included two coefficients for the sine and cosine of the motion direction. The cosine and sine weights were the same for both sample and test stimuli. Thus, this model constrains the geometry of direction tuning to lie on an ellipse in the low-dimensional space (Fig. 3B). The final model, the full model, extends the cosine direction tuning model by allowing different weights for each individual motion direction, rather than constraining the direction information to be cosine tuned.

A majority of the log-likelihood was accounted for by the GMLM without category or direction tuning in all populations, which is consistent with many previous dimensionality reduction results[41]. Including category improved model fit for all populations. We note that category is correlated with direction: adding a category variable can capture aspects of motion tuning. For example, category could capture much of the stimulus tuning in the monkey D category-late population, which shows a strong red-blue direction preference along the first dimension in the low-dimensional space (Fig. 5). As a result, the improvement of model fit does not imply that the populations encoded category-specific information. However, the category-only model still improved the fit to the category-late populations more strongly than the discrimination populations.

Adding cosine direction tuning during the categorization task accounted for a larger improvement of the model fit over the category-only model for the pretrained monkeys than the category-only monkeys (monkey B, category-only $1.0 \pm 0.4\%$; monkey J, category-only $0.2 \pm 0.7\%$; monkey D, discrimination $4.1 \pm 0.3\%$, category-early $1.6 \pm 1.0\%$, category-late $1.9 \pm 0.1\%$; monkey H, discrimination $8.2 \pm 0.6\%$, category-early $11.9 \pm 0.5\%$, category-late $9.9 \pm 0.6\%$; mean percentage cross-validated log-likelihood accounted for by the cosine-tuned GMLM minus the category-only GMLM). Adding the unconstrained full-direction tuning showed a similar improvement over the

category-only model (monkey B, category-only $1.2 \pm 0.3\%$; monkey J, category-only $0.5 \pm 0.8\%$; monkey D, discrimination $4.3 \pm 0.3\%$, category-early $1.8 \pm 1.1\%$, category-late $2.1 \pm 0.2\%$; monkey H, discrimination $8.5 \pm 0.6\%$, category-early $13.1 \pm 0.7\%$, category-late $11.2 \pm 0.6\%$.) This can be seen in the log-likelihoods in Fig. 4A where the red and yellow traces overlap for monkey J (left) while the red and green traces are significantly higher for monkey H (right). Thus, direction-tuning played a stronger role in the pretrained monkeys' LIP activity than in the category-only monkeys.

We tested the adequacy of cosine parameterization of direction tuning by comparing with the more flexible full model. The cosine direction tuning model was comparable to the full model for all populations (monkey B, category-only $0.2 \pm 0.2\%$; monkey J, category-only $0.3 \pm 0.7\%$; monkey D, discrimination $0.2 \pm 0.2\%$, category-early $0.2 \pm 0.1\%$, category-late $0.2 \pm 0.2\%$; monkey H, discrimination $0.3 \pm 0.4\%$, category-early $1.1 \pm 0.5\%$, category-late $1.3 \pm 0.3\%$; mean percentage cross-validated log-likelihood accounted for by the full GMLM minus the cosine-tuned GMLM). For these tasks, the direction tuning in the population could therefore be approximated as an ellipse (and thus embedded within a plane).

The parameterizations above assumed that the direction tuning (but not category tuning) is the same for both sample and test stimuli: that is, the difference between the kernels for two motion directions within the same category is the same for both the sample and test stimulus. Such direction-tuning constancy would be consistent with common bottom-up, direction-tuned input from sensory areas such as MT for the two stimulus presentations. The model still includes test category filters, which allow for different category tuning or direction-independent gain differences between sample and test stimuli. We found that including separate sample-test direction tuning in either model did not improve the model fit (Supplementary Fig. 11A). In addition, comparing sample and test direction weighting in the low-dimensional GMLM component space showed similar direction preferences for the two stimuli (Supplementary Fig. 11B). Thus, the GMLM framework can constrain the parameters to enforce constant direction tuning between the two stimuli, and test statistically whether that assumption holds. In our datasets, these tests showed that direction

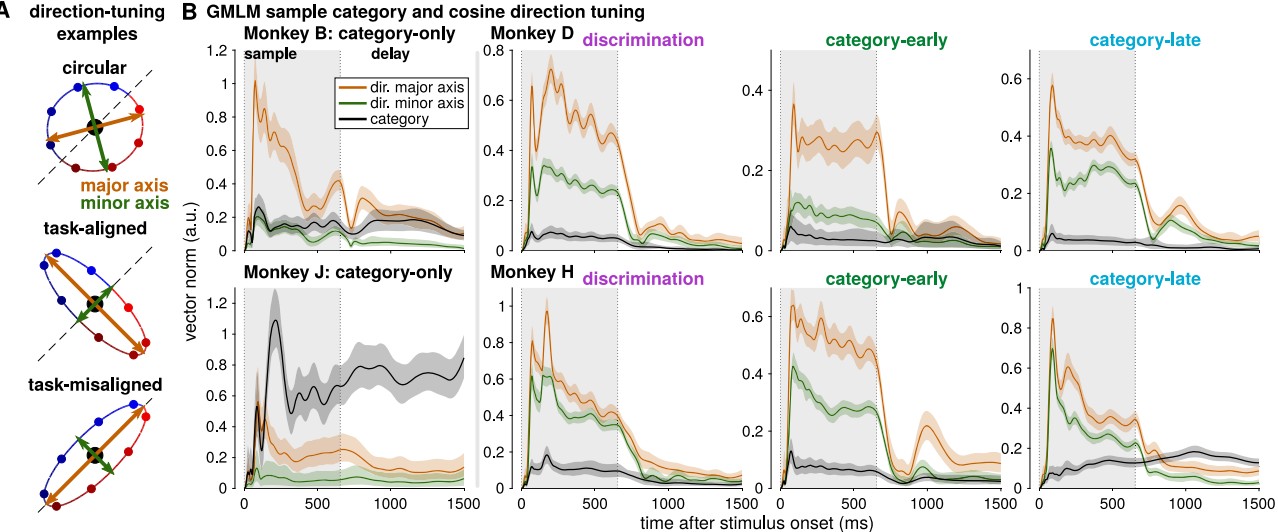

**Fig. 7 | Quantification of category and direction encoding in LIP. A** Diagram of direction encoding in the cosine-tuned GMLM. Motion stimulus direction is encoded as an ellipse in the low-dimensional stimulus space. The ellipse has major (orange arrows) and minor (green arrows) axes that define its shape. If the axes are of similar length, tuning is approximately circular and the motion directions are evenly distributed in the low-dimensional space (top). If the major axis is elongate compared to the minor axis, the population shows a preferred direction which may be aligned with category (middle) or the null direction (bottom) or anywhere in between. Category is encoded as a vector in addition to the direction ellipse, and the category vector is constant for all motion directions within a category (in contrast to the task-aligned direction tuning which places near-boundary directions closer together). **B** Bayesian estimate of the geometry of the sample stimulus tuning for the eight LIP populations. Each plot shows the norm of the sample category vector (black) and the norms of the major (orange) and minor (green) axes of the direction-tuning ellipse for an LIP population as a function of time relative to the sample stimulus onset. The solid lines denote the posterior median at each time point, and the shaded regions denote 99% credible intervals. If the major and minor axes have equal norms, then direction would follow a circle in the low-dimensional space.

encoding was cosine-tuned and constant across the sample and test stimuli, consistent with bottom-up input. This provides a parameterization of both category and direction by leveraging the two stimulus presentations, which have different cognitive demands, to support the approximate disentangling of the two variables.

## Quantifying the geometry of category and direction in the stimulus subspace

To go beyond visualization of the low-dimensional subspace, we wished to quantitatively assess the geometry of category- and direction-dependent responses in LIP. We therefore focused on the cosine-tuned GMLM, because this choice of parameterization decoupled direction and category, while still capturing a similar subspace to the full model. At each point in time, category was encoded along a vector while direction (parameterized by angle) was encoded on an ellipse in the stimulus subspace (Fig. 7A). The ellipse could be circular, which would represent motion directions uniformly, or elongated so that the population representations are biased toward a preferred motion direction. We compared the norm of major and minor axes of the direction ellipse and the norm of the sample category vector as a function of time relative to stimulus onset (Fig. 7B). Because category and direction are correlated, we applied a Bayesian analysis to take into account uncertainty in the direction and category encoding by exploring the tuning over the posterior distribution of the model parameters given the data, rather than only the best fitting parameters (see Methods section "Bayesian analysis of subspace geometry").

The category-only LIP population subspaces showed strong category tuning relative to direction tuning. The category vector in monkey B was of similar magnitude to the minor axis of the direction ellipse during stimulus presentation, and stronger during the delay period. Monkey B's direction-tuned response decayed during the sample period, consistent with the transient direction response in the example cell shown in Fig. 2A (top right). The category vector in monkey J was larger than the direction-tuning ellipse throughout the

stimulus presentation and delay period. The direction-tuning ellipses were elongated along a particular motion direction, rather than circular. In addition, the direction ellipse aligned both with category and with the choice biases in monkeys B and J on trials where the sample motion direction was ambiguous (Supplementary Fig. 12). The sample stimuli on ambiguous trials were placed on the category boundary, and the monkeys were rewarded randomly. The ambiguous trials were not used to fit the GMLM. Thus, the stimulus components in the category-only populations reflected category-specific input selection.

The pretrained monkeys' LIP showed strong direction tuning during both the discrimination and categorization tasks, and task-relevant category tuning emerged after training on the categorization task. LIP reflected strong cosine-like direction tuning during the discrimination task, which in monkey H was nearly circular or uniform across the motion directions (i.e., the major and minor axes of the direction ellipse were of similar length). In the category-early sessions, the direction tuning was less circular than the discrimination and category-late populations, as if the stimulus space was squeezed toward a lower-dimensional space during learning. While a dimensionality shift would be consistent with learning to represent category (a one-dimensional variable) instead of the full space of directions, this effect did not persist after additional training. During the category-late sessions—but not during the discrimination task—monkey D's direction ellipse was aligned with the task category (i.e., the major axis was along the 135°–315° angles; Supplementary Fig. 12C). Monkey D's LIP activity showed similarly shaped elliptical tuning across discrimination and category-late recordings, but the activity realigned to place motion category along the major axis after training on the categorization task. The same task-aligned direction encoding during the categorization task was not observed in monkey H. In both the monkey D category-late and monkey H category-early populations, we found stimulus offset activity in the direction ellipse, but not in the category vector. As a result, individual neurons may appear to respond more strongly for a particular category early in the delay period, but the

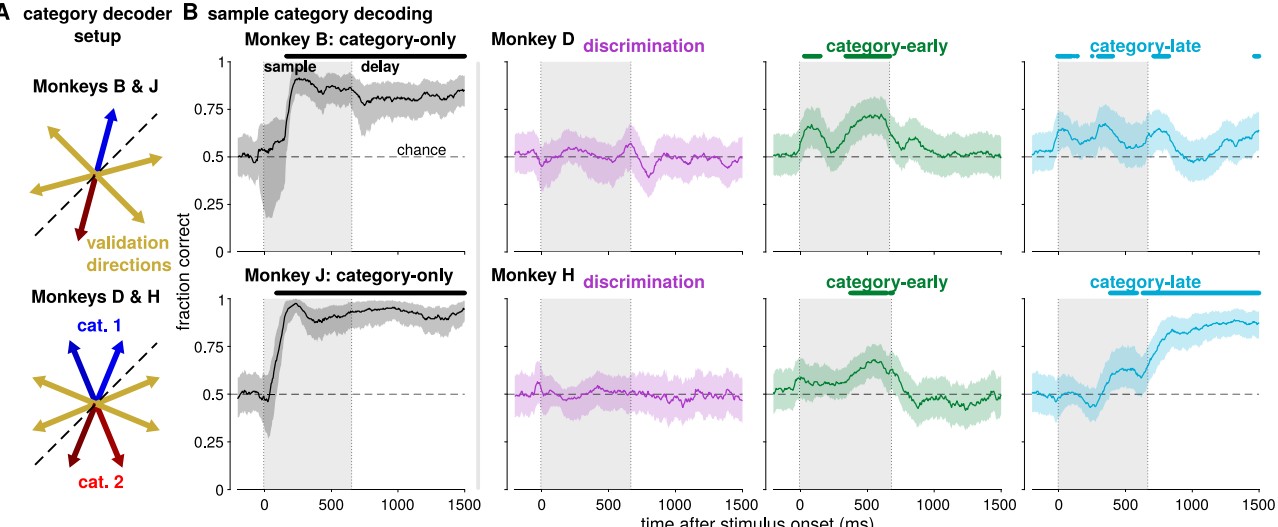

**Fig. 8 | Category and direction decoding in LIP. A** The training and test scheme for a direction-independent category decoder. Pseudopopulation trials were created from 50 random trials sampled with replacement for each direction from each cell. For monkeys B and J, the decoder is trained using one direction from each category (180° apart; red and blue) and validated on the remaining four directions (yellow). For monkeys D and H, the decoder is trained using two adjacent directions from each category (again using opposite directions from each category; red and blue) and validated on the remaining four directions (yellow). **B** Median decoder generalization performance as a function of time for each of the eight LIP populations. The decoder was trained and tested using spike counts in a 200-ms window centered at the time relative to sample stimulus onset on the *x*-axis. The shaded regions denote a 99% confidence interval over 1000 random pseudopopulations and the solid lines show the median. Symbols denote decoding significantly greater than chance (50%; *p* < 0.01 Benjamini–Hochberg corrected, one-sided bootstrap test). Supplementary Fig. 13 shows that we obtain consistent results with higher-rank models. Supplementary Fig. 12 shows that the stimulus tuning in monkeys B and J correlates with behavior on null-direction trials.

model accounted for this as a direction-tuned response rather than category-specific encoding (Fig. 7B). The monkey H-early and monkey D-early and category-late populations did not have large category vectors, and the low-dimensional activity instead reflects an elongated direction tuning ellipse (i.e., the major axis is larger than the minor axis) during stimulus presentation. In the monkey H category-late population, we observed a slow increase in the category vector length over time in the trial, which does not surpass the magnitude of direction tuning until the delay period. In summary, LIP in the pre-trained monkeys continued to show strong direction tuning after training on the categorization task, but the tuning either realigned with the motion categories (monkey D) or showed additional category tuning in addition to the direction-tuned response to the sample stimulus (monkey H).

We then asked how the subspace geometry could affect how decoding methods assess category selectivity in LIP (whereas the GMLM is an encoding model). We applied a linear decoding technique previously proposed to reveal category representations independent of motion direction[13,45]. The decoder classifies the sample category based on spike counts from pseudopopulation trials. We trained and validated the decoder on trials from orthogonal sets of stimulus directions (Fig. 8A). The logic of the decoder is that, if motion direction is represented in the population circularly without any additional category-specific responses, the decoder will not generalize across the training and validation conditions. The discrimination populations provided a control for the method, because the monkey was not yet trained to classify motion category. We indeed found no significant category decoding in the two discrimination populations (Fig. 8B).

The decoder performances during the categorization task were consistent with the GMLM stimulus subspaces. Category could be decoded with high accuracy in monkeys B and J early in the stimulus presentation and throughout the delay period (Fig. 8B). Similarly, the GMLM analysis found strong category tuning beginning early in the stimulus period and continuing through the delay in those populations. The results were different in the pretrained animals. In both

category-early and monkey D's category-late populations, we found decoding above chance during stimulus presentation, but not during the delay. The task-aligned, non-circular response to stimulus direction in monkey H category-early and monkey D category-early and late enabled the decoder to generalize across conditions due to over-representation of signal along the task dimension (135°–315°), rather than a category vector independent of direction. In contrast, in the category-late population for monkey H, the decoder only found weak decoding late in the sample stimulus presentation, which became strong during the delay period. The orthogonal category dimension of monkey H category-late is only stronger than the circular direction coding during the delay period, corresponding to the onset of significant category decoding. The decoder's failure to generalize during the early sample period can therefore be explained by strong direction selectivity swamping the weaker, orthogonal category signal.

## Comparing the sample and test stimuli

The categorization task requires different computations for the test and sample stimuli: the category of the sample stimulus must be computed and stored in short-term memory, while the test stimulus must be compared to the stored sample category. Recent work has suggested that LIP linearly integrates the test and sample stimuli during the test period of the categorization task while PFC shows more nonlinear match/non-match selectivity[46]. We therefore compared the LIP encoding of the test stimulus to the encoding of the sample stimulus. In the GMLM fits to the categorization populations, we found that test category encoding during the test stimulus presentation was weaker than sample category encoding during the sample presentation (Supplementary Fig. 14). This can be visualized in the low-dimensional subspace for monkey H (Fig. 9A). During the sample stimulus, the subspace reflected category tuning orthogonal to the motion direction response (Fig. 5, bottom right). However, we did not find the same category-selective response to the test stimulus in the stimulus subspace. In addition, LIP population activity projected onto the touch-bar (motor response) subspace showed strong match/non-

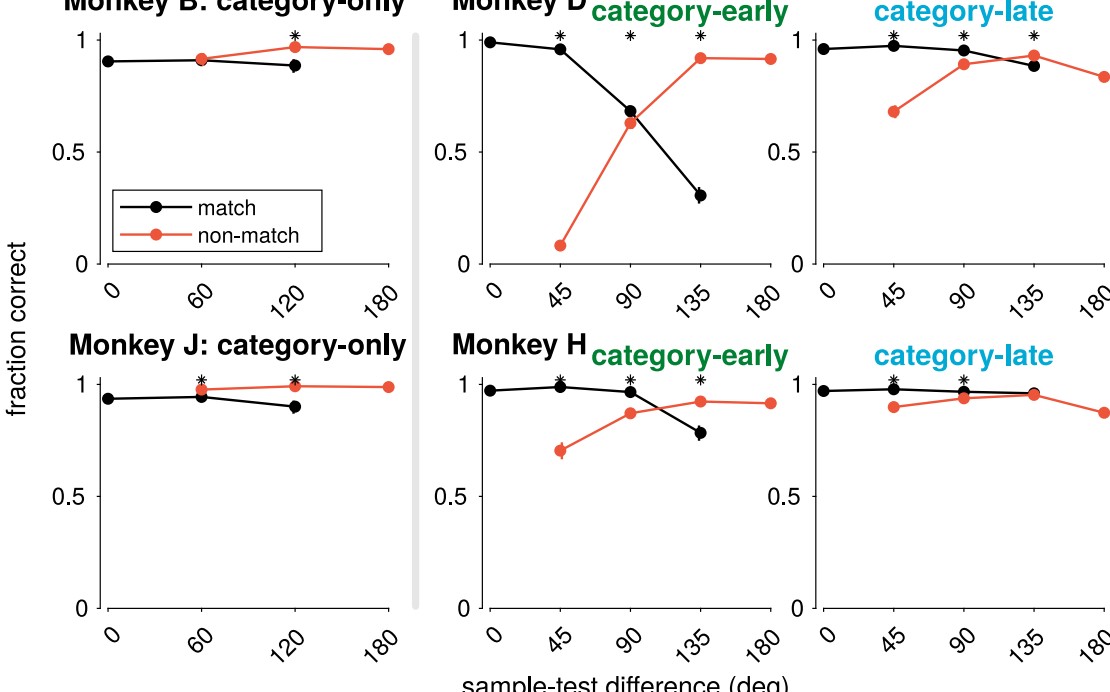

**A** low-dimensional test response

Monkey H: category-late

**B** decoding categories during test period

train: match (non-match) trials
test: non-match (match) trials

**C** behavioral performance: difference between sample and test directions

Monkey B: category-only

Monkey D category-early · category-late

Monkey J: category-only

Monkey H category-early · category-late

sample-test difference (deg)

match separation with little category selectivity (Supplementary Fig. 15). Thus, LIP does not appear simply to extract and sum the categories of the two stimuli to compute match or non-match.

We tested if the stored sample category and incoming test stimulus category were separable in the LIP population responses during the test stimulus presentation. We used linear classifiers to decode the sample category from pseudopopulation spike counts during the first

200 ms of the test stimulus presentation. The decoders were trained on trials of all stimulus directions. However, the training set consisted of only match (or non-match) trials, while the validation set included only non-match (or match trials). For the two category-only animals, monkeys B and J, the sample category decoder generalized across the two conditions (i.e., performed better than chance at 50%; Fig. 9B). We observed the opposite pattern in the pretrained monkeys: the sample

**Fig. 9 | Matching the test stimulus to the stored sample in the categorization task. A** The low-dimensional test stimulus-response for each direction for monkey H, category-late with the mean response removed projected into the same dimensions as in Fig. 5 (bottom right). **B** Decoding accuracy of sample or test category using the spike counts during the first 200 ms of the test stimulus (excluding the motor response for 95.7% of match trials). The decoder was trained on trials from all stimulus directions, but only from match (or non-match) trials and then tested on non-match (or match) trials. Performances for decoding sample and test category are mirror images (reflected over the 50% chance line): the training sets given to the binary classifiers are the same, but the test sets have opposite category labels. The box plots show the median and 25 and 75% range of decoder performance over bootstraps and the whiskers extend to a 1.5 interquartile range from the edges. All decoders generalized significantly different than chance (50%; $p < 0.01$ Benjamini–Hochberg corrected, two-sided bootstrap test with 1000 bootstraps of 50 trials per stimulus direction). **C** Average performance as a function of the difference in angle between the sample and test stimulus, sorted by match/non-match trials in the categorization task (error bars show a 99% credible interval). Asterisks indicate match and non-match are significantly different ($p < 0.01$, two-sided rank sum test, Benjamini–Hochberg corrected). Supplementary Fig. 14 compares the category tuning during the sample and test stimulus presentations. Supplementary Fig. 15 shows that the touch-bar subspace (encoding "match" responses) does not reflect category tuning.

decoder performed below chance. Decoding the test stimulus category instead produces a mirror image of these results (the training sets for the classifier are the same, but the test sets have opposite identities). In the category-only animals, sample category can therefore be read out by a single linear decoder regardless of the test stimulus identity, which is consistent with stronger separability of the remembered sample category and the incoming test stimulus in the category-only monkeys than in the pretrained monkeys. Increased separability suggests a coding scheme that reduces interference between the stored sample stimulus category and the specific test stimulus direction[40].

We then asked how the monkeys' performances depended on the similarity between the sample and test stimuli because this gives insight into the strategy the monkeys used to form their match versus non-match decisions. We compared the monkeys' accuracy as a function of distance between test and sample directions (Fig. 9C). The pretrained monkeys showed a different pattern of accuracy than the category-only monkeys. At small sample-test differences, the pretrained animals showed better performance on match than non-match trials while the category-only monkeys performed similarly or better on non-match. In addition, the pretrained animals showed greater dependence on distance. These effects were greatest during the category-early training phase, but they persisted after extensive training on the order of several months (the total number categorization task training sessions between the category-early and category-late periods was 78 for monkey D and 65 for monkey H). Therefore, stimulus similarity—which was relevant in the discrimination task—affects categorization behavior more strongly in the pretrained monkeys than the category-only monkeys, reflecting the monkeys' strategy.

## Single-trial dynamics during the delay period

Neural dynamics during single-trials may reflect aspects of sensory processing and working memory beyond the mean firing rate[44,47]. For example, working memory may be supported by persistent activity[48] or oscillatory bursts[49] while stimulus-related activity may exhibit strong transient responses with quenched variability[50]. We therefore sought to characterize non-Poisson variability in single trials in LIP during the discrimination and categorization tasks, which could reflect signatures of different strategies in performing the tasks. The GLM framework accounts for non-Poisson variability or single-trial dynamics by conditioning firing rates on recent spiking activity through a spike history filter, an autoregressive term that can reflect a combination of intrinsic factors (e.g., refractory periods, adaptation, facilitation) and network properties (e.g., oscillations)[51–53]. Typically, GLMs are designed assuming the spike history filter to be constant: that is, spike history has the same effect on spike rate throughout a trial. While fixed spike history effects may be an appropriate assumption in early sensory regions under stimulation with steady-state stimulus statistics, spiking dynamics in higher-order areas such as LIP might vary between the stimulus presentation and the delay period due to both the transition in behavioral task demands and different sources of input (bottom-up vs. top-down) between these two periods of the task[24]. In order to quantify spike history effects in the discrimination and

categorization tasks, we extended our model to allow the spike history filter to change over time during the trial (relative to the stimulus onset), which we have called dynamic spike history filters. However, estimating different spike history effects at each time point in the task for each neuron would be highly impractical (and even infeasible for single neurons). By instead using the low-dimensional GMLM to look for common changes in spike history effects that occur across the population, we added the dynamic spike history in the GMLM as a low-rank tensor with a linear spike history component and a gain term relative to the stimulus timing[54] (Fig. 10A). The spike history feature is a kernel shared by the entire population ($H^{dspk}$). We label the filter dynamic because the gain of the dynamic spike history's contribution ($T^{dspk}$) changes during the trial. The effect of spike history on a neuron's instantaneous firing rate therefore changes over the time course of the trial depending on the gain-modulated contribution of the dynamic spike history filter. Because the dynamic spike history filter and gain are shared, the model searches for the main features of spike history that change during the trial in the population. The resulting model can learn to capture distinct spike-history dependent dynamics between stimulus-driven and delay periods of the trial observed across many neurons.

We found the main spike history feature that varied during the task in each LIP population by fitting the GMLM with a single dynamic spike history component. Including dynamic spike history improved the cross-validated model performance for all populations (Supplementary Fig. 16A). The GMLM found similar dynamic spike history kernels for the two category-only monkeys, which showed oscillatory dynamics at approximately 12–14 Hz (low-beta; Fig. 10B). In contrast, the dynamic spike history kernel for the pretrained monkeys at all training stages was dominated by a faster timescale decay (time constants monkey D 10.2, 14.8 and 11.8 ms and monkey H 22.5, 9.0 and 6.6 ms discrimination, category-early, category-late, respectively), which suggests stronger gamma-frequency bursts. The stimulus-timing kernels showed that the dynamic spike history weighting is generally aligned with stimulus onset and offset (Fig. 10C). We note that the individual neurons' fixed spike history kernels act as the mean spike history, effectively centering the timing kernels around zero. The dynamic spike history is therefore reflected by the difference in weight between the sample and delay period, not the absolute value. This change in the dynamic spike history weight would be consistent with a change in network activity between the stimulus-driven and working-memory periods of the task. One notable exception was monkey J: the timing showed only a short transient gain after stimulus onset. The timing of this kernel corresponded with the strong category-dependent transient response in monkey J (Fig. 8B), and thus raises the possibility that the network enters a memory-storage state prior to stimulus offset.

Lastly, we summarized differences in the single trial dynamics across populations by comparing the mean spike history filters across all neurons in each population. Here, we examined population differences in the total spike history: the dynamic spike history filter (which depends on time in the trial) plus the individual neurons' fixed spike history filters. We computed the population mean spike history kernel

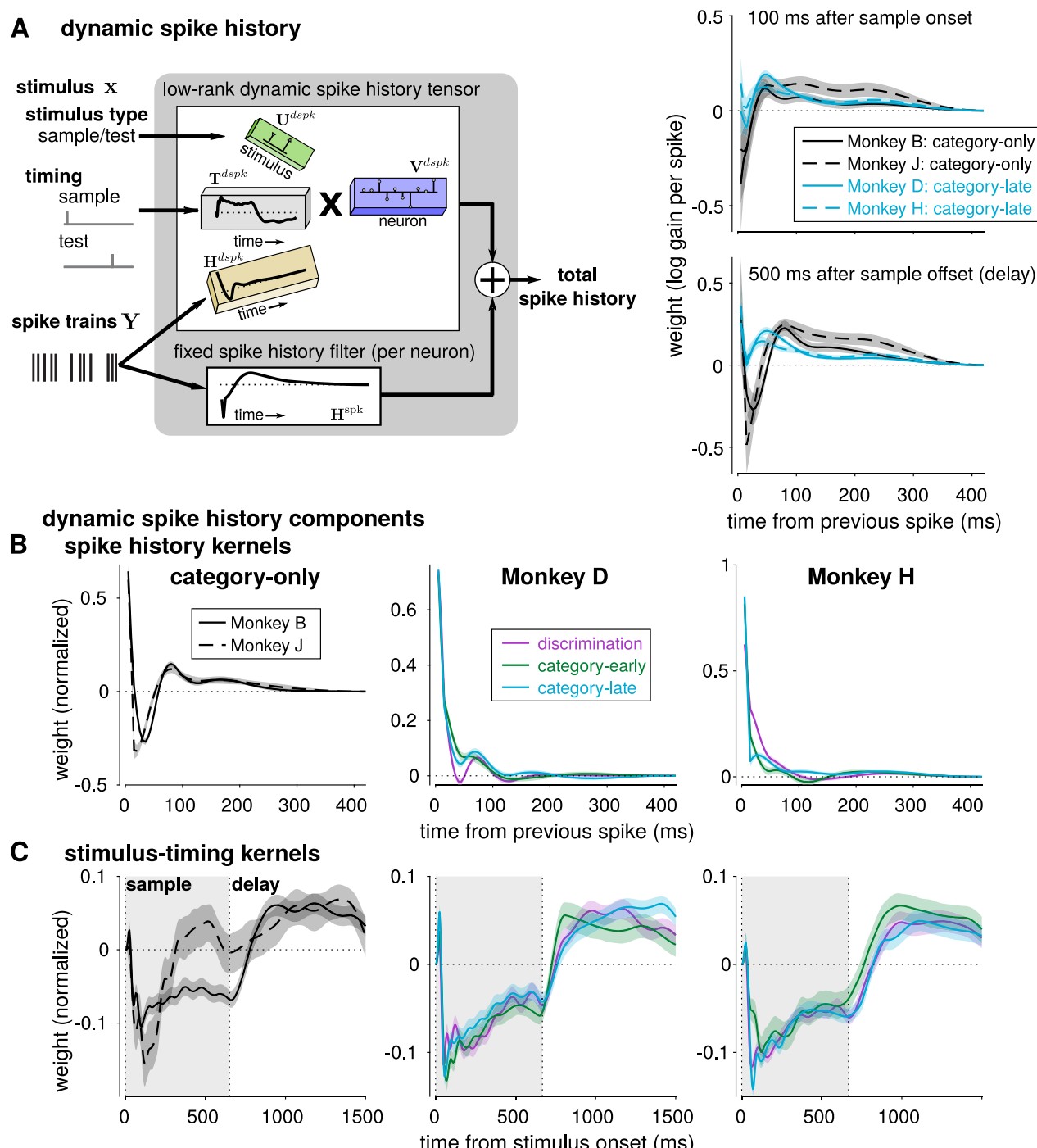

**Fig. 10 | Dynamic spike history captures distinct stimulus-driven and delay-period dynamics. A** The dynamic spike history filter is a low-rank, four-way tensor. The tensor includes two temporal kernels: one which filters spike history (gold) and a second which determines the weighting of the spike history component relative to the stimuli (gray). The spike history is scaled by stimulus identity (for simplicity, limited to sample or test stimulus weights only, without any category information). Each neuron adds the (weighted) dynamically filtered spike history to the neuron's constant spike history filter. The total spike history at any point in a trial is still a linear function of past spiking activity, but the effective linear kernel can change during a trial. The normalized rank-1 dynamic spike history components: **B** dynamic spike history kernels and **C** stimulus-timing kernels for each population (posterior

median and 99% credible interval). The left columns show the two category-only monkeys, and the middle and right columns show all training stages for monkeys D and H respectively. **D** The population mean effective spike history filters at two points in the task for the four fully trained categorization populations (mean of MAP estimate of filters ±2 SEM). (top) The population mean spike history at 100 ms after sample stimulus onset. (bottom) The population mean spike history during the delay period (500 ms after sample stimulus offset). Positive weights indicate that a previous spike at the given lag increases a neuron's probability of firing, while negative weights indicate that spiking is suppressed. Supplementary Fig. 16 quantifies the model fit improvement gained by including dynamic spike history and shows a higher-rank dynamic spike history fits.

at two different points (Fig. 10D): during stimulus-driven activity (100 ms after sample stimulus onset) and during the delay period (500 ms after sample offset). The mean spike history in the category-only monkeys showed a pronounced oscillatory-like trough during the delay period, compared to the pretrained monkeys (Fig. 10D, bottom). Spike history differences between the populations were less evident during stimulus presentation. These results were consistent for higher-rank dynamic spike history tensors (Supplementary Fig. 16A, B). We found similar differences can be observed in the spiking autocorrelation during the delay period (Supplementary Fig. 16C). However, the dynamic spike history in the GMLM did not require specifying a fixed window to compute the autocorrelation: the temporal gain term estimates when the spike history effects change in the trial and the stimulus terms help compute spike history effects in the presence of stimulus-driven changes in mean firing rate. Thus, the structure of single-trial variability during the delay period differed between category-only and pretrained monkeys, but was similar within each pair, which suggests that the balance of beta- and gamma-frequency-driven activity during the delay period differed between the animal pairs.

## Discussion

Here we examined the low-dimensional geometry of task-related responses during a motion-direction categorization task in LIP in two pairs of monkeys performing the same motion categorization task, but with different training histories. In the monkeys that were pretrained on a motion-direction discrimination task, we found similar direction-dependent activity in LIP activity during the discrimination and categorization task: two-dimensional direction-encoding subspaces that reflected the stimulus geometry. Moreover, uniform direction tuning remained a dominant feature of this subspace after training on the categorization task. The common direction tuning observed across the sample and test periods of the task could reflect cosine-like signals from sensory regions such as area MT[12,15,55]. In contrast, the monkeys trained only on the categorization task showed stronger category tuning and category-aligned direction tuning in LIP compared to activity in animals first trained on the discrimination task. Performing the categorization task may involve computations including input selection and local and/or top-down recurrent dynamics. Our findings indicate that differences in the sequences of tasks learned by the animals over long periods may result in different network configurations that perform the same task, perhaps manifesting in different behavioral strategies.

We hypothesize that these differences may be indicative of the pretrained monkeys still using computational strategies learned for the discrimination task. Indeed, the pretrained monkeys' behavior showed greater dependence on the angular difference between sample and test stimuli than category-only monkeys, which was a key factor in solving the discrimination task. Because the tasks used the same stimuli and shared many of the same correct or incorrect sample-test pairings, the same neural machinery and behavioral strategies could be recruited and maintained for the categorization task, despite extensive retraining. While many LIP neurons show delay period encoding of category during the categorization task, we did not see direction tuning in the average firing rates during the delay period in the discrimination task (Figs. 2A and 5). It is possible that direction is maintained in working memory in LIP populations during the delay period by sparse bursting activity, but not by persistent firing, which cannot be seen by our analysis using single-neuron recordings[44]. In addition, previous theoretical work from our lab has demonstrated that recurrent neural networks performing the discrimination task may recruit activity-silent computations to compare sample and test stimuli through short-term synaptic plasticity[56]. In that study using recurrent neural networks trained on both discrimination and categorization tasks, delay-period sustained activity was observed more

often in tasks that required more complex manipulation of the sample stimulus information compared to the discrimination task. Our dynamic spike history analysis revealed single-trial dynamics with low-beta frequency oscillatory structure during the delay period in the category-only monkeys, but not the pretrained monkeys, which could reflect different working memory dynamics in the category-only pair[44]. Together, this raises the possibility that computations learned during the discrimination task which recruited activity-silent working memory during the delay period could explain the observed reduced separability of the sample and test stimulus in the neural subspace in the pretrained monkeys compared to the category-only monkeys[57].

Studying how LIP populations encoded task variables in the discrimination and categorization tasks motivated our development of a statistical dimensionality reduction method to quantify the population code. We extended the GLM framework for individual neurons to perform dimensionality reduction on neural populations using a flexible tensor-regression model. In complex decision-making tasks, the trials may not be aligned such that common dimensionality reduction methods can be applied without artificially re-aligning single-trial firing rates by stretching or time-warping[41]. For example, the touch-bar release ended the trial early at a time determined by the animal in the discrimination and categorization tasks. This modeling framework could extend to many other tasks and questions, given appropriate linearizations of specific tasks. For instance, the tensor could be extended to model slow trial-to-trial changes in stimulus response within a recording session by including coefficients for weighting each trial, thereby generalizing applications of tensor component analysis as in[42,43]. Here, we applied the GMLM to perform dimensionality reduction to find task-relevant features in spike trains when the events in the task were not exactly aligned on every trial, without the need for an aligned trial structure. Our approach is related to reduced-rank regression[58,59] and the recently proposed model-based targeted dimensionality reduction[36] with two important distinctions: (1) our model is fit to spike trains through an autoregressive Poisson observation model and (2) we consider a more general tensor decomposition of task-related dynamics. The tensor decomposition is used to describe low-rank temporal dynamics in response to stimulus events, similar to low-rank receptive field models of early visual neurons[60–62], and those components are shared across all neurons in a population. In contrast to demixed principal components analysis[41] which requires balanced conditions across all variables to recover task-relevant subspaces, the cosine-tuned GMLM takes into account cosine-like direction tuning observed in sensory regions in order to disentangle category and direction information even though category is completely determined by direction. The GMLM takes into account cosine-like direction tuning observed in sensory regions in order to disentangle category and direction information even though direction and category are not separated in the task. Bayesian inference in this model allowed us to quantify uncertainty in the low-dimensional subspace and test hypotheses about the geometry of neural representations. The model showed that direction tuning was similar across the sample and test stimuli, while category-dependent responses differed, in order to support this disentangling of category and direction.

There are several important limitations about the inferred behavioral strategy and neural mechanisms in the present study. Primarily, this study included only a small number of animals, as is the norm in non-human primate experiments. Furthermore, multiple cortical and subcortical areas are involved in decision making (such as prefrontal cortex), and our analysis only considered neural activity in LIP. Even within a single region, it is possible that our results could depend on differences in neuron sampling within LIP between animals or sessions (e.g., either more anterior or more posterior), or other factors not directly related to the animals' training histories. Given the large variation in LIP activity observed between animals within each pair, we cannot measure the magnitude of training

history effects relative to variance due to individual differences. We cannot exclude the possibility that animals could switch behavioral strategy with additional training such that, for example, both pairs of monkeys would perform similarly. Consequently, the possibility remains that LIP representations of the categorization task could change to match the currently adopted behavioral strategy, rather than purely reflecting training history. We think this is unlikely because the LIP recordings were made after all animals had received extensive training on the categorization task and their behavioral accuracy had appeared to asymptote at a high level[13]. We note that, although the LIP recording sessions were performed using different sets of motion directions for the two pairs of monkeys (six directions for the category-only monkeys and eight for the pretrained pair), both pairs of monkeys were trained extensively on a larger set motion directions[13]. We focused on single-neuron recordings and the components of our model described common motifs in the mean responses to the task variables, while more aspects of the single-trial computations involved in categorization would require large population recordings[30,63]. Finally, there are many training and task variables that could impact learning beyond those considered here. For example, training on the categorization task first could influence direction coding in the discrimination task (learning the tasks in the opposite order that we considered) or using noisy motion stimuli (e.g., drifting motion direction) in the categorization task could affect the neural dynamics underlying categorization in LIP.

There are multiple ways that the brain could learn to perform the same task. Average results across animals may therefore fail to reflect the neural mechanisms of decision making in individual animals[64,65]. Individual differences are a major focus of human decision-making research and have led to many insights into cognitive functions including working memory[66,67]. Here, we explored between-subject differences in the dimensionality and the relationship between direction and category tuning in LIP, and we found differences that correlated with long-term training history. Primates in particular may participate in many experiments and receive extensive training in multiple closely related tasks over the course of years. Experimenters should report and consider animals' training histories when interpreting such data and when comparing seemingly conflicting results from different labs. In conclusion, the low-dimensional dynamics that posterior parietal cortex enlists to support abstract visual categorization can manifest differently across subjects, and exploring long-term effects of training over more subjects can provide broader perspectives of the diverse neural computations that give rise to decision-making skills.

## Methods

### Data
All datasets used for this study were the same as in two previous studies from our group (refs. [22] and [13]):

**Tasks.** The details of the tasks have been described previously for monkeys B and J category-only monkeys[22,45] and monkeys D and H pretrained monkeys[13]. For all animals, stimuli were high-contrast, 100% coherent random dot motion stimuli with a dot velocity 12°s$^{-1}$. The motion patch was 9.0° diameter, and the frame rate was 75 frames s$^{-1}$. Monkeys were required to keep fixation within 2° radius of the fixation point during each trial.

For monkeys D and H, there were eight sample stimulus directions for both the discrimination and categorization tasks, spaced 45° apart: {22.5°, 67.5°, 112.5°, 157.5°, 202.5°, 247.5°, 292.5° and 337.5°}. The test stimuli for the discrimination task were 45°, 60°, 75°, or 0° (match) away from the sample stimulus, giving a total of 24 possible motion directions in the task. Test stimuli for the categorization task were the same as the eight sample stimuli. The stimulus presentations were 667 ms and the delay period was 1013 ms.

The categorization task for monkeys B and J used directions spaced evenly in 60° intervals: {15°, 75°, 135°, 195°, 255° and 315°}. The stimulus presentations were 650 ms and the delay period was 1000 ms.

For all monkeys tested on the categorization task in this study, the motion directions were split evenly into two categories separated by a constant boundary at 45° and 225°.

The categorization task for monkeys B and J included a set of null-direction trials (Supplementary Fig. 12A). In these trials, the sample direction was along the category boundary (45° or 225°) and the test direction was either 135° or 315° (one direction from each category, furthest from the boundary). These trials were not used to fit neural models, but examined for behavior in Supplementary Fig. 12B. The monkey's response was randomly rewarded at 50% chance on these trials. We note that monkeys B and J were first trained on a simplified discrimination task where the sample and test stimuli were either match or 180° opposite. This version of discrimination task therefore did not require fine motion direction discrimination, and all correct sample-test response pairs in this task were consistent with the categorization task.

**Electrophysiology.** Neurons in LIP were recorded using single tungsten microelectrodes. During both the discrimination and categorization tasks, the motion stimuli were placed inside an LIP cell's response field, which were assessed using a memory-guided saccade task prior to running the main experiment on each session.

In this study, we included only cells with a mean firing rate of at least 2 sp s$^{-1}$, averaged from sample stimulus onset to test offset. We included $N = 31$ cells from 26 sessions for monkey B, and $N = 29$ from 27 sessions for monkey J. For monkey D, $N = 81$ cells from 39 sessions for the discrimination task, $N = 63$ cells from 33 sessions for the category-early period, and $N = 137$ cells from 59 sessions for the category-late period. For monkey H, $N = 89$ cells from 55 sessions for the discrimination task, $N = 106$ cells from 40 sessions for the category-early period, and $N = 114$ cells from 50 sessions for the category-late period.

**Data used for modeling.** For all the modeling and decoding analyses, we included only correct trials. The null-direction trials for categorization task for monkeys B and J were not included for model fitting.

We considered a time window in each trial starting from sample stimulus onset until 50 ms after the touch-bar release (if a touch-bar release occurred) or 50 ms after the test motion offset. We discretized the spike trains during each trial into 5 ms bins. We note that on non-match trials the animal was required to hold the touch bar until a second test stimulus (which is always a match) appeared. However, the second test stimulus presentation was never included in our analysis.

We used 10-fold cross-validation to compare the models. The trials from each cell were divided into folds evenly by sample directions. For example, if there were 40 trials recorded with a sample motion of 22.5° for one cell, these trials were divided into 10 groups of 4 trials to make the folds.

For plotting the PSTHs in Fig. 2A and Supplementary Figs. 5D and 15, we smoothed the average spike rate over trials conditioned on motion direction using a Gaussian kernel with a 30-ms width.

**Behavioral performance.** For quantifying behavioral performance, we only analyzed behavior during the LIP recording sessions. In the behavioral analyses in Fig. 9A and Supplementary Fig. 12B, we estimated the fraction correct (or fraction touch-bar releases) independently in each condition using a beta-binomial model. The prior parameters in the model were $\alpha = 1$ and $\beta = 1$ (for a beta distribution over the prior fraction correct or touch-bar released trials). In this model, the posterior over the fraction correct (or touch-bar released) is a beta distribution. The point estimate of the fraction correct was the posterior mean, and the error bars denote 99% credible intervals over the posterior.

## Single-neuron tuning curve analysis

We characterized individual neuron category selectivity using a receiver operating characteristic (ROC) analysis using the category tuning index (rCTI) and the direction selectivity index (rDSI). Full details of these metrics are given in refs. [45] and [13]. Briefly, rCTI computes this difference in discriminability (given as the area under the ROC curve) of spike counts observed between two directions. The weighted discriminability between category directions and within category directions is then summarized on a −0.5 to 0.5 scale. An rCTI of 0 indicates no difference in firing in response to motion directions within or across the category boundary, −0.5 indicates perfect separation of within-category directions (but not across-category directions), and 0.5 indicates strong across-category separation of motion directions (but not within-category). Similarly, rDSI is a shuffle-corrected measure of the maximum within-category difference between two directions, computed using the area under the ROC. An rDSI of 0.5 indicates strong direction selectivity, while values 0 (or negative values that result from the shuffle correction) indicate no direction selectivity. We computed these quantities using the spike counts across all trials within a 650-ms bin (0–650 ms after stimulus onset of the sample period fit, and 800–1450 ms for the delay period fit).

We also fit parametric tuning curves of each neuron using a Poisson GLM for the spike counts within in window. The tuning firing rate for a direction of angle $\angle$ where $c_\theta = 0.5$ if $\theta$ belongs to category 1 and −0.5 otherwise was given as:

$$\lambda(\theta) = \exp\left[\beta_0 + \beta_{\text{cat}} \cdot c_\theta + \beta_{\sin} \sin(\theta) + \beta_{\cos} \cos(\theta)\right] \quad (1)$$

where $\beta_0$ determines the mean firing rate, category tuning is given by $\beta_{\text{cat}}$, and the direction coefficients $\beta_{\sin}$ and $\beta_{\cos}$ determine the preferred direction and magnitude of direction tuning. The parameters were fit with maximum likelihood given the same spike counts used to fit rCTI and rDSI.

Category tuning magnitude was determined by $|\beta_{\text{cat}}|$ and direction tuning magnitude was the difference in response over the preferred and anti-preferred direction:

$$\text{direction magnitude} = \max_\theta \beta_{\sin} \sin(\theta) + \beta_{\cos} \cos(\theta) \\ - \beta_{\sin} \sin(\theta + 180°) - \beta_{\cos} \cos(\theta + 180°). \quad (2)$$

Because these values are strictly positive, they are positively biased. We therefore computed model fits on shuffled data (shuffling direction and category labels across trials) and subtracted the mean magnitude computed on the shuffled fits.

Error bars for all single-neuron analyses were determined by fitting 1000 bootstrapped samples. Bootstraps were obtained by resampling (with replacement) an equal number of trials within each sample direction for each cell that was obtained in the original dataset.

## Demixed principal components analysis

We conducted dPCA as described in ref. [41]. The analysis was performed on smoothed spike trains sampled in 5 ms bins from 0 to 1500 ms after sample stimulus onset smoothed with a Gaussian kernel of width 30 ms. The regularization parameter was selected using 5-fold cross-validation. The dPCA analysis included only two marginalizations (i.e., conditional PSTH): the stimulus-independent term (mean rate over all stimuli) and one for the motion directions (mean rate for each sample stimulus direction). In this analysis, motion direction is a categorical variable and cannot be separated from category as separate demixed subspaces. This is because dPCA assumes a fully balanced set of conditions for demixing (i.e., trials covering all combinations of directions and categories), but category and direction are perfectly correlated (i.e., no trials with stimulus category 2 and direction 60° exist in this task).

## GLM for single cells (the full-rank model)

In this section, we define the generalized linear point-process model of single cells during the discrimination and categorization tasks. This class of model for single neurons in decision-making tasks is defined in general in ref. [32]. The GLM defines the distribution of spike count at time $t$ as a Poisson random variable with mean rate given as a linear function of external events (here, stimulus and touch-bar release) and previous spiking activity:

$$\lambda(t) = w + \left(\mathbf{h}^{\text{spk}} * \mathbf{y}\right)(t) + \left(\mathbf{k}^{\text{tbar}} * x^{\text{tbar}}\right)(t) \\ + \sum_{s \in \mathcal{S}} \left(\mathbf{k}^{(s)} * x^{(s)}\right)(t) \quad \text{(log firing rate at time } t) \quad (3)$$

$$\mathbf{y}(t) \sim \text{Poisson}\left(f(\lambda(t))\Delta\right) \quad \text{(spike count for bin } t) \quad (4)$$

$$f(\cdot) = \exp(\cdot) \quad \text{(inverse link function)} \quad (5)$$

The * operator denotes convolution. The bin width is $\Delta$, and the log baseline firing rate parameter is $w$. Recent spiking activity, $\mathbf{y}$, affects the rate through the spike history kernel, $\mathbf{h}^{\text{spk}}$.

The stimulus event regressors $x^{(s)}(t)$ are functions of at time $t$ representing information about the motion stimulus events. The set of all stimulus events is $\mathcal{S}$. The touch-bar event is $x^{\text{tbar}}$. The linear temporal kernels $\mathbf{k}^{(s)}$ and $\mathbf{k}^{\text{tbar}}$ describe the cell's response to each external variable (stimulus or touch-bar, respectively) as a function of time. The stimulus events we consider encapsulate both sample and test stimuli, but the configuration and number of stimulus kernels depend on the specific model parameterization.

We parameterized the temporal kernels using raised cosine basis functions[68]. The stimulus kernel basis consisted of $P_{\text{spk}} = 24$ functions with a nonlinear stretching parameter of 0.2 and peaks spanning 0–1500 ms (Supplementary Fig. 3J, left). We aligned the basis so that the first basis was zero at exact time of stimulus onset, giving peaks between 40 and 1540 ms relative to stimulus onset. The touch-bar basis was constructed using the first $P_{\text{tbar}} = 8$ functions of the stimulus basis (Supplementary Fig. 3J, middle). The functions were reversed and shifted the basis so that the function peaks ranged from −235 to 25 ms relative to the touch-bar release and the fastest temporal resolution of the basis set was near the touch-bar release time. We used $P_{\text{stim}} = 24$ basis functions for the spike history (Supplementary Fig. 3J, right). The first two basis functions were Kronecker delta functions to account for the first two bins (0–5 and 6–10 ms after a spike). The remaining eight functions were a raised cosine basis set with nonlinear stretching parameter of 0.05 and peaks from 10 to 20 ms post spike.

We define the kernels as the bases times a set of coefficients:

$$\begin{aligned} \mathbf{k}^{(s)} &= \mathbf{B}^{\text{stim}} \bar{\mathbf{k}}^{(s)}, & \text{for } s \in \mathcal{S} \text{ where each } \bar{\mathbf{k}}^{(s)} \text{ is a vector of length } P_{\text{stim}} \\ \mathbf{k}^{\text{tbar}} &= \mathbf{B}^{\text{tbar}} \bar{\mathbf{k}}^{\text{tbar}}, & \text{where } \bar{\mathbf{k}}^{\text{tbar}} \text{ is a vector of length } P_{\text{tbar}} \\ \mathbf{h}^{\text{spk}} &= \mathbf{B}^{\text{spk}} \bar{\mathbf{h}}^{\text{spk}}, & \text{where } \bar{\mathbf{h}}^{\text{spk}} \text{ is a vector of length } P_{\text{spk}}. \end{aligned} \quad (6)$$

The parameters that are fit to data are $\phi = \{w, \tilde{\mathbf{h}}^{\text{spk}}, \tilde{\mathbf{k}}^{\text{tbar}}, \tilde{\mathbf{k}}^{(s)} | s \in \mathcal{S}\}$. This choice of basis ensures that the stimulus kernels are causal: the stimulus filters only contribute to firing rate after stimulus onset. In contrast, the touch-bar release is acausal: touch-bar release can contribute to the spike rate before the behavior to reflect buildup to the match decision.

We linearized the task events as point events in time. The touch-bar release is given as

$$x^{(\text{tbar})}(t) = \begin{cases} 1 & \text{if } t = t_{\text{tbar}} \\ 0 & \text{otherwise.} \end{cases} \quad (7)$$

where the time the monkey released the touch-bar to signal a match response is $t_{tbar}$ (if the touch-bar was release in the trial). We similarly consider the stimulus onsets (both sample and test) as point events. The sample stimulus duration is constant across all trials, and although the test stimulus is terminated early by a touch-bar release, this does not factor into the window of the trial we model. However, the model can extend to tasks with variable stimulus duration, as has been shown previously[32]. Each GLM kernel gives a scalar contribution to firing rate of the relative time of the task event.

We considered a set of four nested models of increasing complexity for the motion stimuli. For simplicity of notation, we present the linearization for a single trial. The sample stimulus onset time is $t_{sample\ on}$ and the test stimulus onset is $t_{test\ on}$. The sample and test stimulus directions are $\theta_{sample}$ and $\theta_{test}$ for sample, test $\in \{1, 2, ..., D\}$ where $D$ is the total number of stimulus directions in the task. The stimulus directions belong to categories denoted $c_{sample}, c_{test} \in \{1, 2\}$.

1. No category or direction model (Supplementary Fig. 3A). This model includes only two stimulus regressors/kernels: one for the sample stimulus onset and one for the test stimulus onset (Supplementary Fig. 3A): $\mathcal{S} = \{x^{(sample)}, x^{(test)}\}$. The regressors are defined as point events

$$x^{(sample)}(t) = \begin{cases} 1 & \text{if } t = t_{sample\ on} \ , \\ 0 & \text{otherwise.} \end{cases}$$
$$x^{(test)}(t) = \begin{cases} 1 & \text{if } t = t_{test\ on} \ , \\ 0 & \text{otherwise.} \end{cases} \quad (8)$$

This model captures temporal dynamics in the mean response for each neuron across all stimuli.

2. Category-only model (Supplementary Fig. 3B). This model includes four stimulus kernels: two for each category for the sample stimulus ($x^{(cs1)}$ and $x^{(cs2)}$), and two separate kernels for the test stimulus categories ($x^{(ct1)}$ and $x^{(ct2)}$). The regressors are again point events, but the points are now conditioned on stimulus category (but not specific direction). For each category $k \in \{1, 2\}$:

$$x^{(csk)}(t) = \begin{cases} 1 & \text{if } t = t_{sample\ on} \text{ and } c_{sample} = c_k \ , \\ 0 & \text{otherwise ,} \end{cases}$$
$$x^{(ctk)}(t) = \begin{cases} 1 & \text{if } t = t_{test\ on} \text{ and } c_{test} = c_k \\ 0 & \text{otherwise.} \end{cases} \quad (9)$$

Although the discrimination task does not include category, we still applied this model to those data as if there was a category boundary at 45° and 225.

3. Cosine direction tuning model (Supplementary Fig. 3C). The cosine-tuned model includes both stimulus category and direction tuning, but direction encoding is constrained to a parametric form with cosine tuning. The model includes six stimulus events: two for each category for the sample stimulus ($x^{(cs1)}$ and $x^{(cs2)}$), two for the test stimulus categories ($x^{(ct1)}$ and $x^{(ct2)}$), and two for the sine and cosine of the direction ($x^{(sin)}$ and $x^{(cos)}$). The sample and test category events are defined as in the previous model. The direction regressors are weighted point events, which are shared for both sample and test stimuli:

$$x^{(cos)}(t) = \begin{cases} \cos(\theta_{sample}) & \text{if } t = t_{sample\ on} \\ \cos(\theta_{test}) & \text{if } t = t_{test\ on} \\ 0 & \text{otherwise.} \end{cases}$$
$$x^{(sin)}(t) = \begin{cases} \sin(\theta_{sample}) & \text{if } t = t_{sample\ on} \\ \sin(\theta_{test}) & \text{if } t = t_{test\ on} \\ 0 & \text{otherwise.} \end{cases} \quad (10)$$

4. Full model (Supplementary Fig. 3D). The full model allows for general (non-cosine) direction tuning. However, we constrained

the model to have the same direction tuning for both sample and test stimuli; that is, the difference in tuning between two directions within the same category was the same for both sample and test stimuli. The model included one stimulus regressor event for each directions plus two test category events (for the categorization task with $D = 6$ stimulus directions, there are eight kernels) For each trial and direction $\theta_d$ for $d \in \{1, 2, ..., D\}$, the stimulus regressors are

$$x^{(\theta_d)}(t) = \begin{cases} 1 & \text{if } t = t_{sample\ on} \text{ and } \theta_{sample} = \theta_d \\ 1 & \text{if } t = t_{test\ on} \text{ and } \theta_{test} = \theta_d \\ 0 & \text{otherwise.} \end{cases} \quad (11)$$

The two stimulus events that parameterize the test category responses ($x^{(ct1)}$, and $x^{(ct2)}$) are defined as before. This parameterization maintains identifiability: it would not be identifiable to directly expand the cosine model to have a kernel for each direction plus two sample and two test category kernels. As a result of the identifiability constraint, the interpretation of the corresponding kernels is different compared to the cosine tuning model: in this model, $\mathbf{k}^{(\theta_d)}$ is the kernel for a stimulus in the $\theta_d$ direction plus the response to a stimulus of the category of $c_d$. Therefore, we view $\mathbf{k}^{(ct1)}$ as the kernel to test stimuli of category 1 minus the kernel for a category 1 sample stimulus (thereby subtracting the sample category tuning away from the direction kernel and adding the test category tuning).

We considered two additional models included in supplementary analyses that included independent sample and test direction tuning (Supplementary Fig. 12). The independent direction cosine tuning model had eight kernels total: four for the sample and test category, four for the cosine and sine weights of the sample and test directions. The full independent direction model simply had one kernel for each sample direction and one kernel for each test direction These two models are defined analogously to the common direction tuning models.

**Prior probabilities over model parameters**. We defined zero-mean Gaussian priors over the stimulus kernels. The orthonormal basis functions controlled temporal smoothness of the kernels, and the prior distributions were independent over time. We defined i.i.d. priors for the $i$th coefficients of the stimulus kernels (i.e., a prior over the set of $\{\tilde{\mathbf{k}}_i^{(\cdot)}\}$ for each $i \in \{1, ..., P_{spk}\}$). We describe the priors of the stimulus kernels for each of the nested models.

1. No category or direction model. We considered that the sample and test kernels would likely be correlated if they reflect the dynamics of common bottom-up sensory input. To construct a correlated prior, we assumed that the kernel could be constructed as the sum of a stimulus-independent kernel (a response purely to contrast or motion in general), and a sample kernel or a test kernel (responses to the task epoch):

$$\alpha_{i,0} \sim \mathcal{N}(0, \psi_0^2), \quad \alpha_{i,sample} \sim \mathcal{N}(0, \psi_s^2), \quad \alpha_{i,test} \sim \mathcal{N}(0, \psi_s^2) \quad (12)$$

such that

$$\tilde{\mathbf{k}}_i^{(sample)} = \alpha_{i,0} + \alpha_{i,sample}, \quad \tilde{\mathbf{k}}_i^{(test)} = \alpha_{i,0} + \alpha_{i,test}. \quad (13)$$

Using the rules of linear transformations of Gaussian variables, we obtained a correlated Gaussian prior over the original two sample and test kernels. The set of hyperparameters was $\mathcal{H}_{stim} = \{\psi_0, \psi_s\}$.

2. Category-only model. For the category-dependent kernels, we made a similar Gaussian construction

$$\beta_{i,0} \sim \mathcal{N}(0, \psi_0^2), \quad \beta_{i,csk} \sim \mathcal{N}(0, \psi_c^2), \quad \beta_{i,ctk} \sim \mathcal{N}(0, \psi_c^2) \quad (14)$$

such that

$$\tilde{\mathbf{k}}_i^{(csk)} = \beta_{i,0} + \beta_{i,csk}, \quad \tilde{\mathbf{k}}_i^{(ctk)} = \beta_{i,0} + \beta_{i,ctk}. \tag{15}$$

The set of hyperparameters was $\mathcal{H}_{stim} = \{\psi_0, \psi_c\}$.

3. Cosine direction tuning model. We used the same priors for the four category kernels as in the category-only model. We placed an independent Gaussian prior over the cosine and sine weights:

$$\tilde{\mathbf{k}}_i^{(cos)} \sim \mathcal{N}(0, \psi_d^2), \quad \tilde{\mathbf{k}}_i^{(sin)} \sim \mathcal{N}(0, \psi_d^2). \tag{16}$$

The set of hyperparameters was $\mathcal{H}_{stim} = \{\psi_0, \psi_d, \psi_c\}$.

4. Full model. For constructing the prior over individual direction kernels, we assumed that direction tuning should be smooth as a function of angle. We therefore used a Gaussian process prior over the direction weights. To nest the cosine-tuning model in the full model and provide regularization, we also included latent sine and cosine direction weighting. Sample category weights were included as before. We define the pieces of the prior as

$$
\begin{aligned}
\gamma_{i,0} &\sim \mathcal{N}(0, \psi_0^2), & \gamma_{i,csk} &\sim \mathcal{N}(0, \psi_c^2), \\
\gamma_{i,\cos} &\sim \mathcal{N}(0, \psi_d^2), & \gamma_{i,\sin} &\sim \mathcal{N}(0, \psi_d^2), \\
\gamma_{i,gp}(\theta) &\sim \mathcal{GP}\left(0, \psi_\theta^2 K(\theta, \theta')\right),
\end{aligned}
\tag{17}
$$

where the Gaussian process kernel over angle is[69]

$$
\begin{aligned}
K(\theta, \theta') &= \left(1 + \frac{\tau+4}{\pi} d(\theta, \theta')\right)\left(1 - \frac{1}{\pi} d(\theta, \theta')\right)^{\tau+4}, \\
d(\theta, \theta') &= \arccos(\cos(\theta - \theta')).
\end{aligned}
\tag{18}
$$

The hyperparameter $\tau \geq 0$ determined the arc length over which similar directions are correlated, similar to a length scale in Gaussian process kernels on the real line. The complete direction plus sample category kernel was then defined as for each direction $d \in \{1, \dots, D\}$

$$\tilde{\mathbf{k}}_i^{(\theta_d)} = \gamma_{i,0} + \gamma_{i,csk} + \gamma_{i,\cos}\cos(\theta_d) + \gamma_{i,\sin}\sin(\theta_d) + \gamma_{i,gp}(\theta_d). \tag{19}$$

The test category prior was defined slightly differently than in the previous two models due to the identifiability constraints on our parameterization. We defined the prior using the construction

$$
\begin{aligned}
\gamma_{i,ctk} &\sim \mathcal{N}(0, \psi_c^2), \\
\tilde{\mathbf{k}}_i^{(ctk)} &= \gamma_{i,0} + \gamma_{i,ctk} - \gamma_{i,csk}.
\end{aligned}
\tag{20}
$$

Because $\tilde{\mathbf{k}}^{(\theta_d)}$ and $\tilde{\mathbf{k}}^{(ctk)}$ were again simply linear functions of Gaussian variables, we obtained a Gaussian prior with zero mean for the kernels depending on the hyperparameters set $\mathcal{H}_{stim} = \{\psi_0, \psi_d, \psi_c, \psi_\theta, \tau\}$.

For the supplementary models with independent sample and test direction tuning, the priors followed the same construction as above. The direction hyperparameters were shared for the sample and test direction kernels.

We placed an i.i.d. Gaussian prior on the spike history and touch-bar coefficients

$$
\begin{aligned}
\tilde{\mathbf{k}}_i^{tbar} &\sim \mathcal{N}(0, \psi_{tbar}) & \text{for } i \in \{1, \dots, P_{tbar}\} \\
\tilde{\mathbf{h}}_j^{spk} &\sim \mathcal{N}(0, \psi_{spk}) & \text{for } j \in \{1, \dots, P_{spk}\}
\end{aligned}
\tag{21}
$$

The prior over $w$ was the improper uniform prior: $p(w) \propto 1$.

The complete set of hyperparameters was therefore $\mathcal{H} = \{\mathcal{H}_{stim}, \psi_{tbar}, \psi_{spk}\}$. We defined hyperpriors over each hyperparameter independently as half-$t$ distributions[70]. For each $h \in \mathcal{H}$

$$p(h) \propto \left(1 + \frac{1}{\nu}h\right)^{-(\nu+1)/2} \tag{22}$$

where we set $\nu = 4$.

**MAP estimation with evidence optimization.** We fit the GLMs to each LIP cell using MAP estimation. To set the hyperparameters for the GLM, we used an approximate evidence optimization procedure[61,71,72]. We used a Laplace approximation of the evidence (i.e., the marginal distribution of the data given the hyperparameters), which is performed by taking a Gaussian approximation of the posterior distribution of the parameters given the hyperparameters. We then optimized the log posterior over the hyperparameters. Because the hyperparameters are constrained to be positive, we optimized the log-transformed hyperparameters. We set the hyperparameters and parameters of the GLM independently for each fold of cross-validation.

Specifically, we maximized an approximation of the log posterior of the hyperparameters given the data. The posterior is

$$p(\mathcal{H}|\mathbf{y}, \mathbf{x}) \propto p(\mathbf{y}|\mathcal{H}, \mathbf{x})p(\mathcal{H}) \tag{23}$$

and we want to find

$$\mathcal{H}_{MAP} = \arg\max_{\mathcal{H}} p(\mathcal{H}|\mathbf{y}, \mathbf{x}). \tag{24}$$

The evidence term can be written using Bayes' rule as

$$p(\mathbf{y}|\mathcal{H}, \mathbf{x}) = \frac{p(\mathbf{y}|\phi, \mathbf{x})p(\phi|\mathcal{H})}{p(\phi|\mathcal{H}, \mathbf{y}, \mathbf{x})} \tag{25}$$

where $\phi$ denotes the model parameters. The posterior over the parameters ($p(\phi|\mathcal{H}, \mathbf{y}, \mathbf{x})$) is only given up to an intractable normalizing constant. We therefore took a Laplace approximation of the posterior distribution over parameters. The Laplace approximation was a Gaussian distribution centered around the MAP estimate of the parameters

$$
\begin{aligned}
p(\phi|\mathcal{H}, \mathbf{y}, \mathbf{x}) &\approx \mathcal{N}(\phi; \phi_{MAP}, \Sigma_{MAP}), \\
\phi_{MAP} &= \arg\max_{\phi} p(\mathbf{y}|\phi, \mathbf{x})p(\phi|\mathcal{H}) \\
\Sigma_{MAP}^{-1} &= -\frac{d^2}{d\phi^2}\log p(\phi|\mathcal{H}, \mathbf{y}, \mathbf{x})\Big|_{\phi = \phi_{MAP}}.
\end{aligned}
\tag{26}
$$

The MAP estimate given the hyperparameters, $\phi_{MAP}$, was found numerically (the log posterior over the parameters is log concave). Given this approximation, we evaluated the right side of Eq. (25) at $\phi_{MAP}$. We then maximized the log posterior over the hyperparameters Eq. (23) numerically to find $\mathcal{H}_{MAP}$. The final MAP estimate of the models parameters was $\phi_{MAP}$ given $\mathcal{H}_{MAP}$.

## GMLM definition

The GMLM is a special case of the GLM in which the linear kernels in a population of neurons are assumed to share low-dimensional structure, rather than being modeled independently. In general, the model is a GLM in which the regressors and parameters (or a subset thereof) from all the neurons in a population can be expressed as tensors (or multi-way arrays). The parameters (or a subset of the parameters) are then assumed to have a low-rank structure: the parameter tensor can be decomposed into a small number of components. Because we do not include interactions between neurons here (i.e., noise correlations; the shared filters account for signal correlations), this model can be applied to a set of single-neuron recordings. However, the model can readily be extended to include interactions in simultaneously recorded populations[68,73].

Here, we define the GMLM for the categorization task. Introducing an index for neuron $n \in \{1, 2, ..., N\}$, we defined the model for the spike count in bin $t \in \{1, 2, ..., T\}$ for neuron $n$ as

$$\lambda_n(t) = \mathbf{w}_n + \left(\mathbf{H}_n^{\text{spk}} * \mathbf{Y}_n\right)(t) + \sum_{q=1}^{R_l}\left(\mathbf{T}_q^{\text{tbar}} * x^{\text{tbar}}\right)(t)\mathbf{V}_{n,q}^{\text{tbar}}$$
$$+ \sum_{s \in \{\text{sample, test}\}}\sum_{r=1}^{R_s}\mathbf{Z}_r^{(s)}(t)\mathbf{V}_{n,r}^{\text{stim}}, \tag{27}$$

$$\mathbf{Z}_r^{(s)}(t) = \left(\mathbf{x}^{(\text{direction},s)} \cdot \mathbf{U}_r^{\text{stim}}\right) \cdot \left(\mathbf{T}_r^{\text{stim}} * x^{(\text{timing},s)}\right)(t), \tag{28}$$

$$Y_n(t) \sim \text{Poisson}(f(\lambda_n(t))\Delta). \tag{29}$$

The matrices $\mathbf{H}_n^{\text{spk}}$, $\mathbf{T}^{\text{tbar}}$, and $\mathbf{T}^{\text{stim}}$ denote matrices whose columns are temporal kernels for the stimulus, touch-bar and spike history respectively. Subscripts of those matrices indicate a particular column or kernel. Similarly, $\mathbf{H}^{\text{spk}}$ contains the spike history kernels for each cell. The length $N$ vector $\mathbf{w}$ contains the baseline firing rates for each neuron. The baseline firing rates and spike history kernels are equivalent to the single-cell GLM.

We note that in this model, both the regressors and the parameters are decomposed into components (for the stimulus parameters $\mathbf{T}^{\text{stim}}$, $\mathbf{U}^{\text{stim}}$, and $\mathbf{V}^{\text{stim}}$ and regressors $\mathbf{x}^{(\text{direction}, s)}$ and $x^{(\text{timing},s)}$), and thus we have a simple multilinear form for the stimulus tuning rather than writing out a dense tensor.

The touch-bar kernels are parameterized as low-rank matrix factorization where $\mathbf{T}^{\text{tbar}}$ contains $R_l$ temporal kernels and $\mathbf{V}^{\text{tbar}}$ is a matrix of neuron loading weights of size $N \times R_l$. In our notation, $\mathbf{T}_q^{\text{tbar}}$ denotes the $q$th column or kernel, and $\mathbf{V}_{n,q}^{\text{tbar}}$ is the element in the $n$th row, $q$th column of $\mathbf{V}^{\text{tbar}}$. The touch-bar subspace is the span of the columns of $\mathbf{V}^{\text{tbar}}$. Thus, the model effectively approximates the GLM touch-bar kernel for neuron $n$ as $\mathbf{k}_n^{\text{tbar}} \approx \mathbf{T}^{\text{tbar}}\mathbf{V}_{n,\cdot}^{\text{tbar}\top}$. The touch-bar function, $x^{\text{tbar}}$, is the same as in the GLM.

The stimulus kernels were parameterized as a tensor factorization of rank $R_s$. As we did for the GLM, we defined a matching set of nested models to parameterize the categorization task. As with the GLM definitions, we defined the regressors for a single trial for simplicity of notation. The notation for the stimulus timing and directions are the same as in the GLM definition.

The set of temporal kernels, $\mathbf{T}^{\text{stim}}$, did not depend on the stimulus direction or category. The temporal regressors were point events representing the stimulus onset time for each $s \in \{\text{sample, test}\}$. These were the same for all GMLM parameterizations (Supplementary Fig. 3E):

$$x^{(\text{timing,sample})}(t) = \begin{cases} 1 & \text{if } t = t_{\text{sampleon}} \\ 0 & \text{otherwise.} \end{cases},$$
$$x^{(\text{timing,test})}(t) = \begin{cases} 1 & \text{if } t = t_{\text{teston}} \\ 0 & \text{otherwise.} \end{cases} \tag{30}$$

The set of stimulus weights, $\mathbf{U}^{\text{stim}}$, was a matrix $S \times R_s$ of coefficients for the particular stimulus identity (for example, weights to encode sample, test, direction, and category) where $S$ is the same as the number of stimulus kernels in the matching GLM. Each observation had two stimulus direction regressor vectors: $\mathbf{x}^{(\text{direction, sample})}$ and $\mathbf{x}^{(\text{direction, test})}$. The entries of the stimulus direction regressors ($\mathbf{x}^{(\text{direction}, s)}$) mirrored the kernel structure in the GLM parameterizations (this vector is constant for all $t$ in a single trial). The stimulus direction coefficients depended on the model.

1. No category or direction model (Supplementary Fig. 3F). This model contained two stimulus regressor elements indexed by

{sample, test}, As with the GLM, these elements represent the identity of a stimulus event as sample or test, but do not include category or direction information.

$$\mathbf{x}_{\text{sample}}^{(\text{direction,sample})} = 1, \quad \mathbf{x}_{\text{test}}^{(\text{direction,sample})} = 0,$$
$$\mathbf{x}_{\text{sample}}^{(\text{direction,test})} = 0, \quad \mathbf{x}_{\text{test}}^{(\text{direction,test})} = 1. \tag{31}$$

2. Category-only model (Supplementary Fig. 3G). This model includes four stimulus direction regressors representing the stimulus category and whether it is sample or test. For the indices {cs1, cs2, ct1, ct2}, the regressors are

$$\mathbf{x}_{\text{csk}}^{(\text{direction,sample})} = \begin{cases} 1 & \text{if } c_{\text{sample}} = c_k \\ 0 & \text{otherwise.} \end{cases}, \quad \mathbf{x}_{\text{ctk}}^{(\text{direction,sample})} = 0$$
$$\mathbf{x}_{\text{ctk}}^{(\text{direction,test})} = \begin{cases} 1 & \text{if } c_{\text{test}} = c_k \\ 0 & \text{otherwise.} \end{cases}, \quad \mathbf{x}_{\text{csk}}^{(\text{direction,test})} = 0, \tag{32}$$

for category $k \in \{1, 2\}$.

3. Cosine direction tuning model (Supplementary Fig. 3H). The cosine-tuning model included six stimulus regressors representing the identity of a stimulus event as sample or test, the motion category, and the sine and cosine of the direction. The regressors are indexed by {cs1,cs2,ct1,ct2, cos , sin}. The category terms are the same as in the above model. The direction tuning components are defined as cosine and sine weights of the direction:

$$\mathbf{x}_{\cos}^{(\text{direction,sample})} = \cos(\theta_{\text{sample}}), \quad \mathbf{x}_{\sin}^{(\text{direction,sample})} = \sin(\theta_{\text{sample}}),$$
$$\mathbf{x}_{\cos}^{(\text{direction,test})} = \cos(\theta_{\text{test}}), \quad \mathbf{x}_{\sin}^{(\text{direction,test})} = \sin(\theta_{\text{test}}). \tag{33}$$

4. Full model (Supplementary Fig. 3I). This model includes one regressor for each stimulus direction and two for the test stimulus category indexed by the $D + 2$ coefficients in {ct1, ct2, $\theta_1$, ..., $\theta_D$}. For each $d \in \{1, ..., D\}$

$$\mathbf{x}_{\theta_d}^{(\text{direction,sample})} = \begin{cases} 1 & \text{if } \theta_{\text{sample}} = \theta_d \\ 0 & \text{otherwise.} \end{cases}$$
$$\mathbf{x}_{\theta_d}^{(\text{direction,test})} = \begin{cases} 1 & \text{if } \theta_{\text{test}} = \theta_d \\ 0 & \text{otherwise.} \end{cases} \tag{34}$$

The test category regressors (indexed by ct1 and ct2) are the same as in the previous two models. As with the full GLM, the specific model construction does not include additional weights for the sample category for identifiability. The coefficients in $\mathbf{U}^{\text{stim}}(\theta_i)$ represent the tuning strength for direction $\theta_i$ plus the tuning for the category of $\theta_i$. Therefore, the coefficients in $\mathbf{U}^{\text{stim}}(\text{ctk})$ represent the tuning for a test stimulus of category $k$ minus the tuning for sample stimulus of category $k$.

Together, the matrices $\mathbf{T}^{\text{stim}}$, $\mathbf{U}^{\text{stim}}$, and $\mathbf{V}^{\text{stim}}$ define a CP or PARAFAC decomposition of the GLM stimulus kernels over a population of cells[74]. That is, the $s$th stimulus kernel at time $t$ for neuron $n$ is approximated as the low-rank decomposition

$$\mathcal{K}(t,s,n) = \sum_{r=1}^{R_s}\mathbf{T}_r^{\text{stim}}(t)\mathbf{U}_{s,r}^{\text{stim}}\mathbf{V}_{n,r}^{\text{stim}} \tag{35}$$

Another way to view the dimensionality reduction is that the values of $\mathbf{Z}_{\cdot}^{(s)}(t)$ give an $R_s$ dimensional representation of the response to each stimulus over time. Each neuron's response to the stimulus is given as a linear projection of that low-dimensional stimulus with weights defined as the rows of the matrix $\mathbf{V}^{\text{stim}}$ so that $\mathbf{V}^{\text{stim}}$ defines the stimulus subspace.

The temporal kernels of the GMLM are parameterized using the same basis set as the GLM:

$$
\begin{aligned}
\mathbf{H}^{\text{spk}} &= \mathbf{B}^{\text{spk}}\tilde{\mathbf{H}}^{\text{spk}}, &&\text{where } \tilde{\mathbf{H}}^{\text{spk}} \text{ is a matrix of size } P_{\text{spk}} \times N, \\
\mathbf{T}^{\text{tbar}} &= \mathbf{B}^{\text{tbar}}\tilde{\mathbf{T}}^{\text{tbar}}, &&\text{where } \tilde{\mathbf{T}}^{\text{tbar}} \text{ is a matrix of size } P_{\text{tbar}} \times R_l, \\
\mathbf{T}^{\text{stim}} &= \mathbf{B}^{\text{stim}}\tilde{\mathbf{T}}^{\text{stim}}, &&\text{where } \tilde{\mathbf{T}}^{\text{stim}} \text{ is a matrix of size } P_{\text{stim}} \times R_s.
\end{aligned}
\tag{36}
$$

We set $R_l = 3$ for all GMLM fits and we selected $R_s$ using cross-validation (see section "Rank selection"). The set of parameters that are fit to the data from all the trials in an LIP population is $\phi = \left\{ \mathbf{w}, \mathbf{V}^{\text{stim}}, \mathbf{V}^{\text{tbar}}, \mathbf{U}^{\text{stim}}, \tilde{\mathbf{H}}^{\text{spk}}, \tilde{\mathbf{T}}^{\text{tbar}}, \tilde{\mathbf{T}}^{\text{stim}} \right\}$.

For Bayesian inference, we defined prior distributions the same way we did for the GLM. The prior for the stimulus kernels was defined independently for each component $r$ of the stimulus direction regressor matrix (i.e., each column $\mathbf{U}^{\text{stim}}_{:,r}$). The vectors $\mathbf{U}^{\text{stim}}_{:,r}$ and corresponding GLM kernel parameters $\tilde{\mathbf{k}}^{(\cdot)}_i$ are the same length and are indexed by the same set of stimulus events The prior over the vector $\mathbf{U}^{\text{stim}}_{:,r}$ was therefore the same as the prior over the $\tilde{\mathbf{k}}^{(\cdot)}_i$ for the corresponding GLM. The same stimulus hyperparameter set ($\mathcal{H}_{\text{stim}}$) was used for each model parameterization. However, unlike the single-cell GLM fits, the hyperparameters were shared across all neurons in each LIP population.

The prior for the entries of the spike history kernels, $\tilde{\mathbf{H}}^{\text{spk}}$, was i.i.d. normal with zero mean and variance $\psi^2_{\text{spk}}$. Similarly, the prior for the entries of the touch-bar kernels, $\tilde{\mathbf{T}}^{\text{tbar}}$, was i.i.d. normal with zero mean and variance $\psi^2_{\text{tbar}}$. The prior distribution on the entries of neuron loading matrices ($\mathbf{V}^{\text{stim}}$ and $\mathbf{V}^{\text{tbar}}$) and $\mathbf{T}^{\text{stim}}$ was i.i.d. standard normal. We again used an improper uniform prior on $\mathbf{w}$.

The complete hyperparameters set was $\mathcal{H} = \{\mathcal{H}_{\text{stim}}, \psi_{\text{tbar}}, \psi_{\text{spk}}\}$. The hyperpriors were the same half-$t$ distributions used for the GLM Eq. (22). We note that these priors only affected the GMLM in the MCMC analysis, as rank selection used the maximum likelihood estimate.

**Dynamic spike history.** We augmented the log rate in the GMLM with low-rank dynamic spike history components to allow spike history to change over time relative to task events:

$$
\begin{aligned}
h_n(t) &= \sum_{q=1}^{R_{bh}} \mathbf{Z}^{(\text{bdspk})}_q(t)\mathbf{V}^{\text{bdspk}}_{n,q} + \sum_{s\in\{\text{sample,test}\}}\sum_{r=1}^{R_h} \mathbf{Z}^{(\text{dspk},s)}_r(t)\mathbf{H}^{\text{bdspk}}_q \\
\mathbf{Z}^{(\text{bdspk})}_q(t) &= \left(\mathbf{T}^{\text{bdspk}}_q * x^{\text{tbar}}\right)(t) \cdot \left(\mathbf{H}^{\text{bdspk}}_q * \mathbf{Y}_n\right)(t), \\
\mathbf{Z}^{(\text{dspk},s)}_r(t) &= \left(\mathbf{x}^{(\text{direction},s)^*} \cdot \mathbf{U}^{\text{dspk}}_r\right) \cdot \left(\mathbf{T}^{\text{dspk}}_r * x^{(\text{timing},s)}\right)(t) \cdot \left(\mathbf{H}^{\text{dspk}}_r * \mathbf{Y}_n\right)(t), \\
\lambda^*_n(t) &= \lambda_n(t) + h_n(t)(\text{log rate})
\end{aligned}
\tag{37}
$$

where $\lambda_n(t)$ is given by Eq. (27). For completeness, we included two dynamic spike history tensors to mirror the mean-rate filter terms: one for the motion stimuli and a second for the touch-bar release. However, we found including the touch-bar filters provided little improvement to the model's performance.

The dynamic spike history kernels $\mathbf{H}^{\text{dspk}}$ for the stimulus-dependent spike history (or $\mathbf{H}^{\text{bdspk}}$ for the touch-bar kernel) are shared across all neurons in a population. The stimulus kernels $\mathbf{T}^{\text{dspk}}$ (or $\mathbf{T}^{\text{bdspk}}$ for the touch-bar release kernel) control the contribution of the dynamic spike history kernel relative to stimulus onset (or touch-bar release) . As with the stimulus filter tensor, we allow the dynamic spike history kernels to depend on stimulus information through the stimulus scaling terms $\mathbf{U}^{\text{dspk}}$. For simplicity, we limited the stimulus scaling for the dynamic spike history in $\mathbf{x}^{(\text{direction},s)^*}$ to include only sample or test information as defined for the "No category or direction model" in the previous section. We found that including category or direction information did not significantly affect our results (results not shown). Each neuron weights the dynamic spike history components by the loading matrices $\mathbf{V}^{\text{dspk}}$ and $\mathbf{V}^{\text{bdspk}}$.

At any given time in the trial, the spike history for a neuron is still a linear function of past spiking. The "effective" spike history kernel for a neuron $n$ at time $t$ can be computed by rearranging the terms in Eq. (37) and adding the constant spike history kernel:

$$
\begin{aligned}
\mathbf{H}^{\text{spkeff}}_{n,t}(s) = \mathbf{H}^{\text{spk}}_n(s) + \sum_{q=1}^{R_{bh}} \left(\mathbf{V}^{\text{bdspk}}_{n,q}\left(\mathbf{T}^{\text{bdspk}}_q * x^{\text{tbar}}\right)(t)\right) \cdot \mathbf{H}^{\text{bdspk}}_q(s) \\
+ \sum_{s\in\{\text{sample,test}\}}\sum_{r=1}^{R_h}\left(\mathbf{V}^{\text{dspk}}_{n,r}\cdot\left(\mathbf{x}^{(\text{direction},s)^*}\cdot\mathbf{U}^{\text{dspk}}_r\right)\cdot\left(\mathbf{T}^{\text{bdspk}}_q * x^{\text{tbar}}\right)(t)\right)\cdot\mathbf{H}^{\text{dspk}}_r(s).
\end{aligned}
\tag{38}
$$

The temporal kernels were parameterized using the same basis set as before:

$$
\begin{aligned}
\mathbf{H}^{\text{bdspk}} &= \mathbf{B}^{\text{spk}}\tilde{\mathbf{H}}^{\text{spk}}, &&\text{where } \tilde{\mathbf{H}}^{\text{bdspk}} \text{ is a matrix of size } P_{\text{spk}} \times R_{bh}, \\
\mathbf{T}^{\text{bdspk}} &= \mathbf{B}^{\text{tbar}}\tilde{\mathbf{T}}^{\text{tbar}}, &&\text{where } \tilde{\mathbf{T}}^{\text{bdspk}} \text{ is a matrix of size } P_{\text{tbar}} \times R_{bh}, \\
\mathbf{H}^{\text{dspk}} &= \mathbf{B}^{\text{spk}}\tilde{\mathbf{H}}^{\text{spk}}, &&\text{where } \tilde{\mathbf{H}}^{\text{dspk}} \text{ is a matrix of size } P_{\text{spk}} \times R_h, \\
\mathbf{T}^{\text{dspk}} &= \mathbf{B}^{\text{stim}}\tilde{\mathbf{T}}^{\text{stim}}, &&\text{where } \tilde{\mathbf{T}}^{\text{dspk}} \text{ is a matrix of size } P_{\text{stim}} \times R_h.
\end{aligned}
\tag{39}
$$

The parameter set for the dynamic spike history model was $\phi^* = \phi \bigcup \left\{\tilde{\mathbf{H}}^{\text{bdspk}}, \tilde{\mathbf{T}}^{\text{bdspk}}, \mathbf{V}^{\text{bdspk}}, \tilde{\mathbf{H}}^{\text{dspk}}, \tilde{\mathbf{T}}^{\text{dspk}}, \mathbf{U}^{\text{dspk}}, \mathbf{V}^{\text{dspk}}\right\}$.

We set i.i.d. standard normal priors for $\mathbf{V}^{\text{dspk}}$, $\mathbf{V}^{\text{bdspk}}$, $\tilde{\mathbf{H}}^{\text{bdspk}}$, $\tilde{\mathbf{H}}^{\text{dspk}}$, and $\tilde{\mathbf{T}}^{\text{dspk}}$. The prior for $\tilde{\mathbf{T}}^{\text{bdspk}}$ was i.i.d. zero-mean normal with variance $\psi^2_{\text{bdspk}}$. The Gaussian prior for $\mathbf{U}^{\text{dspk}}$ was defined analogously to the stimulus term (for the no category model) with hyperparameters $\psi^*_0$ and $\psi^*_s$. The complete hyperprior set was then $\mathcal{H}^* = \mathcal{H}\bigcup\{\mathcal{H},\psi_{\text{bdspk}},\psi^*_0,\psi^*_s\}$.

For all dynamic spike history models here, we set $R_{bh} = 1$ and we varied $R_h$ from 1 to 2. For the stimulus filter tensor, we used the cosine direction tuning model with the rank selected in Fig. 4.

**Model inference.** We performed rank selection in the GMLM by testing the cross-validated model performance of the maximum likelihood fit. We used gradient-descent methods to numerically maximize the log-likelihood for the training set. We initialized the GMLM components randomly. The entries of $\mathbf{V}^{\text{stim}}$, $\mathbf{V}^{\text{tbar}}$, and $\tilde{\mathbf{H}}^{\text{spk}}$ were generated as standard normal. The matrices $\tilde{\mathbf{T}}^{\text{stim}}$ and $\tilde{\mathbf{T}}^{\text{tbar}}$ were random ortho-normal matrices. The baseline firing rate parameters, $w$, were drawn independently from a normal distribution.

For the MAP estimates shown in Supplementary Fig. 5 and Fig. 5, we set the hyperparameters to the marginal posterior medians of each hyperparameter estimated using Markov chain Monte Carlo methods (described below). We then maximized the posterior log-likelihood given those hyperparameters.

For the Bayesian analyses of the GMLM, we used MCMC to generate samples from the posterior distribution of the model parameters and hyperparameters given all data from an LIP population. We used Hamiltonian Monte Carlo (HMC) to sample jointly from the posterior of the parameters and the log-transformed hyperparameters. The log transform on the hyperparameters ensures that the hyperparameters are positive. A detailed description of the HMC sampling algorithm is given in ref. [75]. The Hamiltonian equations were solved numerically using a leap-frog integrator with step size $\epsilon$ for $S$ steps. We set $S = \min\left(100, \lceil\frac{1}{\epsilon}\rceil\right)$ where $\lceil\cdot\rceil$ denotes the ceiling operator. The maximum number of steps was 100 to limit computational costs per sample. However, after tuning the sampler during warmup, we found that $S < 100$.

We denote the vectorized set of all parameters and log-transformed hyperparameters for sample $s$ as $\Phi^{(s)} = \{\phi, \mathcal{H}\}$. We initialize the model parameters for the sampler ($s = 1$) by initializing the parameters randomly as we did for the maximum likelihood estimation. The log hyperparameters were initialized as i.i.d. draws from a

standard normal distribution. The HMC sampler requires specifying a $P \times P$ mass matrix, $\mathbf{M}$. Because the model is high dimensional, we assume $\mathbf{M}$ is diagonal.

We tuned the parameters of the sampler ($\epsilon$ and $\mathbf{M}$) by generating 25,000 warmup samples (also known as "burn-in"). The initial value of the step size was $\epsilon = 0.01$. We used the dual-averaging algorithm of [76] to adapt $\epsilon$ at each step for the first 24,000 warmup samples (and fixed for the last 1000 warmup samples) to achieve a desired acceptance rate. We used the parameters given in ref. [77] to control the learning rate and target sample acceptance rate (80%). The mass matrix $\mathbf{M}$ was set at three steps.

1. Sample 1: $\mathbf{M}$ is initialized as an identity matrix.
2. Sample 4001: $\mathbf{M} = \mathrm{Diag}\left(\mathrm{cov}(\Phi_i^{(2001:4000)})^{-1}\right)$. The diagonal is the inverse empirical variance of each parameter given samples 2001–4000.
3. Sample 19,001: $\mathbf{M} = \mathrm{Diag}\left(\mathrm{cov}(\Phi_i^{(4001:19,000)})^{-1}\right)$.

After warmup, we generated 50,000 HMC samples. These samples were used as the estimate of the posterior distribution of the model parameters and hyperparameters.

One source of autocorrelation in the HMC sampler that could reduce the quality of inference is that the GMLM tensor components could be rescaled without changing the likelihood. For any $a, b \neq 0$, the $r$th component of the GMLM stimulus kernel tensor can be rescaled

$$\mathbf{U}^{\mathrm{stim}}_{\cdot,r} \leftarrow a\mathbf{U}^{\mathrm{stim}}_{\cdot,r}, \quad \mathbf{T}^{\mathrm{stim}}_{\cdot,r} \leftarrow b\mathbf{T}^{\mathrm{stim}}_{\cdot,r}, \quad \mathbf{V}^{\mathrm{stim}}_{\cdot,r} \leftarrow \frac{1}{ab}\mathbf{V}^{\mathrm{stim}}_{\cdot,r} \tag{40}$$

without changing the resulting kernel tensor. Thus, the log-likelihood remains constant. One way to is to constrain fix the norm of two of those vectors, and thereby disallowing re-scaling. Inference can then be performed for those parameters on an appropriate manifold (product of sphere manifolds) using geodesic Monte Carlo methods[78,79]. Instead, we took a different approach by including an efficient Metropolis-Hastings (MH) step for rapidly traversing the locally flat region of the likelihood without additional constraints on the model parameters. The MH step was performed independently for each component $r$. We define the current sample $s$

$$u^{(s)} = \| \mathbf{U}^{\mathrm{stim}}_{\cdot,r} \|, \quad t^{(s)} = \| \mathbf{T}^{\mathrm{stim}}_{\cdot,r} \|, \quad v^{(s)} = \| \mathbf{V}^{\mathrm{stim}}_{\cdot,r} \|, \tag{41}$$

$$\mathbf{u}^{(s)} = \frac{1}{u^{(s)}}\mathbf{U}^{\mathrm{stim}}_{\cdot,r}, \quad \mathbf{t}^{(s)} = \frac{1}{t^{(s)}}\mathbf{T}^{\mathrm{stim}}_{\cdot,r}, \quad \mathbf{v}^{(s)} = \frac{1}{v^{(s)}}\mathbf{V}^{\mathrm{stim}}_{\cdot,r}, \tag{42}$$

$$\zeta^{(s)} = u^{(s)}t^{(s)}v^{(s)}. \tag{43}$$

The prior probabilities for each $\mathbf{U}^{\mathrm{stim}}_{\cdot,r}$, $\mathbf{T}^{\mathrm{stim}}_{\cdot,r}$, and $\mathbf{V}^{\mathrm{stim}}_{\cdot,r}$ are multivariate Gaussian with zero mean. Therefore, the prior probability of the vector lengths $p(u^{(s)},t^{(s)},v^{(s)}|\mathbf{u}^{(s)},\mathbf{t}^{(s)},\mathbf{v}^{(s)},\mathcal{H}^{(s)}_{\mathrm{stim}})$ can be factorized into independent chi distributions:

$$p(u^{(s)}|\mathbf{u}^{(s)},\mathcal{H}^{(s)}_{\mathrm{stim}}) = \frac{\eta_u^S}{2^{S/2-1}\Gamma(S/2)}(u^{(s)})^{S-1}\exp\left(-(\eta_u u^{(s)})^2/2\right)$$

$$p(t^{(s)}|\mathbf{t}^{(s)},\mathcal{H}^{(s)}_{\mathrm{stim}}) = \frac{\eta_t^{P_{\mathrm{stim}}}}{2^{P_{\mathrm{stim}}/2-1}\Gamma(P_{\mathrm{stim}}/2)}(t^{(s)})^{P_{\mathrm{stim}}-1}\exp\left(-(\eta_t t^{(s)})^2/2\right)$$

$$p(v^{(s)}|\mathbf{v}^{(s)},\mathcal{H}^{(s)}_{\mathrm{stim}}) = \frac{\eta_v^N}{2^{N/2-1}\Gamma(N/2)}(v^{(s)})^{N-1}\exp\left(-(\eta_v v^{(s)})^2/2\right) \tag{44}$$

$$\eta_u = \left(\mathbf{u}^{(s)\top}\Sigma_u^{-1}\mathbf{u}^{(s)}\right)^{1/2}, \quad \eta_t = \left(\mathbf{t}^{(s)\top}\mathbf{t}^{(s)}\right)^{1/2}, \quad \eta_v = \left(\mathbf{v}^{(s)\top}\mathbf{v}^{(s)}\right)^{1/2} \tag{45}$$

where $\Sigma_u$ is the prior covariance matrix for $\mathbf{U}^{\mathrm{stim}}_{\cdot,r}$ given the hyperparameters $\mathcal{H}^{(s)}_{\mathrm{stim}}$ (the Gaussian priors for the other two vectors have

identity covariance). Our goal is to construct an MH proposal to focus on the case where the total component norm $\zeta^{(s)}$ is constant. Therefore, we perform a change of variables on the prior over $p(u^{(s)},t^{(s)},v^{(s)}|\mathbf{u}^{(s)},\mathbf{t}^{(s)},\mathbf{v}^{(s)},\mathcal{H}^{(s)}_{\mathrm{stim}})$ to $p(u^{(s)},t^{(s)},\zeta^{(s)}|\mathbf{u}^{(s)},\mathbf{t}^{(s)},\mathbf{v}^{(s)},\mathcal{H}^{(s)}_{\mathrm{stim}})$ in order to compute

$$p(u^{(s)},t^{(s)}|\zeta^{(s)},\mathbf{u}^{(s)},\mathbf{t}^{(s)},\mathbf{v}^{(s)},\mathcal{H}^{(s)}_{\mathrm{stim}}) \propto p(u^{(s)},t^{(s)},\zeta^{(s)}|\mathbf{u}^{(s)},\mathbf{t}^{(s)},\mathbf{v}^{(s)},\mathcal{H}^{(s)}_{\mathrm{stim}})$$

$$= \frac{1}{u^{(s)}t^{(s)}}p(u^{(s)}|\mathbf{u}^{(s)},\mathcal{H}^{(s)}_{\mathrm{stim}})p(t^{(s)}|\mathbf{t}^{(s)},\mathcal{H}^{(s)}_{\mathrm{stim}})p(v^{(s)}|\mathbf{v}^{(s)},\mathcal{H}^{(s)}_{\mathrm{stim}}) \tag{46}$$

We can then generate independent scaling factors to perform a random walk on the vector norms:

$$s_u \sim \mathrm{Lognormal}(0,\omega^2), \quad s_t \sim \mathrm{Lognormal}(0,\omega^2)$$
$$u^* = s_u u^{(s)}, \qquad t^* = s_t t^{(s)}, \qquad v^* = \frac{1}{s_u s_t}v^{(s)} \tag{47}$$

We then accept the proposal $u^*, t^*, v^*$ with the MH acceptance probability

$$A(\{u^*,t^*,v^*\},\{u^{(s)},t^{(s)},v^{(s)}\})$$

$$= \min\left[1, \frac{p\left(u^*,t^*|\zeta^{(s)},\mathbf{u}^{(s)},\mathbf{t}^{(s)},\mathbf{v}^{(s)},\mathcal{H}^{(s)}_{\mathrm{stim}}\right)q(u_s^{-1})q(t_s^{-1})}{p\left(u^{(s)},t^{(s)}|\zeta^{(s)},\mathbf{u}^{(s)}\right),\mathbf{t}^{(s)},\mathbf{v}^{(s)},\mathcal{H}^{(s)}_{\mathrm{stim}})q(s_u)q(s_t)}\right] \tag{48}$$

where $q(s) = \mathrm{Lognormal}(s; 0, \omega^2)$. Because the likelihood remains constant in this proposal, we only need to compute the prior to determine the MH acceptance probability. As a result, this step is very fast to compute. We applied the same class of MH proposal to the touch-bar components. We set $\omega = 0.2$ and interleaved 10 MH steps for each tensor component between every HMC step.

**Rank selection.** We applied cross-validation to select the stimulus kernel tensor rank ($R_s$) for the GMLM. To do so, we computed the mean test log-likelihood per trial per cell. For neuron $n$,

$$\mathrm{lp}_{\mathcal{M}}(n) = \frac{1}{M_n}\sum_{k=1}^{K}\sum_{l=1}^{M_n^k}\log p(\mathbf{y}^*_{n,k,l}|\phi_k,\mathbf{x}^*_{n,k,l},\mathcal{M}) \tag{49}$$

where $K$ is the number of folds ($K = 10$ for all the analyses conducted here). The trials in the test set are given as $\mathbf{y}^*_{n,k,l}$ and $\mathbf{x}^*_{n,k,l}$ which represent the spike train and regressors respectively for test trial $l$ in fold $k$. The number of test trials in fold $k$ for the neuron is $M_n^k$, and the total number of trials is $M_n = \sum_{k=1}^{K}M_n^k$. The model parameters for the model $\mathcal{M}$ fit to the training data for fold $k$ is $\phi_k$.

We then took the average across all cells

$$\overline{\mathrm{lp}}_{\mathcal{M}} = \frac{1}{N}\sum_{n=1}^{N}\mathrm{lp}_{\mathcal{M}}(n). \tag{50}$$

For normalization, we subtracted the $\overline{\mathrm{lp}}_{\mathcal{M}}$ of the GMLM without any stimulus terms (the "rank 0" model):

$$\Delta\overline{\mathrm{lp}}_{\mathcal{M}} = \overline{\mathrm{lp}}_{\mathcal{M}} - \overline{\mathrm{lp}}_{\mathcal{M}_{R_s=0}}. \tag{51}$$

Figure 4A shows the $\Delta\overline{\mathrm{lp}}_{\mathcal{M}}$ for each GLM and GMLM of from $r = 1$–12.

The fraction of log-likelihood explained by the GMLM was computed relative to the full GLM (the "full-rank" model, denoted $\mathcal{M}_{\mathrm{GLM}}$):

$$\mathrm{frac}(\mathcal{M}) = \frac{\Delta\overline{\mathrm{lp}}_{\mathcal{M}}}{\Delta\overline{\mathrm{lp}}_{\mathcal{M}_{\mathrm{GLM}}}} \tag{52}$$

Supplementary Fig. 4A shows the $\mathrm{frac}(\mathcal{M})$ for each GLM and GMLM of from $r = 1$–12 for the monkey D, category-late population. This provides a pseudo-$R^2$ (in which the GLM serves as the saturated model) which is comparable to variance explained, but more

appropriate for Poisson models than the variance explained in methods such as PCA[80]. We selected the rank $r$ for the full GMLM by selecting the smallest $r$ for which $\text{frac}(\mathcal{M}_{\text{GMLM},R_s=r}) > 0.9$ (i.e., the number of model components needed to explain 90% of the likelihood that could be explained by this GLM framework).

The error bars over the cross-validated log-likelihood in Fig. 4A were computed by computing $\overline{\text{lp}}_{\mathcal{M}}$ on the test trials for each fold separately (instead of averaging over all $K$ folds). The error bars show two standard errors of $\Delta\overline{\text{lp}}_{\mathcal{M}}$ over the folds.

For the dynamic spike history, we compared model predictive performance with leave-one-out cross-validation estimated with Pareto-smoothed importance sampling using the MCMC samples[81]. The leave-one-out cross-validated log-likelihoods were computed for each trial, and then we computed the mean cross-validated log-likelihood for each neuron.

**Visualizing the GMLM parameters.** Supplementary Fig. 5A shows the individual components of the MAP fit of the full GMLM, which included kernels for each stimulus direction and two kernels for the test stimulus category. For scale, we normalized each component by placing the magnitude of each tensor component in the neuron loading dimension:

$$\mathbf{V}^{\text{stim}}_{\cdot,r} \leftarrow \|\tilde{\mathbf{T}}^{\text{stim}}_{\cdot,r}\| \, \|\mathbf{U}^{\text{stim}}_{\cdot,r}\| \, \mathbf{V}^{\text{stim}}_{\cdot,r},$$
$$\tilde{\mathbf{T}}^{\text{stim}}_{\cdot,r} \leftarrow \frac{1}{\|\tilde{\mathbf{T}}^{\text{stim}}_{\cdot,r}\|} \tilde{\mathbf{T}}^{\text{stim}}_{\cdot,r}, \tag{53}$$

$$\mathbf{U}^{\text{stim}}_{\cdot,r} \leftarrow \frac{1}{\|\mathbf{U}^{\text{stim}}_{\cdot,r}\|} \mathbf{U}^{\text{stim}}_{\cdot,r}. \tag{54}$$

The $r$th row of the left column of Supplementary Fig. 5A shows the rescaled $\mathbf{B}^{\text{stim}}\tilde{\mathbf{T}}^{\text{stim}}_{\cdot,r}$. The middle column shows those temporal kernels scaled by the direction weights: the $r$th row plots $\mathbf{U}^{\text{stim}}_{\theta_d,r}\mathbf{T}^{\text{stim}}_r(t)$ for each direction $d$. The right column shows the temporal kernels scaled by the additional category weights for the test stimulus: the $r$th row plots for $\mathbf{U}^{\text{stim}}_{\text{ctk},r}\mathbf{T}^{\text{stim}}_r(t)$ for both categories $k$. The loading weights in the box plot of Supplementary Fig. 5B show the elements $\mathbf{V}^{\text{stim}}_{\cdot,r}$ for each component $r$.

The sample stimulus kernels for the example cells in Supplementary Fig. 5C are the sample direction kernels scaled by the neuron's loading weights for each component. The $r$ row for example neuron $n$ shows $\mathbf{V}^{\text{stim}}_{n,r}\mathbf{U}^{\text{stim}}_{\theta_d,r}\mathbf{T}^{\text{stim}}_r(t)$ for each direction $d$. The total GMLM tuning (Supplementary Fig. 5D, top row) was the sum over the $r$ components.

To visualize the subspaces, we projected the components of the full GMLM into the top three dimensions. The loading weights of the tensor decomposition used to define the model ($\mathbf{V}^{\text{stim}}$) are not constrained to be orthonormal (as is standard for the PARAFAC decomposition). Therefore, we took a higher-order singular value decomposition (HOSVD, a specific form of Tucker decomposition) to find the three-dimensional subspace that captures most of the variance in the population's stimulus-tuning structure. We took the stimulus kernel tensor of $\mathcal{K}(t,\theta_d,n)$ Eq. (35) for all sample stimulus directions. We then took the HOSVD of the stimulus kernel tensor such that

$$\mathcal{K}(t,\theta_d,n) \approx \hat{\mathcal{K}}(t,\theta_d,n) = \mathcal{T} \times_1 \hat{\mathbf{T}} \times_2 \hat{\mathbf{U}} \times_3 \hat{\mathbf{V}}$$
$$\mathcal{K}^*(t,\theta_d,i) = \frac{1}{\sqrt{N}} \mathcal{T} \times_1 \hat{\mathbf{T}} \times_2 \hat{\mathbf{U}} \tag{55}$$

where $\mathcal{T}$ is the core tensor of size $R_s \times D_{\text{sample}} \times 3$, and $\hat{\mathbf{T}}$, $\hat{\mathbf{U}}$, and $\hat{\mathbf{V}}$ are orthonormal matrices. The filter tensor projected into the top three subspace dimensions ($i \in \{1, 2, 3\}$) for each direction over time is then $\mathcal{K}^*(t,\theta_d,i)$.

To find the mean-removed space, we took

$$\overline{\mathcal{K}}(t,s,\theta_d) = \mathcal{K}(t,s,\theta_d) - \frac{1}{D_{\text{sample}}} \sum_{j \in \text{sample directions}} \mathcal{K}(t,s,\theta_j) \tag{56}$$

we performed the HOSVD on $\overline{\mathcal{K}}(t,s,\theta_d)$ to obtain the mean-removed subspace.

For visualizing the rank-1 dynamic spike history components in Fig. 10B, we plot the posterior median and pointwise 99% credible intervals computed using MCMC for the normalized temporal filters, $\mathbf{T}^{\text{dspk}}/\|\mathbf{T}^{\text{dspk}}\|$ and $\mathbf{H}^{\text{dspk}}/\|\mathbf{H}^{\text{dspk}}\|$. Because the sign of individual components in the PARAFAC decompositions is not identifiable, we set the sign of the posterior median components with the following transformation in order to better compare across populations:

$$\mathbf{T}^{\text{dspk}} \leftarrow \text{mode}(\text{sign}(\mathbf{V}^{\text{dspk}})) \cdot \text{sign}(\mathbf{H}^{\text{dspk}}(1)) \cdot \text{sign}(\mathbf{U}^{\text{dspk}}(1)) \cdot \mathbf{T}^{\text{dspk}},$$
$$\mathbf{H}^{\text{dspk}} \leftarrow \text{sign}(\mathbf{H}^{\text{dspk}}(1)) \cdot \mathbf{H}^{\text{dspk}}. \tag{57}$$

To quantify the timescales of the dynamic spike history kernels (for the pretrained monkeys only), we fit the MAP estimate of the rank-1 dynamic spike history kernel with an exponential function with a least-squares fit.

**Bayesian analysis of subspace geometry.** We defined tuning metrics in the low-dimensional space estimated by the GMLM with cosine direction tuning to analyze the geometry of task encoding. The metrics were constant over rotations and translations of the latent subspace. We used the posterior distribution of the model parameters estimated using MCMC to establish credible intervals over the metrics.

At each time point $t$ relative to stimulus onset, the cosine-tuned GMLM defines direction tuning in the population as an ellipse embedded in $\mathbb{R}^{R_s}$ parameterized by angle as

$$\mathbf{E}_t(\theta) = \sum_{r=1}^{R_s} \mathbf{T}^{\text{stim}}_r(t)\left(\mathbf{U}^{\text{stim}}_{r,\cos}\cos(\theta) + \mathbf{U}^{\text{stim}}_{r,\sin}\sin(\theta)\right)\mathbf{R}_{\cdot,r},$$
$$\mathbf{R} = \frac{1}{\sqrt{N}}\text{orth}(\mathbf{V}^{\text{stim}})^{\top}\mathbf{V}^{\text{stim}}, \tag{58}$$

where $\text{orth}(\mathbf{V}^{\text{stim}})$ denotes a matrix whose columns contain an orthonormalized basis for the span of the columns of $\mathbf{V}^{\text{stim}}$. Here, the orthogonalized $R_s$-dimensional output space, $\mathbf{R}$, is normalized by the number of cells. We computed the angle of the major axis of the ellipse as $\theta_{\max}$ where

$$\theta_{\max} = \underset{\theta}{\arg\max}\, D_t(\theta),$$
$$D_t(\theta) = \|\mathbf{E}_t(\theta) - \mathbf{E}_t(\theta + 180°)\|,$$
$$t_0 = \frac{1}{2}\text{arccot}\left(\frac{\vec{f}_1 \cdot \vec{f}_1 - \vec{f}_2 \cdot \vec{f}_2}{2}\right), \quad \vec{f}_1 = \mathbf{E}_t(0), \quad \vec{f}_2 = \mathbf{E}_t(90°), \tag{59}$$
$$\Rightarrow \theta_{\max} = t_0 \text{ or } t_0 + 90.°$$

Because $\theta_{\max}$ is identifiable only up to a factor of 180°, we added the constraint $\theta_{\max} \in [45°, 225°]$ to relate the angle to category in the task. The norms of the major and minor axes are $D_t(\theta_{\max})$ and $D_t(\theta_{\max} + 90°)$ respectively.

The category tuning vector is the difference in the low-dimensional tuning space between the category one and category two kernels:

$$\mathbf{F}_t = \sum_{r=1}^{R_s} \mathbf{T}^{\text{stim}}_r(t)\left(\mathbf{U}^{\text{stim}}_{r,\text{cs1}} - \mathbf{U}^{\text{stim}}_{r,\text{cs2}}\right)\mathbf{R}_{\cdot,r}. \tag{60}$$

Category tuning norm at each time $t$ relative to stimulus onset is then the norm of the vector, $\|\mathbf{F}_t\|$.

For the Bayesian analysis, we computed $\theta_{max}$, $D_t(\theta_{max})$, $D_t(\theta_{max} + 90°)$, and $\|\mathbf{F}_t\|$ for each sample from the posterior distribution of the model parameters. We then computed the posterior median and a 99% credible interval covering 0.5–99.5% of the posterior for each time $t$.

For the supplementary analyses in Supplementary Figs. 11 and 14, we performed component-wise analyses of the GMLM fits. We note that the GMLM posterior has multiple modes: the order of the components can be permuted or a sign flip could occur between $\mathbf{U}^{stim}$ and $\mathbf{T}^{stim}$. These modes define equivalent subspaces and kernel tensors, and the prior distributions are the same at each mode. We did not find that the HMC sampler jumped between these modes, and thus we could simply analyze the individual components of the GMLM tensor. For the component-wise analysis, we looked at each $r \in \{1, ..., R_s\}$ individually. The direction tuning for the component was

$$\theta^{(r)} = \arctan 2\left(\mathbf{U}^{stim}_{r,\sin}, \mathbf{U}^{stim}_{r,\cos}\right), \qquad \text{(angle)}$$
$$a^{(r)} = \sqrt{\mathbf{U}^{stim\,2}_{r,\sin} + \mathbf{U}^{stim\,2}_{r,\cos}}. \qquad \text{(direction magnitude)} \qquad (61)$$

The sample and test category tuning for the component was

$$C^{(r)}_{sample} = |\mathbf{U}^{stim}_{r,sample}|, \quad \text{(sample category magnitude)} \qquad (62)$$

$$C^{(r)}_{test} = |\mathbf{U}^{stim}_{r,test}|. \quad \text{(test category magnitude)} \qquad (63)$$

**Decoding analyses**

All decoders were linear, binary classifiers on pseudopopulation trials spike counts fit with logistic regression in MATLAB using the `fitclinear` function. The training set spike counts were $z$-scored and the decoder was fit with ridge regression with penalty 0.1. Because the neurons were recorded independently, we constructed pseudopopulation of 50 trials per stimulus. Each pseudopopulation trial consisted of one randomly sampled (with replacement) trial from each neuron in the recorded population for a particular stimulus direction. We repeated the decoding analysis on 1000 random pseudopopulations to obtain bootstrapped confidence intervals.

To decode sample category as a function of time from stimulus onset, we fit and tested decoders using spike counts in a sliding 200 ms window (centered at the decoding time). To test for direction-independent category tuning, training and validation conditions were trials from different directions to test direction-independent category encoding[13]. We therefore fit two decoders, each using trials only from a subset of motion directions. Generalization was evaluated using the withheld directions for each decoder, and the total generalization performance was averaged across the two decoders. The two sets of monkeys had a different set of sample directions, and thus different train/validation conditions. For monkeys B and J, each training set contained two motion directions, spaced 180° apart: {15° and 195°} and {75° and 225°}. Test sets were then the four remaining motion directions in each condition (135° and 315° trials were in the validation set for both decoders). For monkeys D and H, the training sets were {67.5°, 112.5°, 247.5° and 292.5°} or {157.5°, 202.5°, 337.5° and 22.5°}.

For decoding category decoding during the test stimulus, we used pseudopopulation spike counts in a window from 0 to 200 ms after test motion onset. For these decoders, the training and validation sets included pseudopopulation trials from all motion directions. The decoders were trained using only match (or non-match) trials and tested for generalization on non-match (or match). The total performance was the average across the match-trained and non-match-

trained decoders. We trained separate decoders for sample and test category. The discrimination populations were excluded from this analysis, because the test stimulus directions depended on the sample stimulus.

**Reporting summary**

Further information on research design is available in the Nature Portfolio Reporting Summary linked to this article.

## Data availability

The datasets analyzed during the current study are available from the corresponding authors of the original studies (refs. [13] and [22]) on reasonable request. Source Data are provided with this paper for all figures.

## Code availability

All GLM and GMLM analyses were performed using custom software for MATLAB (MathWorks) and CUDA (Nvidia). The GMLM tools are available publicly[82] and can be found at https://github.com/latimerk/GMLM_dmc for both MATLAB and Python. Higher-order singular value decompositions for visualizing the subspaces were performed with Tensor Toolbox for MATLAB[83].

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

## Acknowledgements

This work was supported by a Chicago Fellows Fellowship from The University of Chicago Biological Sciences Divison (K.W.L.) and grants NIH R01 EY019041, NIH R01 NS107609 and a DOD VBFF award (D.J.F.). We thank Rheza Budiono, Jeffrey Johnston, Pantea Moghimi, Barbara Peysakhovich, Jonathan Pillow, Matthew Rosen, Jacob Yates, and Oliver Zhu for helpful comments and discussions.

## Author contributions

K.W.L. and D.J.F. designed the study and wrote the paper. K.W.L. developed the code and performed computational analyses.

## Competing interests

The authors declare no competing interests.
