## [Peer Review File · Nature Communications]

Low-dimensional encoding of decisions in parietal cortex
reflects long-term training historyREVIEWER COMMENTS

Reviewer #1 (Remarks to the Author):

Latimer and Freedman have conducted an intriguing study examining how long-term training history impacts neural encoding of a task in the parietal cortex. This study joins several other recent endeavors (e.g., Brody and colleagues, DeAngelis and Angelaki and their colleagues) that mount the case that parietal cortex cares about history (both short-term and long-term), with the present study making a strong case for an impact of long-term history on parietal cortex. This is an important advance over the historical view of parietal cortex, that it reflects only the immediate present (e.g., sensory attention or motor intention) or the near future (e.g., updating internal models).

The questions address in the work are important and timely. The data are intriguing, and appropriate to the questions asked. The methods are sophisticated and appear rigorous. The greatest area for improvement in the manuscript is a disconnect between the questions being asked and the format of the data presented. The introduction and figure 1 provide a straightforward, albeit conventional, presentation of the task and the data. There is then a lengthy digression into an elaborate new method that is crucial to be able to digest the main findings. One is left wondering whether more-straightforward single-neuron tuning analyses, or already-established directed-dimensionality-reduction techniques might have been sufficient to demonstrate the key findings, and if so, that might make for a more impactful study.

Major concerns:

1) This study feels like part of a larger study.

Of course, this is always true for any new result, and researchers must be free to publish important advances, even when it is already pretty clear what study should come next. But here, the additional work that's needed might end up altering how we interpret the current study in hindsight. Specifically, readers will wonder what if you took a monkey in the other direction, from DMC to DMS during its training history. The main finding of the study is that monkeys first trained on DMS retain a precise encoding of stimulus memory in parietal cortex once they go on to perform DMC. What will happen when a DMC monkey needs to perform DMS? Will parietal cortex take on that precise information? What if there is an incentive to cease retaining the precise stimulus information, such as for example if several different motion directions are presented, and the animals must extract the "gist" of the category. Then, would precise direction information disappear from parietal cortex?

2) Is the GMLM method really essential here? Can more-conventional analyses, such as direct quantifications of single-unit tuning curves, be used instead to demonstrate the main findings? Or, can more-conventional dimensionality-reduction tools be employed? Some rationale is offered for the

development of a new technique - namely, being able to fit trial-by-trial information rather than just averages, and the ability to rationally construct a series of increasingly elaborate models, where each new explanatory term has a simple description. Although these benefits of the GMLM approach are nice, it would be good to know the actual value that they provide, in terms of explanatory power, perhaps beyond "directed dimensionality reduction" or other already-established tools. Overall, the results feel exploratory, despite coming to clear conclusions. A more hypothesis-driven approach might give the narrative more momentum. The authors may wish to consider starting off with familiar techniques, seeing what they already reveal about the data, and then showcasing the added explanatory power of GMLM, such as for the single-trial dynamics. Another approach here is to stick with GMLM as the main tool, but shift the explanation and validation (figs 2-4) to the supplement, and get right into the really cool things it reveals about the data. As it stands, the paper starts with a really great setup, but then we must first learn a new tool before we can appreciate and understand the findings. (A note that if standard dimensionality reduction techniques are used, they must be selected and validated to ensure that they will be usable on "pseudo-population" recordings. Other authors (e.g., Fusi, Churchland) have done these validations. This is especially important when "single-trial" dynamics are considered, because there are in fact no single-trial population recordings here.)

Minor concerns, in the order they are encountered in the manuscript:

1) Is the monkey's best strategy to ignore the stimulus and simply keep holding the touch bar? On what fraction of trials would that get him a reward? And, does that fraction differ for the two tasks? (I realize the data have already been published, and that performance is largely good, so this is definitely a minor concern. But, it does give some indication as to whether the animal is always actively attending to the task, or maybe zoning out for occasional trials.)

2) There might be a better organization and flow if the focus is on the neural mechanisms whereby the animals might perform the DMC (and in comparison, DMS). Lots of insights into the perceptual and cognitive computations are offered in the paragraph beginning on line 102. But to get there, first we must go through the GMLM technique. A suggestion to first discuss what you think is going on in the brain. As it stands, we really only get a clear sense for what you are looking for when we finally see it, in figure 5.

3) In fig 2 it's unclear what is being fit and what is being input. It seems like the things to the left of the gray box are inputs, but then where are the recorded spike trains? Also, I'm confused by a discrepancy between the statement in the caption that "Because we do not include interactions between neurons here, this model can be applied to a set of single-neuron recordings" and the statement in the results "We aimed to describe how task-related responses in LIP are shared across neurons in a population by reducing the dimensionality of the eight LIP populations using GMLMs." How can the interactions not be modeled but responses shared across neurons can be described? The resolution, probably, is that there

are "common motifs" that many neurons express, but one must think longer than is ideal to come to this realization.

4) In fig 3, the y axis of some of the temporal filters could be flipped (e.g., 1, 5, 6, 7) for a better visual match to the middle column, since those units are arbitrary, including in sign. In B, how should we interpret component 3 having smaller weights than component 2? Are the components being normalized with respect to each other? It seems like it might be interesting to see how the loading of a single unit changes across components.

5) I don't really understand the distinction between the full GMLM and the individual-neuron GLMs. Here is a place where a comparison between what GMLM gives you and what more-common techniques like dPCA (or other directed / hypothesis-driven dimensionality reduction, where you look for the best population correlate of a known signal) would show. Also, a suggestion to build a GLM from one neuron and see how well it models other neurons.

6) In fig 4, what question is answered by the dimensionality analysis? Also, it might be more intuitive for readers unfamiliar with these tools to express the y axis in units of % variance explained, rather than the log likelihood of the model. Also, what does it mean to express the y axis as a "per trial" quantity?

5) Figure 5 is where the results start to come in. Devoting figs 2-4 to methods explanation, development, and validation is a lot of "real estate" for a paper whose focus is on scientific results. Is there a way to put some of this material into the supplement, and get to the results within the first couple of figures? (I say this knowing that I just provided a lot of specific queries about figs 2-4 above. Those concerns aren't completely side-stepped if the paper is restructured to be "results-first" rather than "technique-first".)

6) Also in fig 5, how are the dimensions selected? Are these the most important ones for visualization, or are they the most-variance-explained dimensions?

7) In fig 7B, the "sample" and "test" plots look to be almost mirror-images. Is there an obligatory linking between these metrics?

8) In fig 8, it's not clear what B is showing. This is meant to be the dynamic spike history, and yet it's shown as a static single vector. Does it become dynamic because it's weighed by the stimulus-timing kernels? I actually expected to see more of what's in D, where it's different depending on the time point, but maybe that's because it's the weighting that makes it dynamic? In any event, a clearer "expectation" going into this figure will be helpful.

Reviewer #2 (Remarks to the Author):

In this work, the authors examine population-level responses in LIP during a match-to-category task. They propose a novel method for dimensionality reduction that extends a generalized linear model (GLM) by incorporating low-rank tensor decomposition. This approach (termed generalized multilinear model, or GMLM) finds a low-dimensional set of descriptors (kernels) to fit the population activity. As the authors note, this approach to dimensionality reduction explicitly incorporates task variables (reach direction, category, etc.) and can be used for tasks with flexible timing or free behavior, and even if the population activity is not recorded simultaneously. They use this method to examine how responses in LIP during a match-to-category task differ depending on the animal's training history. They report that monkeys first trained on a match-to-sample task exhibited greater direction-specific responses in LIP than did monkeys only trained on a match-to-category task. Monkeys trained only on the match-to-category task showed higher category-dependent activity. The geometry of this category-specific population activity was also different across training histories. The neural results generally point to different strategies between the two animal groups, which is supported by behavioral results that show a stronger direction-dependent effect on task success for the animals pre-trained on the match-to-sample task.

The question of how training (and/or overtraining) impacts neural activity is very relevant and an important next-step for the field. The paper is generally well written and the results (namely, the training-based heterogeneity of response characteristics in LIP) are fairly convincing. However, I have a few concerns about the implementation of the new method and absence of corroborating evidence regarding the scientific findings. As written, the manuscript attempts to both introduce a new computational approach and also use it to support a scientific finding. However, the lack of comparisons/validations using existing analyses makes it difficult to judge the validity of either one.

Major comments:

1) In general, the new method proposed (GMLM) appears to be a useful tool for achieving low-dimensional descriptions of population activity collected during different sessions with variable trial timing. This flexibility with respect to task event timing is stated by the authors as one of the primary justifications for not applying existing methods. However, the results as presented appear to suggest that the responses can be trial-aligned and averaged. All major results of the paper (e.g. Figs 1, 6, 8) are provided in PSTH-style plots.

Given that this dataset seems perfectly appropriate for trial-averaged approaches, it would be nice to see some comparison to results from known methods. If the kernels obtained via GMLM reflect true underlying low-dimensional latent activity, I would expect to see similar results from a method like dPCA. I recognize that dPCA could not entirely replace the analysis here, because category and direction

are completely correlated and not suited for dPCA. Still, my confidence in these results would be much higher if there was some comparison or validation of the results. For example, does the geometry subscribed by the GMLM kernels roughly match those found via PCA? Do other dimensionality reduction methods also show the same major/minor axis results in Figure 6? Validating the results that can be against known methods would strengthen the results that can purportedly only be achieved via GMLM (category/direction disentangling).

2) Similarly, the dynamic spike history result in Figure 8 is intriguing, but could use additional validation. Whether DMC-only animals exhibit stronger low-beta activity should certainly be testable outside of the GMLM framework using standard frequency spectra analysis. If the results confirm the same result, it would strengthen the GMLM approach. I don't know of another approach that can characterize both population-wide spike rate phenomena and spectral content, so this could be a very useful tool. However, I have no way of judging whether the dynamic spike history filter accurately reflects frequency band power.

3) The results are quite inconsistent across monkeys within the same group, but the discussion does not directly provide explanations other than possible strategy differences. Monkey H displayed significant category tuning during DMC late, but monkey D showed almost none. However, the major axis of the tuning model in Monkey D (but not monkey H) aligned with the test axis during DMC late. Are these differences between monkeys "real" or do they reflect a tradeoff during model fitting between direction and category terms? I can imagine a case in which two very similar sets of population activity give rise to very different model fits; one fit with a highly elongated tuning ellipse (2D) and the other with a near circular tuning ellipse and additional categorical component (3D). It is difficult to determine how the method itself and the choices made during fitting (such as chosen dimensionality) affect the results/interpretation.

4) The work done is rigorous and quite complete, but the large number of results in the manuscript make it difficult to address each with enough care and depth to feel that a satisfactory conclusion has been reached. For example, only one aspect of the major/minor axis result (Figure 6) was mentioned, and it was a single-monkey result related to alignment with the category axis. However, there are a number of results shown that receive little attention. Why does tuning appear to become more circular with DMC training? Does this represent a shift in strategy? Why are there no neural results shown for the test presentation (other than Fig 1c)? Why does the sign of the dynamic spike history weight flip after stimulus offset? If the results aren't going to be discussed, their inclusion only confuses the general point of training-induced differences.

Minor comments:

71: reference should be to 1c

Fig 3: What is the fit quality across neurons? The GMLM seems to leave little flexibility in the possible task-dependent modulations across the population. For example, any direction-based difference in activity can only take the temporal form of components 5 or 6. Is that correct? The single neuron reconstructions in Figure 3d look okay, but I would like to see a quantitative summary of how well the model fits the activity across the population.

207: “The category-only GMLM captured a greater percentage of the log likelihood over the no category model in the DMC late populations than in the DMS populations”

I am seeing 2.8 vs. 2.1 (monkey D) and 9.5 vs 3.3 (monkey H). Are those really significant?

214: “Cosine direction tuning during the DMC task accounted for a larger improvement of the model fit over the category only model for the pretrained monkeys than the DMC-only monkeys”

I am confused about why the inclusion of cosine tuning improved the fit more for monkey H during DMC early/late (11.9/9.9%) than monkey D (1.6/1.9%). Figure 6 seems to suggest that monkey D only exhibits directional tuning, and not categorical. This seems counterintuitive and at least requires some explanation.

222: “The cosine direction tuning model was comparable to the full model for all populations”

Perhaps report % as compared to condition-only.

266: “We conducted a Bayesian analysis of the GMLM’s sample stimulus subspace to take into account uncertainty in the model fit given the posterior distribution of the model parameters.”

What does this mean? Perhaps make a short reference to the methods.

Reviewer #3 (Remarks to the Author):

General comments:

This manuscript describes neural recordings in area LIP during two kinds of manual Go/NoGo tasks: A delayed match-to-sample (DMS) task in which monkeys compare visual motion directions; and a delayed

match-to-category (DMC) task in which they must simply indicate whether the motion directions fall within the same arbitrarily defined category. Two pairs of monkeys are studied: One pair that first learned the DMS task and later the DMC task, and a second pair that only learned the DMC task. Analyses are done by fitting neural activity with generalized multi-linear models (GMLM) that aim to reconstruct how neural activity is a sum of specific theoretical components, and then analyzing the properties of the fitted models. It is reported that stronger cosine tuning is found during the DMC task in the monkeys previously trained on the DMS task.

The key question addressed here is the question of how training history can change the properties of neural activity. In that light, it is interesting that cosine tuning is stronger in the monkeys with previous training in the DMS task. That is not very surprising of course, and perfectly compatible with the reasonable proposal that they retain some of the strategy they used to solve the DMS task when later trained in the DMC task. Indeed, their pattern of errors at category boundaries supports that notion, as if it is not quite clear to the pretrained monkeys exactly how the task has changed. I assume the hypothesis is that monkeys retain some neural properties that result from previous training and that this retention involves LIP, and this then predicts differences between the two pairs of animals. But obviously there must be differences in neural activity between the animals, if only to explain the differences in behavior. So is the key finding that these differences are seen in LIP? Judging the importance of that is not really possible unless the LIP results are compared to other regions, such as MT or V1, and maybe hand-related regions like AIP, PMv or M1. How did these results compare to what was seen in PFC (refs 6 & 7)?

On that note, why was LIP chosen as the recording site to address the question about training history? This is a motion direction task (implicating motion sensitive areas like MT) in which responses are made manually (implicating hand-related regions like AIP, PMv, M1) and the monkey must fixate (i.e. reduce responses in saccade-related regions like LIP and FEF). Sure, many people have studied LIP during decisions, but those were decisions about eye movements. I would have guessed the place to go would be MT. I understand that the senior author has found interesting results in LIP in previous studies, but what is the real hypothesis here? That LIP is a general place for working memory of visual motion? A general place for decisions about visual motion? I find these proposals very questionable, but regardless of my opinion, the hypothesis should be stated explicitly.

I have concerns about the way in which the authors analyze their data. In fact, they don't analyze the data directly, but perform a GMLM fit to build a model and then analyze the model instead. This approach is very precarious because it is entirely dependent on the assumptions that are put into the model, all of which must be true for the results to be interpretable. In particular, one can call the model "general", but it can only be as general as the components that the authors include. Here, they include the basic pieces of the paradigm (stimulus motion direction, category, timing) as if they assume that these are the components from which neural activity in LIP is constructed, through a simple linear summation, and nothing else need be considered. But where do those components come from? Furthermore, what if neurons are related to other things that are not included in the GMLM? For

example, since LIP is a saccade-related region, one might expect that even in a task with constant fixation, like this one, neural activity is related to the relative placement of the stimulus with respect to the visual motion (how the motion aligns with the potential saccade vector toward the motion field). Other factors could also be involved, such as previous trial success, expected probability of correct answers, etc., all of which have been documented in LIP in many studies. The conclusions from such a model fit are very strongly based on what components are included, and leaving one out will change the weights assigned to the others. So to trust such a model one would have to assume that all of the components are correct, none are missing, and that their linear summation is all that LIP does. Note that I'm not suggesting that the authors pile all kinds of other factors into their GMLM as well. What I'm saying is that the approach of doing GMLM fits and then viewing all of the data solely through that one lens suggests that the authors think they already know all of the components that must be included in a model of LIP. I'm not willing to go along with that.

Unfortunately, all of the analyses in the manuscript rely exclusively on data processed through the GMLM. So if the authors' assumptions about the generality of the model are false, then their results are more about departures from the model than about the data itself, and hence, uninterpretable. Maybe some conclusions survive, but how do we know which ones? In my opinion, if the findings are robust then it should be possible to demonstrate them through other, independent analytical approaches. Preferably, these would minimize excess data processing. After all, if the LIP populations in the pretrained monkeys are more cosine tuned than those in the DMC-only monkeys, then it should be trivial to show that simply by plotting distributions of tuning parameters. Why was this not done?

In general, I found the topic of this manuscript very interesting, but I'm not convinced by the manner in which the data was analyzed. If these results are correct, then it should be possible to demonstrate them through a method that does not rely on so many questionable assumptions. Unfortunately that would require rewriting the paper almost from the beginning.

Specific comments:

In figure 1B, monkey H's performance does not appear to be strongly affected by the change in the task. Also, monkey H's fraction correct in DMC early looks like a bimodal distribution. Is it because H's performance was improving rapidly (i.e., split of early-early and early-late), or was H exploring different strategies, some better than the other, on different sessions?

Line 235, "... The two DMC-only LIP populations showed primarily two-dimensional response..." This seems only true in monkey B. In monkey J the 3rd dimension is capturing something about the brightest red and darkest blue directions.

Figure 3. Are these components meant to indicate something that is present in the real brain, or are they just the bases necessary to produce a good fit? I understand the algorithm just implies the latter, but are the authors interpreting them as the former? I note that the fits shown in Figure 3D are not really so great.

In figure 6B, the estimated vector norm for monkey B looks like it has transient directional tuning during the sample presentation period. Is that what the neural data looked like?

Minor comments:

The behavioral results are presented before the results section.

Figure 3, left-align the subplot titles C and B.

Figure 5, caption, last lines "Supplementary Fig. 5 shows the three-dimensional trajectories (of the cosine-tuned model) as a function of time" (remove [are shown]).

Line 286-293, if this is about figure 6B, mentioning so here instead of at line 265 would be easier to follow.

Figure 6: Misspelled "Pseudopopulation".

Line 325-327, "We therefore compared the LIP responses to the test stimulus to the population responses to the sample stimulus." This sentence is hard to follow. Please simplify.

Line 543 "B, and N = 29..." (capitalize B).

Line 592, "functions of the stimulus basis (Supplementary Fig. 1j, middle)." (add closing parenthesis)

Line 637, " $x(\cos)$ " (replace sin with cos)

The supplementary figures are mentioned almost as if they are not supplementary, but essential to understanding the main text.

Response to Reviewers

We thank the reviewers for their many helpful comments and suggestions. Our revised manuscript has been extensively reorganized and includes several new analyses to address concerns raised by the reviewers. Briefly, our revised paper more directly highlights our hypotheses and scientific findings, and provides clearer motivation for our modeling approach. We have introduced additional analysis using single-neuron tuning curves and autocorrelations. Additionally, we conducted control analyses to validate our methods using common dimensionality reduction methods (demixed principal components analysis), and analyses to confirm the consistency of our model fitting approach. We believe these revisions have addressed the reviewers' concerns and have substantially improved the manuscript.

Our point-by-point response to each of the reviewers' comments are below. The original reviewer text is copied and shown in bold font, and our replies are in normal font.

Reviewer #1 (Remarks to the Author):

Latimer and Freedman have conducted an intriguing study examining how long-term training history impacts neural encoding of a task in the parietal cortex. This study joins several other recent endeavors (e.g., Brody and colleagues, DeAngelis and Angelaki and their colleagues) that mount the case that parietal cortex cares about history (both short-term and long-term), with the present study making a strong case for an impact of long-term history on parietal cortex. This is an important advance over the historical view of parietal cortex, that it reflects only the immediate present (e.g., sensory attention or motor intention) or the near future (e.g., updating internal models).

The questions address in the work are important and timely. The data are intriguing, and appropriate to the questions asked. The methods are sophisticated and appear rigorous. The greatest area for improvement in the manuscript is a disconnect between the questions being asked and the format of the data presented. The introduction and figure 1 provide a straightforward, albeit conventional, presentation of the task and the data. There is then a lengthy digression into an elaborate new method that is crucial to be able to digest the main findings. One is left wondering whether more-straightforward single-neuron tuning analyses, or already-established directed-dimensionality-reduction techniques might have been sufficient to demonstrate the key findings, and if so, that might make for a more impactful study.

We appreciate the reviewer's thoughtful comments and suggestions for improving the manuscript. We have performed additional analyses and revisions to address the reviewer's comments.

Major concerns:

1) This study feels like part of a larger study.

Of course, this is always true for any new result, and researchers must be free to publish important advances, even when it is already pretty clear what study should come next.

But here, the additional work that's needed might end up altering how we interpret the current study in hindsight. Specifically, readers will wonder what if you took a monkey in the other direction, from DMC to DMS during its training history. The main finding of the study is that monkeys first trained on DMS retain a precise encoding of stimulus memory in parietal cortex once they go on to perform DMC. What will happen when a DMC monkey needs to perform DMS? Will parietal cortex take on that precise information? What if there is an incentive to cease retaining the precise stimulus information, such as for example if several different motion directions are presented, and the animals must extract the "gist" of the category. Then, would precise direction information disappear from parietal cortex?

We agree that these are important questions, and we have elaborated on these points in the revised discussion (lines 771-774) We are especially interested in the possibility of DMC-to-DMS retraining or training regimes that could incentivize losing specific direction information ("training on the categorization task first could influence direction coding in the discrimination task"). However, these tasks may be too simple to cause the parietal cortex to remove learned information, because learned category from the DMC task may not interfere with direction discrimination when performing the DMS task. Therefore, to study the removal of precise stimulus representations (rather than the addition of category) would require new behavioral tasks. We believe that the new experimental work needed to conclusively answer these questions is beyond the scope of the current manuscript. However the revised discussion section more explicitly points out predictions for future experiments to address this issue, and we hope that the current work motivates more studies of population activity during category learning, and helps provide a novel statistical basis for analyzing results of those (and other) studies.

2) Is the GMLM method really essential here? Can more-conventional analyses, such as direct quantifications of single-unit tuning curves, be used instead to demonstrate the main findings? Or, can more-conventional dimensionality-reduction tools be employed? Some rationale is offered for the development of a new technique - namely, being able to fit trial-by-trial information rather than just averages, and the ability to rationally construct a series of increasingly elaborate models, where each new explanatory term has a simple description. Although these benefits of the GMLM approach are nice, it would be good to know the actual value that they provide, in terms of explanatory power, perhaps beyond "directed dimensionality reduction" or other already-established tools. Overall, the results feel exploratory, despite coming to clear conclusions. A more hypothesis-driven approach might give the narrative more momentum. The authors may wish to consider starting off with familiar techniques, seeing what they already reveal about the data, and then showcasing the added explanatory power of GMLM, such as for the single-trial dynamics. Another approach here is to stick with GMLM as the main tool, but shift the explanation and validation (figs 2-4) to the supplement, and get right into the really cool things it reveals about the data. As it stands, the paper starts with a really great setup, but then we must first learn a new tool before we can appreciate and understand the findings. (A note that

if standard dimensionality reduction techniques are used, they must be selected and validated to ensure that they will be usable on "pseudo-population" recordings. Other authors (e.g., Fusi, Churchland) have done these validations. This is especially important when "single-trial" dynamics are considered, because there are in fact no single-trial population recordings here.)

We appreciate the Reviewers' thoughtful suggestions about reorganizing the manuscript and we have incorporated these suggestions in key ways to help clarify the presentation. These major changes are:

1. We have elaborated to clearly outline our specific hypothesis about the effects of training on direction and category coding in LIP in the introduction (lines 68-102) and moved the presentation of the dataset & task details to the first section of the Results.
2. We have included analyses based on single-neuron tuning curves (Results subsection "Quantifying direction and category tuning in LIP"). We thank the reviewer for this suggestion, because it helps motivate why population-level analysis methods help analyze population encoding (i.e., if orthogonal direction and stored category are orthogonal) beyond what single-neuron tuning curves provide.
3. We expanded the first part of the GMLM results section to lay out our modeling goals: to quantify category & direction responses while using the complete trial with both sample and test stimuli. Comparing the responses across the sample and test periods is used to validate our interpretation of category and direction encoding, because the sample and test share the same stimuli (bottom-up drive) but differ in meaning for the task (section "Low-dimensional representation of direction and category in LIP").
4. As suggested, we have moved much of the initial GMLM presentation to the Supplement. Now, Figure 3 (instead of Figure 5) shows the subspaces, allowing us to get to the discussion of the main results of the study more directly. The original Figures 3 and 4 are now in the supplementary, while putting part of the rank selection summary into Figure 2. We better explain that the purpose of this figure is to confirm and quantify the low-dimensionality of the task-relevant LIP activity. However, we have kept the model diagram figure because we feel that it is an important aid to outline and explain our approach.
5. We show that we get similar subspaces from dPCA (Results section "Comparison to demixed principal components analysis"; Figure 4 and Supplementary Figures 9 and 10). Similarly, we performed PCA on the GLM fits to obtain similar results. We use this to verify and motivate our approach: our model finds good subspaces, but can do more than dPCA.
6. We have chosen to stick with the GMLM as the primary tool for our analyses. The main reason is that dPCA/PCA don't fully solve the same problems we look at with the GMLM. That is, to apply these familiar methods to get at the questions we're asking, we actually had to build a regression model on top of the output of dPCA. We found similar results doing just this (although dPCA is more noisy - as we have observed using bootstraps and simulations). However, such a model becomes so close in spirit to the GMLM that it does not simplify presentation - and messier because it requires multiple steps to fit the model. We did not include the regression model on the dPCA in the revised manuscript,

because adding another model made the paper even harder to read. We use this rationale to motivate our modeling choice, which we explain in the revised manuscript (lines 105-121, 235-266) Additionally, we found that focusing on the one method that could be applied to look at all the features of the data we analyzed (including dynamic spike history) rather than interpreting results from multiple different dimensionality reduction methods kept the writing clearer, even if the reader is already familiar with methods like dPCA. Taking this all together, we view the GMLM as a good way to more directly test our hypotheses about the low-dimensional activity while still remaining within a familiar family of generalized linear models.

Minor concerns, in the order they are encountered in the manuscript:

1) Is the monkey's best strategy to ignore the stimulus and simply keep holding the touch bar? On what fraction of trials would that get him a reward? And, does that fraction differ for the two tasks? (I realize the data have already been published, and that performance is largely good, so this is definitely a minor concern. But, it does give some indication as to whether the animal is always actively attending to the task, or maybe zoning out for occasional trials.)

This is a good question, and helped us realize that we needed to provide a clearer and more detailed explanation of the behavioral tasks. The reason is that, on non-match trials, the monkey is not rewarded for simply holding the lever throughout the entire trial. Instead, on non-match trials the monkey needs to release the lever during the presentation of the second test stimulus, which always matches the sample. Thus, the monkey must release the lever on every trial to receive a reward—during either the first or second test stimulus depending on which one matches the sample. If the monkey adopted a strategy of always holding the lever throughout the first test period, the monkey would perform at the chance level (50%), receiving a reward only on the 50% of trials that happened to be non-match trials. Additionally, the monkey is encouraged to release the bar because releasing results in being rewarded sooner. We have clarified the task design and level of chance performance in the revised results and methods sections. We have also included this information in the Figure 1 caption. Furthermore, Figure 6C (old figure number 7C) shows behavior divided into match and non-match trials.

Additionally, the monkey was ultimately required to release the touch bar to receive a reward on all trials. On non-match trials, the touch bar response was made during a second “test” stimulus (which was always a match) following the first (non-match) test stimulus. We only analyzed correct trials, and we did not analyze the neural activity during the second test stimulus on non-match trials (see Methods). Because the touch-bar response controlled the reward, the monkey was encouraged to release the bar to obtain rewards faster. While occasional lapse trials cannot be discounted, we believe the monkeys’ good performance reflects engagement.

2) There might be a better organization and flow if the focus is on the neural mechanisms whereby the animals might perform the DMC (and in comparison, DMS). Lots of insights into the perceptual and cognitive computations are offered in the paragraph beginning on line 102. But to get there, first we must go through the GMLM technique. A suggestion

to first discuss what you think is going on in the brain. As it stands, we really only get a clear sense for what you are looking for when we finally see it, in figure 5.

We have followed the Reviewer's suggestion, and we have reorganized the paper extensively to more clearly establish what features of population coding of the category task we are looking for across animals. The introduction discusses our hypothesis of task encoding differences in terms of category- and direction-tuned responses (lines 96-102) A new results subsection now first examines single-neuron tuning coding before going to dimensionality reduction to show we are interested in differences in category & direction coding (Section "Quantifying direction and category tuning in LIP"). We have also moved Figure 5 to Figure 3 to more quickly get to the subspaces and differences in coding questions as the Reviewer suggests.

3) In fig 2 it's unclear what is being fit and what is being input. It seems like the things to the left of the gray box are inputs, but then where are the recorded spike trains? Also, I'm confused by a discrepancy between the statement in the caption that "Because we do not include interactions between neurons here, this model can be applied to a set of single-neuron recordings" and the statement in the results "We aimed to describe how task-related responses in LIP are shared across neurons in a population by reducing the dimensionality of the eight LIP populations using GMLMs." How can the interactions not be modeled but responses shared across neurons can be described? The resolution, probably, is that there are "common motifs" that many neurons express, but one must think longer than is ideal to come to this realization.

We have now added text to the Figure 2 caption to explain what is being fit and where the spike trains. Briefly, the recorded spike trains are on the right side of the diagram: the spikes are the outputs (defined by the likelihood) of the encoding/regression model.

We appreciate the Reviewer's suggestion of the term "common motifs", and we think that it improved the description of what our model captures. We now state that the goal of our model is finding those motifs in the responses as suggested (lines 250-254) We now specify that we are not looking at interactions in the sense of noise correlations, but instead in the mean responses (or signal correlations).

4) In fig 3, the y axis of some of the temporal filters could be flipped (e.g., 1, 5, 6, 7) for a better visual match to the middle column, since those units are arbitrary, including in sign. In B, how should we interpret component 3 having smaller weights than component 2? Are the components being normalized with respect to each other? It seems like it might be interesting to see how the loading of a single unit changes across components.

Thanks for the suggestions, which we agree help clarify the results. We have flipped the axes to match as suggested. The revised Figure caption explains how the filters were normalized (the Figure is now in the supplementary as per the Reviewer's earlier suggestion; Supplementary Figure 5). The temporal filters were normalized to have the same vector length (1st column). The stimulus-direction weight vectors (which were multiplied onto the filters in the 2nd and 3rd

column) were also normalized to be unit length. The remaining neuron loading weights, given the normalized stimulus weights and temporal kernels, are shown in B. The components were ordered by the sum of squares of the neuron loading weights, which is similar (but not identical) to the ordering of singular values in PCA. However, we avoid saying that because the weights of this tensor decomposition cannot be interpreted as “percent variance” as with the singular values of a matrix. To address the question of how the loading of a single unit changes across components, the right three columns show the loadings for each component across 3 example units.

5) I don't really understand the distinction between the full GMLM and the individual-neuron GLMs. Here is a place where a comparison between what GMLM gives you and what more-common techniques like dPCA (or other directed / hypothesis-driven dimensionality reduction, where you look for the best population correlate of a known signal) would show. Also, a suggestion to build a GLM from one neuron and see how well it models other neurons.

The distinction between the GLM and the GMLM is a very good question that we now directly explain in the revised manuscript (lines 255-258) The GMLMs can be thought of as a special case of the GLM where the GLM filters for all neurons, which can be arranged as a multiway array (time x stimulus condition x neuron), has low-rank structure. As the rank of the GMLM tensor grows, it approaches GLMs fit to each neuron. The kernel/filter for one neuron (n) to one stimulus direction (d) at one time point in the low rank tensor is:

$k(t,d,n) = \sum_{r \in \{1, \dots, R\}} T_r(t) U_r(d) V_r(n)$ where T_r , U_r , V_r are the vectors representing the rth component.

As R increases, this sum can converge to any tensor, where the full-rank tensor can have any set of values for each $k(t,d,n)$, which gets us back to the GLM. By making this low-rank constraint of the filters across neurons & stimuli, the GMLM combines the GLM approach with dimensionality reduction in a single model without the need for multiple disconnected steps: we directly ask what are the main components that explain the spike trains and determine how they relate to task variables, which requires estimating fewer parameters.

We have included an additional analysis using dPCA for comparison (Results subsection “Comparison to demixed principal components analysis”; Figure 4, Supplementary Figure 9). Similarly, we performed PCA on the GLM filters - which are another estimator for the effect of stimulus on rate (instead of just the smoothed spike trains in dPCA) (Supplementary Figure 10). Both methods returned similar subspaces, but the GMLM approach is more direct.

We apologize that we don't fully understand what the reviewer suggests by building a GLM from one neuron to see how well it models other neurons. We have tried to answer this the best we can with the new PCA analysis on the GLM filters: the GLM fits themselves were low-rank (implying that the GLM for one cell can be constructed as the combination of filters from other cells).

6) In fig 4, what question is answered by the dimensionality analysis? Also, it might be more intuitive for readers unfamiliar with these tools to express the y axis in units of % variance explained, rather than the log likelihood of the model. Also, what does it mean to express the y axis as a "per trial" quantity?

We have clarified in the caption that this figure (now Figure 2C and Supplementary Figure 4) is asking if the responses are in fact low dimensional. This figure also quantifies how much of the response is explained by the task variables (category and motion). Additionally, the comparison of the "full GMLM" the cosine-tuned GMLM demonstrates that the cosine-direction tuning model provides a good approximation of the motion tuning in the data.

With our Poisson model for single trials, we cannot give the fraction of variance explained: we're not optimizing squared error of a PSTH. This has motivated the use of fraction log likelihood as a pseudo- R^2 . The pseudo- R^2 is comparable to the fraction of variance explained, but it uses the loss function in our model which respects that our observations are discrete spike counts and not continuous values (while the squared-error does not). Benjamin et al, 2018 is one example source motivating this choice. We have added some clarifications and references to explain this connection and give more intuition about the meaning of log likelihood improvement to the Figure caption, but we cannot change the wording to say "% variance explained" while remaining correct. The log likelihood can be computed separately for the spike train from each (cross-validated) trial. To put this in "per trial" units, we simply divided the total cross-validated log likelihood over all trials by the number of trials. This choice was to normalize over datasets, which included different numbers of trials.

5) Figure 5 is where the results start to come in. Devoting figs 2-4 to methods explanation, development, and validation is a lot of "real estate" for a paper whose focus is on scientific results. Is there a way to put some of this material into the supplement, and get to the results within the first couple of figures? (I say this knowing that I just provided a lot of specific queries about figs 2-4 above. Those concerns aren't completely side-stepped if the paper is restructured to be "results-first" rather than "technique-first".)

We have extensively reorganized the paper as described above in Major Comment #2 to address these organizational comments. We now get to the subspaces in Figure 3, rather than Figure 5, as we describe in response to Major Comment #2. This way, we get to the key results quicker.

6) Also in fig 5, how are the dimensions selected? Are these the most important ones for visualization, or are they the most-variance-explained dimensions?

The top 3 dimensions are given by a higher-order SVD of the low-dimensional response tensor to explain the most variance. This was briefly mentioned in the methods (subsection "Visualizing the GMLM parameters"), but we have added this to the Figure 3 caption for clarity.

7) In fig 7B, the "sample" and "test" plots look to be almost mirror-images. Is there an obligatory linking between these metrics?

We thank the Reviewer for pointing this out - the way these decoders are setup these two plots are indeed expected to be mirror images because the training/test sets for the two decoders are the same, but the test sets have opposite labels. We state this on lines 565-567 and we have simplified the figure to reflect this insight.

8) In fig 8, it's not clear what B is showing. This is meant to be the dynamic spike history, and yet it's shown as a static single vector. Does it become dynamic because it's weighed by the stimulus-timing kernels? I actually expected to see more of what's in D, where it's different depending on the time point, but maybe that's because it's the weighting that makes it dynamic? In any event, a clearer "expectation" going into this figure will be helpful.

We realize that this was unclear in the original manuscript. We have clarified that our proposed spike history is “dynamic” because the contribution (weighting) of past spikes on the instantaneous spike rate changes during the trial. In the low-dimensional GMLM, the spike history filter changes over time because it is scaled by stimulus-timing kernels (line 616-623). We accomplished this using a low-dimensional decomposition of a “full-dimensional” dynamic spike history model: a model that would allow the spike history filter to be different at every time point in the trial, for every neuron (which is impractical to estimate or visualize). The low-rank decomposition represents the main feature that changes in spike history in the population as the decomposition of the spike history filter and the spike-timing gain kernel.

Reviewer #2 (Remarks to the Author):

In this work, the authors examine population-level responses in LIP during a match-to-category task. They propose a novel method for dimensionality reduction that extends a generalized linear model (GLM) by incorporating low-rank tensor decomposition. This approach (termed generalized multilinear model, or GMLM) finds a low-dimensional set of descriptors (kernels) to fit the population activity. As the authors note, this approach to dimensionality reduction explicitly incorporates task variables (reach direction, category, etc.) and can be used for tasks with flexible timing or free behavior, and even if the population activity is not recorded simultaneously. They use this method to examine how responses in LIP during a match-to-category task differ depending on the animal's training history. They report that monkeys first trained on a match-to-sample task exhibited greater direction-specific responses in LIP than did monkeys only trained on a match-to-category task. Monkeys trained only on the match-to-category task showed higher category-dependent activity. The geometry of this category-specific population activity was also different across training histories. The neural results generally point to different strategies between the

two animal groups, which is supported by behavioral results that show a stronger direction-dependent effect on task success for the animals pre-trained on the match-to-sample task.

The question of how training (and/or overtraining) impacts neural activity is very relevant and an important next-step for the field. The paper is generally well written and the results (namely, the training-based heterogeneity of response characteristics in LIP) are fairly convincing. However, I have a few concerns about the implementation of the new method and absence of corroborating evidence regarding the scientific findings. As written, the manuscript attempts to both introduce a new computational approach and also use it to support a scientific finding. However, the lack of comparisons/validations using existing analyses makes it difficult to judge the validity of either one.

We greatly appreciate the reviewer's comments on the manuscript. We have responded to each of the reviewer's comments below.

Major comments:

1) In general, the new method proposed (GMLM) appears to be a useful tool for achieving low-dimensional descriptions of population activity collected during different sessions with variable trial timing. This flexibility with respect to task event timing is stated by the authors as one of the primary justifications for not applying existing methods. However, the results as presented appear to suggest that the responses can be trial-aligned and averaged. All major results of the paper (e.g. Figs 1, 6, 8) are provided in PSTH-style plots.

Given that this dataset seems perfectly appropriate for trial-averaged approaches, it would be nice to see some comparison to results from known methods. If the kernels obtained via GMLM reflect true underlying low-dimensional latent activity, I would expect to see similar results from a method like dPCA. I recognize that dPCA could not entirely replace the analysis here, because category and direction are completely correlated and not suited for dPCA. Still, my confidence in these results would be much higher if there was some comparison or validation of the results. For example, does the geometry subscribed by the GMLM kernels roughly match those found via PCA? Do other dimensionality reduction methods also show the same major/minor axis results in Figure 6? Validating the results that can be against known methods would strengthen the results that can purportedly only be achieved via GMLM (category/direction disentangling).

We agree with the Reviewer's comment, along with Reviewer 1, that a comparison with existing dimensionality reduction methods would substantially strengthen the current study. We have included a comparison with dPCA (Results section "Comparison to demixed principal components analysis"; Figure 4 and Supplementary Figure 9) and we found similar subspaces using dPCA with comparable category and direction coding properties. One important limitation

of dPCA is that looking for cosine direction & category tuning with dPCA required formulating a regression model on top of dPCA output. Because dPCA does not give such a decomposition on its own, it requires performing an analysis in stages: marginalization, dimensionality reduction, and regression (and noise can accumulate across these steps). In contrast, our model combines the steps into a single coherent framework. With dPCA (or PCA) augmented with a regression model, we found similar direction and category tuning results. However, we are hesitant to add this regression to the main paper because including yet another model did not help readability and did not impact our results.

We also highlight that results from our modeling approach better support our claims of disentangling of direction and category than is achieved by methods like dPCA (lines 340-362). Because direction determines category, these variables cannot be demixed under the dPCA assumption. Our model instead could use information across both the sample and test stimuli to perform dimensionality reduction. Specifically, we show that motion direction tuning is the same for the sample and test stimuli (consistent with bottom-up input) while category responses are distinct. By modeling these two stimulus presentations that have different cognitive meaning to the animal, we could increase our confidence in our ability to quantify the responses. Classic methods like PCA and dPCA which consider these two stimulus presentations independently cannot answer these questions as directly, and the differently timed touch-bar release across trials (according to the monkey's reaction times) would impact how effectively dPCA could characterize the test stimulus.

We also obtained similar subspaces to the GMLM by performing PCA on the GLM filters (as expected from the GMLM's formulation; Supplementary Figure 10). Our model is nested within the set of individual neuron GLMs, but it recovers the low-dimensional components by fitting directly to the data, without the intermediate single neuron GLMs. This enables us to address more complex questions, like the dynamic spike history model, but we agree that confirming the model's results with these classic approaches to the tractable sample stimulus encoding question was an important addition to the manuscript.

2) Similarly, the dynamic spike history result in Figure 8 is intriguing, but could use additional validation. Whether DMC-only animals exhibit stronger low-beta activity should certainly be testable outside of the GMLM framework using standard frequency spectra analysis. If the results confirm the same result, it would strengthen the GMLM approach. I don't know of another approach that can characterize both population-wide spike rate phenomena and spectral content, so this could be a very useful tool. However, I have no way of judging whether the dynamic spike history filter accurately reflects frequency band power.

This is a good suggestion. We have included standard auto-correlations of cells from the populations during the delay period (lines 654-659, Supplementary 16c). We also see signatures of the oscillations in the autocorrelation. We have clarified that the GMLM tries to correct for stimulus-dependent means while also characterizing this time-varying autocorrelation. This allowed us to characterize the differences in autocorrelations over the

course of the trial without predefined analysis windows. This was important for modeling monkey J's LIP activity, which shows a more transient response in the dynamic spike history after stimulus onset than observed in the other LIP populations.

3) The results are quite inconsistent across monkeys within the same group, but the discussion does not directly provide explanations other than possible strategy differences. Monkey H displayed significant category tuning during DMC late, but monkey D showed almost none. However, the major axis of the tuning model in Monkey D (but not monkey H) aligned with the test axis during DMC late. Are these differences between monkeys “real” or do they reflect a tradeoff during model fitting between direction and category terms? I can imagine a case in which two very similar sets of population activity give rise to very different model fits; one fit with a highly elongated tuning ellipse (2D) and the other with a near circular tuning ellipse and additional categorical component (3D). It is difficult to determine how the method itself and the choices made during fitting (such as chosen dimensionality) affect the results/interpretation.

We appreciate these very important points about consistency. We now explain that we used a Bayesian fit to account for the uncertainty for differentiating between the elongated ellipse and category differences by considering the range of coefficients that can account for the data (“Because category and direction are correlated, we applied a Bayesian analysis to take into account uncertainty in the direction and category encoding by exploring the tuning over the posterior distribution of the model parameters given the data, rather than only the best fitting parameters”, line 460-464). The reviewer is correct that confounds may remain if direction tuning isn't exactly captured by the sine/cosine parameterization, and we now state neuron sampling differences “within LIP between animals or sessions” as another source of variability between the populations (lines 748-751). We also emphasize that our study “included only a small number of animals” and the variation between animals means that we cannot gauge the magnitude of “training history effects relative to variance due to individual differences” (lines 755-757).

We have included several additional analyses that suggest our results are consistent with respect to our modeling choices (like dimensionality, as the reviewer pointed out):

1. The dPCA and PCA results discussed in comment #1.
2. We included single-neuron tuning analyses that show differences in category tuning between the two pairs of monkeys (ROC-based category tuning index; Figure 1D, Supplementary Figures 1-2).
3. We refit the GMLM fit with 16 components (more than considered previously) and found similar results (Supplementary Figure 13) to show that our choice of model rank was not a determining factor.
4. We have emphasized that our original analysis showing direction tuning was the same across both sample and test stimuli - consistent with a bottom-up input. Answering this question was one of the primary motivations for the model-based analysis (lines 105-121, 250-254).

4) The work done is rigorous and quite complete, but the large number of results in the manuscript make it difficult to address each with enough care and depth to feel that a satisfactory conclusion has been reached. For example, only one aspect of the major/minor axis result (Figure 6) was mentioned, and it was a single-monkey result related to alignment with the category axis. However, there are a number of results shown that receive little attention. Why does tuning appear to become more circular with DMC training? Does this represent a shift in strategy? Why are there no neural results shown for the test presentation (other than Fig 1c)? Why does the sign of the dynamic spike history weight flip after stimulus offset? If the results aren't going to be discussed, their inclusion only confuses the general point of training-induced differences.

We agree that these points were not adequately discussed or presented clearly enough. We have reorganized our paper following Reviewer 1's recommendations, as well as this Reviewer's comments and questions, and specifically expanded on these results:

- 1) Why does tuning appear to become more circular with DMC training? Does this represent a shift in strategy? We now discuss in greater detail about why these differences may arise. We see that in Monkey D, the ellipse of direction tuning is similarly elongated in both the DMS and DMC late populations - but the elongation is only aligned to category in the DMC late population (Figure 3; lines 483-493). This could also be due to neuron sampling differences (Discussion, lines 748-750). However, we note that previous work also found direction tuning in LIP aligned to categories in the category task (Engel, T. A., Chaisangmongkon, W., Freedman, D. J., & Wang, X. J. (2015). Choice-correlated activity fluctuations underlie learning of neuronal category representation. *Nat Comms*).
- 2) For example, only one aspect of the major/minor axis result (Figure 6) was mentioned. We have added more details (including those for the first point) to this section to discuss differences in the tuning between animals and training stages.
- 3) Why are there no neural results shown for the test presentation (other than Fig 1c)? We have revised our presentation of the results that included the test presentation. We point out that test period results are shown in the "Comparing the sample and test stimuli"; Figure 6 and Supplementary Figures 11, 14, and 15. We have emphasized that using both the sample and test stimulus response was important to better motivate that we could disentangle - at least partially - motion and category information (see response Major Comment 1). We also point to a recently published paper from our lab (Zhou, Rosen et al., eLife) that primarily focused on decision-related encoding in the test period in different data sets.
- 4) Why does the sign of the dynamic spike history weight flip after stimulus offset? We have added an explanation in this section about why the sign change occurs (lines 634-639). The main point we focus on is that the weight is different between sample and delay periods (not that the sign changes). This shows when a change in spike history occurs: spike history has a different contribution to each neuron's firing during stimulus presentation than during the working memory period.

We expect that the timing weights in our model will vary around 0. This is because the fixed spike history terms for each neuron determine the “mean” spike history. Including these mean spike history terms in the model therefore centers the temporal gain term to a mean of 0 (for identifiability and to encourage this centering, a 0 mean Gaussian prior is placed on the dynamic spike history terms).

In math, the total spike history filter for one neuron at time t in the trial, where $gain(t)$ is the dynamic spike history weight over time, is

$$H_{total}(t) = H_{spk} + gain(t) * H_{dynamic}$$

Where the neuron’s fixed spike history is H_{spk} and the dynamic spike history filter is $H_{dynamic}$.

If $gain$ has mean $\mu \neq 0$, this can be rewritten as

$$H_{total}(t) = (H_{spk} + \mu * H_{dynamic}) + (gain(t) - \mu) * H_{dynamic}$$

Setting

$$H_{spk}^* = H_{spk} + \mu * H_{dynamic}$$

$$gain^* = gain - \mu \text{ (now with 0 mean)}$$

$$H_{total}(t) = H_{spk}^* + gain^*(t) * H_{dynamic}$$

The prior zero mean Gaussian prior on the gain term will therefore encourage gain to have a mean closer to 0, with the individual neurons’ fixed spike history terms absorbing the mean. Because any constant added to the gain is not dynamic, it’s desirable to shrink out the mean from the dynamic spike history weight.

Minor comments:

71: reference should be to 1c

Thanks, this has been corrected.

Fig 3: What is the fit quality across neurons? The GMLM seems to leave little flexibility in the possible task-dependent modulations across the population. For example, any direction-based difference in activity can only take the temporal form of components 5 or 6. Is that correct? The single neuron reconstructions in Figure 3d look okay, but I would like to see a quantitative summary of how well the model fits the activity across the population.

We agree that this is an important issue to examine in more detail. We have added a single neuron comparison to the total model fits showing that the GMLM does not favor certain cells (Supplementary Figure 4d-e). The reviewer is correct that direction-based differences in activity can only take the temporal form of components 5 or 6 for that model fit: we have mentioned that this shows the low-dimensionality in direction tuning that was fit (but not enforced) by the model (which we now point out in the caption; now Supplementary Figure 5). We also note that the goal of the model isn’t to exactly fit each individual cell, but instead to recover components that give the gist of the responses (as with other dimensionality reduction methods like PCA).

207: “The category-only GMLM captured a greater percentage of the log likelihood over the no category model in the DMC late populations than in the DMS populations”
I am seeing 2.8 vs. 2.1 (monkey D) and 9.5 vs 3.3 (monkey H). Are those really significant?

We have included a supplementary panel to show that these values are significant. The numbers in monkey D are smaller because less of the activity is explained by direction (category and specific direction) in the monkey D DMC late population than monkey H. We now plot the normalized differences: the fraction of the direction-dependent activity explained by each model (Supplementary Figure 4C).

214: “Cosine direction tuning during the DMC task accounted for a larger improvement of the model fit over the category only model for the pretrained monkeys than the DMC-only monkeys”

I am confused about why the inclusion of cosine tuning improved the fit more for monkey H during DMC early/late (11.9/9.9%) than monkey D (1.6/1.9%). Figure 6 seems to suggest that monkey D only exhibits directional tuning, and not categorical. This seems counterintuitive and at least requires some explanation.

We appreciate this question, and we have added the following clarification (lines 400-406) This result can in part be explained by the dimensionality of the direction and category tuning. Monkey H shows 3 dimensional tuning, but monkey D shows only 2 dimensional tuning. Category is correlated with direction and “adding a category variable can capture aspects of motion tuning” in monkey D. Thus, there’s more left for cosine tuning to capture in monkey H than in monkey D. Additionally, the raw percentages was not the clearest choice to communicate this difference in the original manuscript, rather than a value normalized by the likelihood accounted for by the direction/category-independent model. We’ve added an additional supplementary panel that shows that the fraction of direction-dependent activity is similar (Distance of orange line between red/blue traces in Supplementary Figure 4a,c).

222: “The cosine direction tuning model was comparable to the full model for all populations”

Perhaps report % as compared to condition-only.

Thanks for the suggestion. We weren’t sure if the reviewer meant to say “category-only” instead of “condition-only”. If that’s the case, we have added this (lines 424-427).

266: “We conducted a Bayesian analysis of the GMLM’s sample stimulus subspace to take into account uncertainty in the model fit given the posterior distribution of the model parameters.”

What does this mean? Perhaps make a short reference to the methods.

Thanks for pointing out this confusing sentence. We expanded on this sentence to explain that our analysis took into account uncertainty in the direction and category tuning given in our model fit, rather than only using a single “best” fit (lines 460-464; See also the response to major comment #3) and referenced the methods section.

Reviewer #3 (Remarks to the Author):

General comments:

This manuscript describes neural recordings in area LIP during two kinds of manual Go/NoGo tasks: A delayed match-to-sample (DMS) task in which monkeys compare visual motion directions; and a delayed match-to-category (DMC) task in which they must simply indicate whether the motion directions fall within the same arbitrarily defined category. Two pairs of monkeys are studied: One pair that first learned the DMS task and later the DMC task, and a second pair that only learned the DMC task. Analyses are done by fitting neural activity with generalized multi-linear models (GMLM) that aim to reconstruct how neural activity is a sum of specific theoretical components, and then analyzing the properties of the fitted models. It is reported that stronger cosine tuning is found during the DMC task in the monkeys previously trained on the DMS task.

The key question addressed here is the question of how training history can change the properties of neural activity. In that light, it is interesting that cosine tuning is stronger in the monkeys with previous training in the DMS task. That is not very surprising of course, and perfectly compatible with the reasonable proposal that they retain some of the strategy they used to solve the DMS task when later trained in the DMC task. Indeed, their pattern of errors at category boundaries supports that notion, as if it is not quite clear to the pretrained monkeys exactly how the task has changed. I assume the hypothesis is that monkeys retain some neural properties that result from previous training and that this retention involves LIP, and this then predicts differences between the two pairs of animals. But obviously there must be differences in neural activity between the animals, if only to explain the differences in behavior. So is the key finding that these differences are seen

in LIP? Judging the importance of that is not really possible unless the LIP results are compared to other regions, such as MT or V1, and maybe hand-related regions like AIP, PMv or M1. How did these results compare to what was seen in PFC (refs 6 & 7)?

On that note, why was LIP chosen as the recording site to address the question about training history? This is a motion direction task (implicating motion sensitive areas like MT) in which responses are made manually (implicating hand-related regions like AIP, PMv, M1) and the monkey must fixate (i.e. reduce responses in saccade-related regions like LIP and FEF). Sure, many people have studied LIP during decisions, but those were decisions about eye movements. I would have guessed the place to go would be MT. I understand that the senior author has found interesting results in LIP in previous

studies, but what is the real hypothesis here? That LIP is a general place for working memory of visual motion? A general place for decisions about visual motion? I find these proposals very questionable, but regardless of my opinion, the hypothesis should be stated explicitly.

The reviewer raises an interesting and important set of questions about the rationale for focusing on LIP in the current study. Indeed, our laboratory has focused extensively on LIP for our studies of abstract visual categorization. While we have laid out the rationale for our focus on LIP in a number of recent experimental and review papers, which we now cite in the revised manuscript, we realize that we did not explain this rationale sufficiently in the original submission. The revised manuscript explains that the primary reason that we have focused so strongly on LIP in this and other studies of abstract categorical decisions is that we have also examined a large number of other candidate cortical areas. Comparisons of results among these regions suggests that LIP is the leading candidate (at least so far) in mediating these kinds of decisions. We elaborate on this point below.

First, the reviewer suggests that MT would seem to be an important brain area to consider as playing a major role in motion categorization. In fact, we started our investigation of neural mechanisms underlying motion-based categorical decisions by recording from MT. 16 years ago we published our first paper comparing activity in MT and LIP during this motion categorization task (Freedman and Assad, *Nature* 2006). This revealed that MT showed direction tuning, but not explicit category tuning. In contrast, LIP showed strong binary-like category tuning (as well as some direction tuning). This finding ruled out the idea that long-term training on this motion categorization task altered motion representations in MT. Instead, MT's role appeared to be representing specific directions in a veridical manner. Thus, we proposed at that time (and followed up extensively since) that the computation of motion categories occurs downstream from MT, in areas which receive input from MT such as LIP.

The reviewer asked about the role of PFC. That's another good question, and a hypothesis that we have considered and tested in previous experimental work from our lab (Swaminathan and Freedman, *Nature Neuroscience*, 2012). In that study, we directly compared neuronal activity in LIP and PFC in the same monkeys performing the same motion categorization task used here (i.e. not requiring saccadic eye movements to report decisions). What we found suggested that LIP played a preferential role in encoding the learned motion categories compared to PFC: 1) LIP showed shorter latency encoding of categories compared to PFC; 2) LIP showed stronger category encoding; 3) LIP activity was more predictive of the monkeys correct vs error trials compared to PFC.

The reviewer also asks whether we think LIP's role is specific to categorical decisions about motion. This is another very interesting question that we have wondered about and worked on. In short, we indeed believe that LIP's role in abstract categorical decisions extends beyond motion. In a previous study, we trained monkeys to switch (in blocks of trials) between shape categorization (pair-association) and motion categorization tasks, and found categorical encoding during decisions for both kinds of stimuli (Fitzgerald, Freedman, and Assad, *Nature*

Neuroscience, 2011). This suggests that LIP plays a more general role in flexible abstract decisions for visual stimulus features beyond motion.

Finally, we recently performed causal experiments to ask whether LIP actually plays a causal role in visual motion categorization (Zhou and Freedman, Science, 2019). We used reversible inactivation (muscimol) and found that inactivation specifically impaired the monkeys' categorical judgements about the category of motion stimuli. Interestingly, the inactivation produced only minor impacts on the monkeys' saccadic eye movement choices.

We have added additional discussion of the rationale for focusing on LIP both in the introduction and discussion sections in order to better motivate the current study (lines 75-99).

- 1) We first point to LIP's role in stimulus processing, not just in planning saccades or attention (lines 77-84). LIP shows motion selectivity, because it receives input from motion-selective sensory regions including MT (Lewis & Van Essen, 2000).
- 2) We state concretely that LIP is known to show abstract category-related activity (lines 84-86). We state that this contrasts with MT, which does not develop category-related activity in the same task (lines 86-87). Additionally, training affects MT's contribution to decision making tasks despite not explicitly representing the decision variables (Liu & Pack, 2017; Chowdhury & DeAngelis, 2008; lines 88-89).
- 3) LIP shows a causal role in the categorization task, as shown in our lab's recent inactivation study (Zhou & Freedman; lines 94-96)
- 4) LIP and PFC are anatomically connected and both contribute to cognitive tasks. We reference studies showing the working-memory and stimulus encoding in PFC are training dependent (e.g., Qi & Constantinidis, 2013; lines 89-94).

I have concerns about the way in which the authors analyze their data. In fact, they don't analyze the data directly, but perform a GMLM fit to build a model and then analyze the model instead. This approach is very precarious because it is entirely dependent on the assumptions that are put into the model, all of which must be true for the results to be interpretable. In particular, one can call the model "general", but it can only be as general as the components that the authors include. Here, they include the basic pieces of the paradigm (stimulus motion direction, category, timing) as if they assume that these are the components from which neural activity in LIP is constructed, through a simple linear summation, and nothing else need be considered. But where do those components come from? Furthermore, what if neurons are related to other things that are not included in the GMLM? For example, since LIP is a saccade-related region, one might expect that even in a task with constant fixation, like this one, neural activity is related to the relative placement of the stimulus with respect to the visual motion (how the motion aligns with the potential saccade vector toward the motion field). Other factors could also be involved, such as previous trial success, expected probability of correct answers, etc., all of which have been documented in LIP in many studies. The conclusions from such a model fit are very strongly based on what components are included, and leaving one out will change the

weights assigned to the others. So to trust such a model one would have to assume that all of the components are correct, none are missing, and that their linear summation is all that LIP does. Note that I'm not suggesting that the authors pile all kinds of other factors into their GMLM as well. What I'm saying is that the approach of doing GMLM fits and then viewing all of the data solely through that one lens suggests that the authors think they already know all of the components that must be included in a model of LIP. I'm not willing to go along with that.

We appreciate the reviewer's concerns. For the reasons we describe below (and in the revised manuscript) we believe that statistical methods with explicitly defined models are essential for accurately analyzing modern neural datasets. First, we would like to point out that, like our model, tuning curves are also statistical regression models which rely on similar assumptions. No analyses are truly without model-based assumptions — even if the model isn't written down explicitly. For example, common statistical analyses used in systems neuroscience like ANOVA are examples of general linear models. In many ways, our model is a fancier, regularized ANOVA (or MANCOVA) which is better suited for spike trains by using a Poisson likelihood and includes the population coding questions in all phases of the analysis (in contrast to tuning curve approaches which requires fitting models to each cell individually and then separately combining the population results). We believe our model uses the normal range of assumptions about data as other analyses. We favor formally taking a modeling approach because it requires that we make our assumptions about what we're testing in the data as openly and specifically as possible.

To answer where the components come from, they can be viewed as “common motifs” that appear in the data (we have added this terminology in response to Reviewer 1, lines 250-254). Following suggestions from Reviewers 1 & 2, we have included a comparison to demixed Principal Components Analysis (a common, if not as flexible, dimensionality reduction approach in neuroscience that shares a similar goal of finding the main features of population responses) in the revised manuscript to better motivate our particular take on dimensionality reduction in this paper.

The reviewer is absolutely correct that other variables could be involved or considered in the model or analysis. For some of those variables, like the saccade alignment, this was controlled (constant) in our task and therefore can arguably be left out of the model. For the many other variables that could contribute to LIP responses, we agree that it's always a possibility that more is going on than we're accounting for. However, those variables would impact interpretation of all other analyses and measures including tuning curves in a similar manner.

We note that the term “generalized” comes from its use in statistics for describing model class with a firmly established literature (e.g., the classic book *Generalized linear models* by McCullagh & Nelder), and GLMs are a well-established tool for performing statistical inference on data in neuroscience. The term GMLM has also been used in neuroscience, although for different applications (see Robinson et al., 2016). While the reviewer is by no means the first to

disagree with the term (including some of the statisticians who coined it), we feel we must stick with the field's terminology for clarity.

Unfortunately, all of the analyses in the manuscript rely exclusively on data processed through the GMLM. So if the authors' assumptions about the generality of the model are false, then their results are more about departures from the model than about the data itself, and hence, uninterpretable. Maybe some conclusions survive, but how do we know which ones? In my opinion, if the findings are robust then it should be possible to demonstrate them through other, independent analytical approaches. Preferably, these would minimize excess data processing. After all, if the LIP populations in the pretrained monkeys are more cosine tuned than those in the DMC-only monkeys, then it should be trivial to show that simply by plotting distributions of tuning parameters. Why was this not done?

We agree that more verifications of our approach help establish the validity and rigor of our modeling methods. First, we have included tuning curve analysis of individual neurons which do indicate population differences consistent with our original results (Results subsection "Quantifying direction and category tuning in LIP"; Figure 1D, Supplementary Figures 1-2). We use these results as motivation that we need to take a population-level approach to understand how these variables are organized across many cells over time in the trial (First section of the "Results"). We have included an analysis comparison using demixed PCA - a well established method for analyzing neural populations. We found very consistent results which validate our approach (Figure 4 and Supplementary Figure 9-10). We also found consistent results after increasing the number of components in the GMLM (Supplementary Figure 13), which verifies that our conclusions did not depend on our rank selection.

In general, I found the topic of this manuscript very interesting, but I'm not convinced by the manner in which the data was analyzed. If these results are correct, then it should be possible to demonstrate them through a method that does not rely on so many questionable assumptions. Unfortunately that would require rewriting the paper almost from the beginning.

Specific comments:

In figure 1B, monkey H's performance does not appear to be strongly affected by the change in the task. Also, monkey H's fraction correct in DMC early looks like a bimodal distribution. Is it because H's performance was improving rapidly (i.e., split of early-early and early-late), or was H exploring different strategies, some better than the other, on different sessions?

We think this is an interesting question too. Unfortunately it is difficult to answer, because we do not see an early-early early-late split in the behavior. Finding an early-late split during recordings was unlikely because the animals were trained for a long time on the category task prior to the DMC-early recordings (104 training sessions for monkey D and 70 for monkey H).

As a result, there's not a clear pattern of behavioral change over time. We think that future studies would need to be specially designed to really probe different strategies on single sessions during learning.

Line 235, "... The two DMC-only LIP populations showed primarily two-dimensional response..." This seems only true in monkey B. In monkey J the 3rd dimension is capturing something about the brightest red and darkest blue directions.

Thanks for pointing this out. We have fixed this to clarify that Monkey J does indeed show weak, 3rd dimensional direction tuning (lines 310-311).

Figure 3. Are these components meant to indicate something that is present in the real brain, or are they just the bases necessary to produce a good fit? I understand the algorithm just implies the latter, but are the authors interpreting them as the former? I note that the fits shown in Figure 3D are not really so great.

We now explain that the components of the model are the pieces that "best captures the common motifs in the mean responses to the task" (lines lines 250-254). This way, we can summarize and test what the population responses look like with a small, manageable set of values (in contrast to trying to view PSTHs from all neurons). These components provide a summary of features present in the brain's response, similar to principal components, but we place them inside a regression model to allow us to test for a wider range of features in the data (e.g., comparing direction/category responses across the sample and test stimuli even in the presence of the touch-bar release and spike history effects). By summarizing the datasets with a set of interpretable components, we aim to provide directions for new studies. For example, the differences in category & direction tuning we observed suggest that adding noise to the motion stimuli may impact pre-trained monkeys' performance on the categorization task more than the category-only monkeys. Additionally, our dynamic spike history results may guide computational & experimental studies about training-dependent network mechanisms for recruiting oscillatory dynamics to support working memory.

In figure 6B, the estimated vector norm for monkey B looks like it has transient directional tuning during the sample presentation period. Is that what the neural data looked like?

Yes, we think this is a good observation: some cells do indeed show transient direction tuning. We now include one in the example cells in Figure 1C (DMC-only top) to show this and we mention the transient direction tuning on lines 468-469).

Minor comments:

The behavioral results are presented before the results section.

Thanks for the suggestion. We have now moved the behavioral data to the first subsection of the results as part of the extensively reorganized revision. We also specifically state that all animals “learned to perform the categorization task at high levels” to motivate that the observed differences aren’t simply due to one pair of animals learning the task poorly.

Figure 3, left-align the subplot titles C and B.

Figure 5, caption, last lines "Supplementary Fig. 5 shows the three-dimensional trajectories (of the cosine-tuned model) as a function of time" (remove [are shown]).

Line 286-293, if this is about figure 6B, mentioning so here instead of at line 265 would be easier to follow.

Figure 6: Misspelled “Pseudopopulation”.

Line 325-327, "We therefore compared the LIP responses to the test stimulus to the population responses to the sample stimulus." This sentence is hard to follow. Please simplify.

Line 543 "B, and N = 29..." (capitalize B).

Line 592, "functions of the stimulus basis (Supplementary Fig. 1j, middle)." (add closing parenthesis)

Line 637, "x(cos)" (replace sin with cos)

We have fixed these points, thanks!

The supplementary figures are mentioned almost as if they are not supplementary, but essential to understanding the main text.

Thanks for the suggestion. We have taken out several references as suggested to keep them to a minimum to not disrupt the main text.

REVIEWERS' COMMENTS

Reviewer #2 (Remarks to the Author):

I would like to congratulate the authors for a wonderful reworking of this manuscript. They have thoughtfully addressed all of the major reviewer concerns, and I think the paper has benefited greatly from it. Due to the improved methodological description of GMLM and comparisons to existing techniques, I am much more comfortable with the results and the conclusions drawn from them. The only comment I have is that I found Supplementary Figure 4A to be perhaps the most compelling visual representation of the main result of the paper, but only a reduced version of it exists in the main text (and as panel C in a methodology-based figure, at that). I would recommend making Figure 2A,B its own figure, and combining Figure 2C,D with Supplementary Figure 4A to make a separate results figure. However, I certainly do not view this as a requirement for publication. I think this paper will greatly contribute to the field and lead to more thoughtful consideration of how training experience shapes observed population responses.

The reviewer's comments are in bold and our response is in normal font.

Response to Reviewer 2

I would like to congratulate the authors for a wonderful reworking of this manuscript. They have thoughtfully addressed all of the major reviewer concerns, and I think the paper has benefited greatly from it. Due to the improved methodological description of GMLM and comparisons to existing techniques, I am much more comfortable with the results and the conclusions drawn from them. The only comment I have is that I found Supplementary Figure 4A to be perhaps the most compelling visual representation of the main result of the paper, but only a reduced version of it exists in the main text (and as panel C in a methodology-based figure, at that). I would recommend making Figure 2A,B its own figure, and combining Figure 2C,D with Supplementary Figure 4A to make a separate results figure. However, I certainly do not view this as a requirement for publication. I think this paper will greatly contribute to the field and lead to more thoughtful consideration of how training experience shapes observed population responses.

We thank the reviewer for their effort reviewing our revised manuscript and their encouraging words. We have reorganized our figure presentation of the model selection figures as recommended and we agree that it better highlights our results.